# TimeBase: The Power of Minimalism in Long-term Time Series Forecasting.

## Abstract

Long-term time series forecasting (LTSF) has traditionally relied on models with large parameters to capture extended temporal dependencies. However, time series data, unlike high-dimensional images or text, often exhibit strong periodicity and low-rank structures, especially in long forecasting horizons. This characteristic can lead many models focusing on redundant patterns, resulting in inefficient use of computational resources. In this paper, we introduce TimeBase, an ultra-lightweight network with fewer than $0.4k$ parameters, designed to harness the power of minimalism in LTSF. TimeBase extracts core periodic features by leveraging full-rank typical period representations under orthogonality constraints, enabling accurate prediction of future cycles. Extensive experiments on real-world datasets demonstrate that TimeBase not only achieves minimalism in both model size and computational cost, reducing MACs by 35x and parameter counts by over 1000 times compared to standard linear models, but also wins state-of-the-art forecasting performance, ranking Top1-Top5 in all 28 prediction settings. Additionally, TimeBase can also serve as a very effective plug-and-play tool for patch-based forecasting methods, enabling extreme complexity reduction without compromising prediction accuracy. Code is available at https://anonymous.4open.science/r/TimeBase-fixbug.

## 1 Introduction

Long-term time series forecasting (LTSF) has been studied with significant interest in various domains, ranging from energy management, traffic accident preservation, and extreme disaster warning Wu et al. (2021); Zhou et al. (2023b); Lin et al. (2023a); Wang et al. (2023). With the rapid advancement of deep learning, an increasing number of models have been proposed, including MLP-based Liu et al. (2022); Huang et al. (2024a), RNN-based Lin et al. (2023b), and Transformer-based Liu et al. (2021a); Zhang & Yan (2023), approaches, all of which employ thousands to millions of parameters to capture long-range dependencies and forecast future outcomes.

Generally, a higher number of parameters increases the model capacity, which can lead to better predictive performance Zhou et al. (2023a). In the fields of computer vision (CV) and natural language processing (NLP), large models have achieved significant success He et al. (2016); Liu et al. (2023). For instance, Vision Transformers (ViT) Dosovitskiy (2020) have demonstrated outstanding capabilities in image recognition, while large language models (LLM) have made breakthrough advances across various language tasks Devlin (2018); Radford et al. (2019). Recently, large models are being explored for LTSF to capture complex temporal patterns and long-range dependencies Jin et al. (2023). For example, some LLM-based methods are proposed with tens of billions of parameters. However, despite their impressive performance on specific forecasting tasks, these models suffer from high computational costs, interpretability challenges, and resource-intensive requirements.

In fact, images and text, as high-dimensional data, contain multiple dependencies and complex underlying physical rules Liu et al. (2021b), which necessitate the use of more parameters to model their rich semantic structures. However, as shown in Figure 1(a), one-dimensional time series data is typically much more regular, exhibiting obvious periodicity. Moreover, in long-term time series data, this regularity can even manifest as low-rank characteristics Liu et al. (2012), as different cycles often exhibit similar temporal patterns Jones & Brelsford (1967); Hochreiter & Schmidhuber (1997). This raises an important question: Is it truly necessary to employ such a large number of parameters to learn these regular time series patterns Tan et al. (2024)?

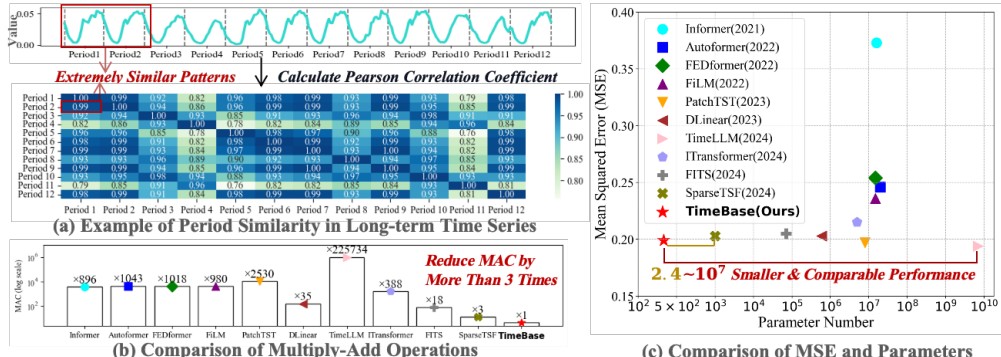

Figure 1: (a) Time series of real-world traffic reveals a clear periodicity, with highly similar patterns recurring in each cycle. (b) Comparison of multiply-add operations (MAC) in log scale. (c) Comparison of MSE and parameters between TimeBase and other mainstream methods on the Electricity dataset, with a forecast horizon of 720. The results from (b) and (c) demonstrate that by leveraging the periodic characteristics of time series, it is possible to achieve the desired forecasting performance using only a minimal model size.

In this study, we design an extremely lightweight time series forecasting network, TimeBase, which is centered around basis period extraction and period-level forecasting. As illustrated in Figure 1(b) and (c), TimeBase utilizes only $0.39k$ parameters, reducing MACs by 35 times and parameter count by more than $10^3$ times compared to vanilla linear model. In seven real-world datasets, TimeBase demonstrates superior predictive performance. Additionally, TimeBase can also serve as a very effective plug-and-play tool for patch-based forecasting methods, enabling extreme complexity reduction, i.e., 77.74%~93.03% for PatchTST in MACs, without compromising prediction accuracy. Our contributions can be summarized as follows:

- Considering the periodic characteristics of time series and the similarities between cycles, we demonstrate that basis period extraction is an effective approach for LTSF. This method can significantly reduce the unavoidable ultra-high complexity and large model parameters associated with current LTSF models.

- Based on basis period extraction, we propose TimeBase, which is currently the lightest time series forecasting model. It requires only $0.39k$ parameters, achieving a threefold reduction in MAC compared to the previously lightest model and a 2.4-fold decrease in the number of parameters, making it highly applicable to real-world forecasting scenarios.

- TimeBase not only maintains an extremely small model size but also achieves competitive forecasting performance across seven real-world datasets. Specifically, TimeBase ranks in the Top1–Top5 in all 28 prediction settings when compared to the ten state-of-the-art baselines, and it also achieves state-of-the-art performance in few-shot and zero-shot scenarios.

## 2 RELATED WORK

Long-term time series forecasting (LTSF) aims to predict future sequences of considerable length using extended historical windows. The advancement of deep learning has significantly enhanced the accuracy of LTSF, with various foundational models, such as Transformers Zhou et al. (2021); Zhang & Yan (2023), Temporal Convolutional Networks (TCNs) Luo & Wang (2024), and Recurrent Neural Networks (RNNs) Lin et al. (2023b), being employed to design long-term forecasting networks.

These models are designed based on the inherent properties of time series, such as series decomposition Wu et al. (2021), frequency domain Xu et al. (2024), and periodic characteristics Wu et al. (2023). As to series decomposition, for instance, Autoformer Wu et al. (2021) introduces a series decomposition block that utilizes moving average techniques to decompose complex temporal variations into seasonal and trend components, each undergoing separate time series modeling. Additionally, FEDformer Zhou et al. (2022) further enhances the representation capabilities of the series decomposition block by employing multiple kernels moving average to decompose data at

various granularities, thereby improving forecasting performance. Considering the frequency domain characteristics of time series, FITS Xu et al. (2024) operates on the principle that time series can be manipulated through interpolation in the complex frequency domain, achieving performance comparable to state-of-the-art models for time series forecasting. On the other hand, periodicity is a significant factor considered by many LTSF methods. TimesNet Wu et al. (2023)proposes the use of Fourier Transform to capture multiple periodic lengths of time series, expanding one-dimensional time series into several two-dimensional components, which are processed through two-dimensional networks to handle high-dimensional data. CrossGNN Huang et al. (2024b) employs moving average techniques based on periodicity to expand single-granularity time series data into multi-granularity data, enriching the information contained within the dataset. SparseTSF Lin et al. (2024) directly utilizes the prior periodicity of time series to transform LTSF at the time-point level into segmented prediction tasks at the periodic level, thereby reducing the scale of network parameters.

On the other hand, many MLP-based models have emerged, aiming to achieve lightweight forecasting solutions Wang et al. (2024a). DLinear Zeng et al. (2023) introduces a linear model based on trend and seasonal decomposition, whose competitive forecasting performance empirically demonstrated the feasibility of using MLPs for LTSF. Following this, TiDE Das et al. provides theoretical proof that the simplest linear analogue could achieve near-optimal error rates for linear dynamical systems. Later, numerous Mixer-based works emerge, such as MTS-Mixer Li et al. (2023), TSMixer Ekambaram et al. (2023) and HDMixer Huang et al. (2024a), which stack standard MLP layers to efficiently capture correlations across different dimensions of multivariate time series. Furthermore, Koopa Liu et al. (2024b) addresses the challenge of dynamic and unstable time series systems by disentangling time-variant and time-invariant components using Fourier filters and designing a Koopman Predictor to advance the respective dynamics. TimeMixer Wang et al. (2024b) tackles the issue of different granularity levels in micro and macro series by proposing mixing blocks, fully leveraging disentangled multi-scale series in both past extraction and future prediction phases. These works represent efficient time series forecasting models based on MLP structures ($1.03\,\text{M} \sim 31.07\,\text{M}$). However, it still remains challenging when faced with stricter deployment constraints on edge devices and higher efficiency demands Chatfield (2013). To address this, We propose TimeBase to sigficantly reduce data complexity by extracting basis periods, transforming time-step-level prediction tasks into segment-level (or period-level) forecasting tasks. Our approach requires only $0.06\,\text{k} \sim 0.39\,\text{k}$ parameters to achieve competitive predictive performance. More detailed differences between TimeBase and MLP-based models is summarized at Appendix A

## 3 METHOD

### 3.1 PROBLEM DEFINITION

In LTSF, the objective is to predict future values over an extended time horizon based on very long look-back windows. Formally, let $\mathbf{X} = [x_1, x_2, ..., x_T] \in \mathbb{R}^T$ denote the historical time series data, where $T \gg 1$ is the length of look-back window. The goal is to forecast the future values $\mathbf{Y} = [x_{T+1}, x_{T+2}, ..., x_{T+L}] \in \mathbb{R}^L$ with a forecasting horizon $L \gg 1$. However, the exceptionally long horizon scale $T$ and $L$ substantially increases model size, leading to a rapid and considerable growth in the number of parameters, which may be unnecessary for time series data that follow simple and regular patterns. Consequently, our focus shifts to designing models that not only deliver robust and efficient performance but also remain extremely lightweight.

### 3.2 TIMEBASE

In practical, regular time series often exhibit prominent periodic patterns Lin et al. (2024), with approximate low-rank structures across these periods Jones & Brelsford (1967). For example, traffic flow typically follows a daily period, with similar patterns recurring each day. To effectively leverage period characteristics and accomplish efficient forecasting, we propose **TimeBase**, implemented through **Basis Extraction** and **Period Forecasting** by two extremely small-scale linear layers. This approach drastically **reduces the model parameters to the hundred level** while maintaining state-of-the-art (SOTA) forecasting performance. Most existing multivariate time series are homogeneous, meaning that each sequence within the dataset exhibits similar periodicity. This characteristic allows them to be organized as a unified multivariate time series. Based on this property, we employ the Channel Independence Nie et al. (2023) to simplify the forecasting of MTS data into separate univariate forecasting tasks. An overview of TimeBase is shown in Figure 2.

Figure 2: Overview of TimeBase. The core of TimeBase lies in extracting basis periods, utilizing full-rank typical period features under orthogonality constraints to accomplish segmentation-level forecasting.

### 3.2.1 BASIS PERIOD RECONSTRUCTION

First, we extract the period $P$ from the time series, which can be categorized into two scenarios: **(1) The time series has a predefined prior period:** For instance, in domains like electricity or traffic, the cyclic patterns often follow a daily periodicity, allowing us to directly assign $P = 24$ as the known period from hourly sampled data. **(2) The time series lacks significant prior knowledge of its period:** In this case, we can perform a period analysis, i.e., Fast Fourier Transformation (FFT), on the available dataset, such as the training set $\mathbf{X}_{\text{train}}$, to determine the corresponding period Wu et al. (2023). Based on the period $P$, we segment the one-dimensional time series $\mathbf{X} \in \mathbb{R}^T$ into $N = \lceil \frac{T}{P} \rceil$ sub-sequences, denoted as $\mathbf{X}_{\text{his}} = [X_1, X_2, ..., X_N] \in \mathbb{R}^{N \times P}$, each of length $P$. When the length of $X_N$ is insufficient to meet $P$, the corresponding values from $X_{N-1}$ will be used to fill in the gaps. The segmentation operation can be represented as:

$$\mathbf{X}_{\text{his}} = \text{Segment}_{[N,P]}(\mathbf{X}) \tag{1}$$

where $N$ and $P$ in $\text{Segment}_{[N,P]}(\cdot)$ represent the number of rows and columns of the transformed 2D matrix. The maximized rank of the matrix $\mathbf{X}_{\text{his}}$ is $R_{max} = \min(N, P)$. Given that typical time series exhibit similar temporal patterns across periods, we have $R \ll \min(N, P)$ Jones & Brelsford (1967). However, directly utilizing low-rank time-series data for prediction can result in unnecessary resource consumption and an oversized model. Fortunately, in structured time series data, a representative periodic pattern can be identified, referred to as the basis period $\mathbf{X}_{\text{basis}} \in \mathbb{R}^{R \times P}$ which can be utilized to capture compact information and minimize the model size. Just as any vector in a coordinate system can be represented as a linear combination of its basis vectors, the combination of basis periods in specific time series can represent its any periodic pattern Hochreiter & Schmidhuber (1997). Conversely, we can approximate the full-rank basis periods using the low-rank temporal periods:

$$\mathbf{X}_{\text{basis}} = \text{BasisExtract}(\mathbf{X}_{\text{his}}) \tag{2}$$

where $\text{BasisExtract}(\cdot)$ is implemented by a simple linear layer. Formally, $\mathbf{X}_{\text{his}}$ can be expressed as a linear combination of basis components, represented as $\mathbf{X}_{\text{his}}^{\top} = \mathbf{X}_{\text{basis}}^{\top} W_E + B$, where $W_E \in \mathbb{R}^{R \times N}$ is the combined weight and the bias term $B \in \mathbb{R}^{R \times N}$ denote the temporal noise $\epsilon$. By rearranging, we derive $\mathbf{X}_{\text{basis}}^T = \mathbf{X}_{\text{his}}^{\top} W^{\dagger} - B W^{\dagger}$. Thus, the objective of Eq. (2) is to learn a linear layer of $W_{\text{his}} = W^{\dagger}$ and $B_{\text{his}} = -B W^{\dagger}$, allowing for an accurate approximation of the basis periods. Next, we combine the period basis to generate the future periods as forecasting:

$$\mathbf{X}_{\text{pred}} = \text{PeriodForecast}(\mathbf{X}_{\text{basis}}) \tag{3}$$

Here, $\mathbf{X}_{\text{pred}} \in \mathbb{R}^{N' \times P}$ represents the future periods, where $N' = \lceil \frac{L}{P} \rceil$ denotes the number of future periods. The operation $\text{PeriodForecast}(\cdot)$, implemented also through a linear layer, aggregates the basis periods for forecasting. Finally, $\mathbf{X}_{\text{pred}}$ is unfolded to produce the prediction result $\mathbf{Y} \in \mathbb{R}^L$:

$$\mathbf{Y} = \text{Flatten}(\mathbf{X}_{\text{pred}})_{1:L} \tag{4}$$

### 3.2.2 BASIS ORTHOGONAL RESTRICTION

To ensure the learned $\mathbf{X}_{\text{basis}}$ effectively captures the essential temporal patterns and serves as a true set of basis periods, an orthogonal constraint is needed. In this context, enforcing orthogonality on

$\mathbf{X}_{\text{basis}}$ not only promotes a more interpretable representation but also improves model robustness by encouraging decorrelated temporal features.

Thus, we introduce an orthogonal regularization term into the loss function. Specifically, we penalize the deviation of $\mathbf{X}_{\text{basis}}$ from an orthogonal set by adding a regularization loss $\mathcal{L}_{\text{orth}}$, defined as:

$$\mathbf{G} = \mathbf{X}_{\text{basis}}^{\top}\mathbf{X}_{\text{basis}} \tag{5}$$

$$\mathcal{L}_{\text{orth}} = \|\mathbf{G} - \text{diag}(\mathbf{G})\|_F^2 \tag{6}$$

where $\mathbf{G}$ is the gram matrix of $\mathbf{X}_{\text{basis}}$ and $\|\cdot\|_F$ denotes the Frobenius norm. This term encourages $\mathbf{X}_{\text{basis}}$ to approach an orthogonal configuration, ensuring that each basis vector captures unique and uncorrelated temporal patterns. The overall training objective is then updated to:

$$\mathcal{L} = \mathcal{L}_{\text{prediction}} + \lambda_{\text{orth}}\mathcal{L}_{\text{orth}} \tag{7}$$

Here, $\mathcal{L}_{\text{prediction}}$ represents the original prediction loss, i.e., mean squared error (MSE) for regression, and $\lambda_{\text{orth}}$ is a hyperparameter that controls the weight of the orthogonal regularization term. By tuning $\lambda_{\text{orth}}$, we balance the trade-off between prediction accuracy and enforcing the orthogonality constraint.

### 3.3 PARAMETER SCALE OF TIMEBASE

**Theorem 1** (Parameter Scale of TimeBase). *Let $T$ denote the length of look-back window, $L$ is the length of the forecast period, $P$ represents the length of the periodicity, and $R$ gives the number of basis periods. The parameter scale of TimeBase can be expressed as:*

$$Number = \lceil\frac{T + L + P}{P}\rceil \times R + \lceil\frac{L}{P}\rceil \tag{8}$$

*where $\lceil\cdot\rceil$ denotes the ceiling function, which rounds up to the nearest integer. Detailed derivation is available in Appendix B.*

Theorem 1 demonstrates that the parameter scale of TimeBase is linearly dependent on both the length of the look-back window and the forecast horizon. In LTSF, where $T$ and $L$ are both set to 720, the parameter number of TimeBase, given by $\lceil\frac{T+L+P}{P}\rceil \times R + \lceil\frac{L}{P}\rceil$, is significantly smaller than both $\frac{T \times L}{P^2}$ of SparseTSF Lin et al. (2024) and $2 \times T \times L$ of DLinear Zeng et al. (2023).

## 4 EFFECTIVENESS ANALYSIS OF TIMEBASE

In this section, we provide an effectiveness analysis of TimeBase on long-term time series with obvious period, and error bound on more general time series (i.e., non-period). Detailed derivation is available in Appendix B.

**Definition 1.** *For a long-term time series $\mathbf{X} = [x_1, x_2, \ldots, x_T] \in \mathbb{R}^T$, where the time length $T$ is very large and a periodicity $P$ exists, the series can be divided into $N = \lceil\frac{T}{P}\rceil$ sub-sequences $[X_1, X_2, \ldots, X_N]$, and can be represented as $\mathbf{X}_{his} \in \mathbb{R}^{N \times P}$.*

**Low Rank Characteristics** Let $\mathbf{X}_{\text{his}} \in \mathbb{R}^{N \times P}$ be a matrix with singular value decomposition (SVD) given by $\mathbf{X}_{\text{his}} = U\Sigma V^T$, where $U$ is an $N \times N$ orthogonal matrix, $\Sigma$ is an $N \times P$ diagonal matrix containing the singular values, and $V$ is an $P \times P$ orthogonal matrix. We can establish a threshold to compute the approximate rank of this matrix.

$$\text{Rank}(\mathbf{X}_{\text{his}}) \approx \#\{\sigma_i \in \Sigma : \sigma_i > \epsilon\} \tag{9}$$

where $\sigma_i$ is the singular values of $\mathbf{X}_{\text{his}}$ for a small threshold $\epsilon > 0$, and $\#\{\cdot\}$ is the number.

Figure 3 illustrates the singular value distribution from real-world long-term time series data. The rapid decay of the singular values indicates that a large portion of the matrix's information is captured by only a few dominant components, thereby confirming its low-rank nature. Thus, Due to the extreme similarity in temporal patterns across periods, long-term time series $\mathbf{X}_{\text{his}} \in \mathbb{R}^{N \times P}$ often exhibits approximate low-rank characteristics:

$$\text{Rank}(\mathbf{X}_{\text{his}}) \ll \min(N, P) \tag{10}$$

**Definition 2.** *For a period $X_n \in \mathbb{R}^P$, it can be decomposed into a set of basis periods $E = [e_1, e_2, \ldots, e_R] \in \mathbb{R}^{R \times P}$, along with a noise (or residual) term $\epsilon_n \in \mathbf{R}$:*

$$X_n = \sum_{i=1}^{R} e_i + \epsilon_n \tag{11}$$

*where $R$ denotes the number of basis period in specific time series data.*

**Effectiveness of TimeBase**   Based on Definitions 1 and 2, the predicted time series $\mathbf{X}_{\text{pred}}^T$ can be obtained as:

$$\begin{aligned} \mathbf{X}_{\text{pred}}^T &= E^T \times W_{\text{pred}} + B_{\text{pred}} \\ &\approx (\mathbf{X}_{\text{his}}^T \times W_{\text{his}} + B_{\text{his}}) \times W_{\text{pred}} + B_{\text{pred}} \end{aligned} \tag{12}$$

where $W_{\text{his}} = W_E^\dagger$ and $B_{\text{his}} = -B \times W_E^\dagger$ in $\mathbf{X}_{\text{his}}^T = E^T \times W_E + B$. It demonstrates that TimeBase transforms the time series forecasting task into a basis period learning problem. By approximating $\mathbf{X}_{\text{his}}^T \times W_{\text{his}} + B_{\text{his}}$ to $E^T$, the model learns well-represented periodic basis vectors, which are then used to reconstruct $\mathbf{X}_{\text{pred}}$ based on $E^T$.

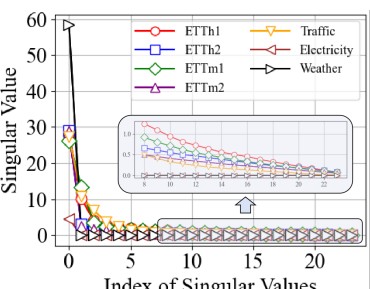

**Generalization of TimeBase**   Given any historical sequence $\mathbf{X}_{\text{his}} \in \mathbb{R}^{N \times P}$, and its corresponding learned basis vectors $\mathbf{E} \in \mathbb{R}^{R \times P}$, the upper fitting error of the model can be expressed as:

$$\|\mathbf{X}_{\text{pred}}^T - \mathbf{E}^T W_{\text{pred}} - B_{\text{pred}}\|_2 \le \frac{1}{\lambda_{\min}(\mathbf{E}\mathbf{E}^T)} \|\mathbf{X}_{\text{pred}}\|_2, \tag{13}$$

Figure 3: The singular value distribution of real-world period-segmented long-term time series. The rapid decay indicates the low-rank characteristics.

where $\lambda_{\min}(\mathbf{E}\mathbf{E}^T)$ denotes the smallest eigenvalue of the Gram matrix $\mathbf{E}\mathbf{E}^T$. This result highlights that the generalization capability of TimeBase to arbitrary time series relies on learning a well-repres'ented basis matrix $\mathbf{E}$. If $\mathbf{E}$ exhibits a favorable eigenvalue distribution (i.e., $\lambda_{\min}(\mathbf{E}\mathbf{E}^T)$ is large), the upper bound on prediction error is lower, highlighting the importance of a high-quality basis vector space and the necessity of orthogonal constraint.

## 5  EXPERIMENT

In this section, we demonstrate the advantages of TimeBase on mainstream LTSF benchmarks, both in prediction performance and running efficiency. Additionally, we conduct further analysis through basis visualization, parameter count, and model architecture to validate the effectiveness of Time-Base. Experimental results showcasing TimeBase as a plug-and-play tool for significant complexity reduction in patch-based methods are provided in Appendix M.

### 5.1  EXPERIMENT SETUPS

**Datasets and Baselines**   We conduct experiments on seven widely-used and publicly available real-world datasets, including ETTh1, ETTh2, ETTm1, ETTm2[1], Weather[2], Electricity[3], and Traffic[4]. To validate the effectiveness of TimeBase, we compare it against 10 state-of-the-art time series models, including ITransformer Liu et al. (2024a), FITS Xu et al. (2024), SparseTSF Lin et al. (2024), TimeLLM Jin et al. (2024), PatchTST Nie et al. (2023), DLinear Zeng et al. (2023), TimesNet Wu et al. (2023), FEDformer Zhou et al. (2022), Autoformer Wu et al. (2021), and Informer Zhou et al. (2021). **To ensure fair comparison, we have addressed the following concerns: (1) the "drop_last=True" bug in "test_loader" is corrected to "drop_last=False" in all models to avoid insufficient evaluation; (2) input lengths of all models are set to 720.**

---

[1] https://github.com/zhouhaoyi/ETDataset

[2] https://www.bgc-jena.mpg.de/wetter

[3] https://archive.ics.uci.edu/ml/datasets

[4] https://pems.dot.ca.gov/

**Implementation Details** In TimeBase, Channel Independence Nie et al. (2023) is involved to simplify the multivariate forecasting process to univariate time series forecasting. For datasets with shorter periodicity, such as ETTh1, ETTh2, Electricity, and Traffic, we use a prior period $P = 24$ for subsequence segmentation and set the number of basis periods $R = 6$ for efficient basis extraction. For datasets with extremely long periods (which exceed the look-back window), such as ETTm1, ETTm2, and Weather, we use smaller $P$ values and larger $R$ values for effective basis extraction. For the orthogonality loss weight $\lambda_{orth}$, due to varying loss scales across datasets, we perform hyperparameter searches over $\lambda_{orth} \in [0.04, 0.08, 0.12, 0.16, 0.2]$ for each dataset. Due to TimeBase's extremely small parameter count, and the need to reconstruct the basis period based on respectively longer historical information, we choose a larger input length , i.e., $L = 720$. Section 5.4 presents the sensitivity analysis on the number of basis periods $R$ and the orthogonality loss weight $\lambda_{orth}$. More implementation details are provided in the Appendix F.

## 5.2 MAIN RESULTS

Table 1: MSE results of long-term time series forecasting comparing TimeBase with other baselines on short-period datasets, i.e. $P = 24$, shorter than the input length $L = 720$. The top 4 results are highlighted in bold.

| Dataset | ETTh1 | | | | ETTh2 | | | | Electricity | | | | Traffic | | | |
|---|---|---|---|---|---|---|---|---|---|---|---|---|---|---|---|---|
| Horizon | 96 | 192 | 336 | 720 | 96 | 192 | 336 | 720 | 96 | 192 | 336 | 720 | 96 | 192 | 336 | 720 |
| Informer(2021) | 1.269 | 1.487 | 1.544 | 1.481 | 5.189 | 6.514 | 5.284 | 4.955 | 0.395 | 0.405 | 0.404 | 0.429 | 0.829 | 0.902 | 0.949 | 1.430 |
| Autoformer(2021) | 0.555 | 0.599 | 0.853 | 0.899 | 0.541 | 1.207 | 0.825 | 1.772 | 0.225 | 0.223 | 0.233 | 0.261 | 0.668 | 0.703 | 0.666 | 0.697 |
| FEDformer(2022) | 0.485 | 0.481 | 0.522 | 0.604 | 0.401 | 0.425 | 0.427 | 0.462 | 0.226 | 0.220 | 0.224 | 0.271 | 0.664 | 0.613 | 0.612 | 0.664 |
| TimesNet(2023) | 0.437 | 0.456 | 0.494 | 0.632 | 0.349 | 0.500 | 0.445 | 0.438 | 0.202 | 0.218 | 0.232 | 0.299 | 0.605 | 0.627 | 0.631 | 0.700 |
| PatchTST(2023) | **0.377** | **0.413** | **0.436** | 0.455 | **0.276** | **0.342** | **0.364** | **0.395** | 0.141 | 0.156 | 0.172 | **0.207** | **0.363** | **0.382** | **0.399** | **0.432** |
| DLinear(2023) | **0.378** | **0.415** | 0.449 | 0.507 | **0.294** | 0.412 | 0.471 | 0.740 | 0.141 | **0.155** | 0.170 | 0.209 | 0.396 | 0.404 | 0.420 | 0.457 |
| iTransformer(2024a) | 0.389 | 0.424 | 0.456 | 0.545 | 0.305 | 0.405 | 0.411 | 0.448 | **0.135** | **0.155** | **0.169** | **0.204** | **0.374** | **0.393** | **0.409** | **0.450** |
| FITS~\cite{xufits} | 0.380 | 0.415 | **0.439** | **0.433** | **0.271** | **0.332** | **0.355** | **0.378** | 0.147 | 0.159 | **0.169** | 0.214 | 0.402 | 0.419 | 0.423 | 0.459 |
| TimeLLM(2024) | 0.390 | 0.427 | 0.459 | **0.452** | 0.300 | 0.365 | 0.367 | 0.411 | **0.135** | 0.156 | **0.160** | **0.197** | **0.377** | **0.385** | **0.399** | **0.436** |
| SparseTSF(2024) | **0.362** | **0.404** | **0.435** | **0.426** | 0.294 | **0.340** | **0.360** | **0.383** | **0.139** | **0.155** | **0.167** | **0.208** | **0.389** | **0.399** | **0.417** | **0.449** |
| TimeBase(ours) | **0.349** | **0.387** | **0.408** | **0.439** | **0.292** | **0.341** | **0.358** | **0.400** | **0.139** | **0.153** | **0.169** | **0.208** | 0.394 | 0.403 | **0.417** | 0.456 |

Table 2: The MSE results for long-term time series forecasting compare TimeBase with other baselines on datasets with a period much larger than the input, indicating that there is no periodicity in the input. The top four results are highlighted in bold.

| Dataset | ETTm1 | | | | ETTm2 | | | | Weather | | | |
|---|---|---|---|---|---|---|---|---|---|---|---|---|
| Horizon | 96 | 192 | 336 | 720 | 96 | 192 | 336 | 720 | 96 | 192 | 336 | 720 |
| Informer (2021) | 0.632 | 1.131 | 1.391 | 1.397 | 1.870 | 2.807 | 4.442 | 5.258 | 0.283 | 0.445 | 0.587 | 0.953 |
| Autoformer (2021) | 0.455 | 0.562 | 0.737 | 0.503 | 0.325 | 0.369 | 0.418 | 0.612 | 0.323 | 0.389 | 0.497 | 0.573 |
| FEDformer (2022) | 0.406 | 0.450 | 0.436 | 0.462 | 0.339 | 0.397 | 0.449 | 0.451 | 0.289 | 0.340 | 0.370 | 0.420 |
| TimesNet (2023) | 0.359 | 0.368 | 0.429 | 0.477 | 0.200 | 0.274 | 0.340 | 0.384 | 0.176 | 0.219 | 0.277 | 0.344 |
| PatchTST (2023) | **0.298** | **0.335** | **0.366** | 0.420 | **0.165** | **0.219** | **0.268** | **0.352** | **0.149** | **0.193** | **0.240** | **0.312** |
| DLinear (2023) | **0.307** | 0.347 | **0.367** | **0.415** | **0.163** | 0.223 | 0.291 | 0.407 | 0.176 | 0.216 | 0.262 | 0.326 |
| iTransformer (2024a) | 0.315 | 0.349 | 0.381 | 0.437 | 0.179 | 0.239 | 0.309 | 0.387 | **0.159** | **0.203** | **0.253** | **0.317** |
| FITS (2024) | **0.313** | **0.339** | **0.367** | **0.417** | **0.166** | **0.218** | **0.271** | **0.352** | 0.176 | 0.217 | 0.261 | 0.325 |
| TimeLLM (2024) | 0.316 | **0.338** | 0.368 | 0.430 | 0.183 | 0.241 | 0.292 | 0.362 | **0.155** | **0.191** | **0.246** | **0.313** |
| SparseTSF (2024) | 0.314 | 0.348 | 0.368 | **0.419** | 0.167 | **0.219** | **0.271** | **0.353** | 0.174 | 0.216 | **0.260** | 0.325 |
| TimeBase(ours) | **0.310** | **0.338** | **0.364** | **0.413** | **0.166** | **0.218** | **0.270** | **0.352** | 0.174 | 0.215 | **0.260** | **0.323** |

In our experiments, we evaluated the Mean Squared Error (MSE) performance of TimeBase against several state-of-the-art baselines on both short and long-term datasets. As indicated in Tables 1 and 2, TimeBase consistently outperform other models across different forecasting horizons. Even with an extremely small number of parameters, TimeBase remains competitive with other models that have significantly larger parameter counts, achieving top-four rankings nearly throughout all settings. Notably, on short-period datasets, TimeBase excels on the ETTh1 dataset, and even demonstrating an average 6% improvement in term of MSE, significantly outperforming some large-scale benchmark models. The full forecasting results are provided in Appendix N.

Table 3: Efficiency comparison of TimeBase and other state-of-the-art models on the Electricity dataset with a forecasting length of 720. To ensure fair comparison, the look-back window is set as 720 for all models.

| Model | Parameters | MACs | Max Mem.(MB) | Epoch Time(s) | Infer Time (CPU) |
|---|---|---|---|---|---|
| Informer(2021) | 22.45M | 7.85G | 1424.99 | 143.05 | 72.67ms |
| Autoformer(2021) | 22.14M | 8.97G | 4348.89 | 225.78 | 126.75ms |
| FEDformer(2022) | 22.14M | 10.48G | 2361.76 | 558.03 | 203.31ms |
| PatchTST (2023) | 8.69M | 14.17G | 18034.33 | 827.34 | 249.02ms |
| DLinear(2023) | 1.04M | 333.04M | 158.21 | 41.08 | 3.25ms |
| FITS (2024) | 10.5K | 79.9M | 496.7 | 35.00 | 2.85ms |
| iTransformer(2024a) | 5.47M | 1.79G | 828.32 | 65.62 | 30.41ms |
| SparseTSF(2024) | 1.0K | 12.71M | 125.2 | 31.30 | 2.59ms |
| TimeBase(ours) | **0.39K** | **2.77M** | **88.89** | **20.6** | **0.98ms** |
| *Reduction(%)* | *61.0%* | *78.2%* | *29.0%* | *34.2%* | *62.2%* |

## 5.3 EFFICIENCY ANALYSIS

**Main Efficiency Comparision**   In addition to its impressive predictive performance, TimeBase offers another major advantage: its exceptionally lightweight design. Here, we provide a more comprehensive comparison, examining both static and runtime metrics, which include **(1) Parameters:** *The total number of trainable parameters, reflecting the model's size.* **(2) MACs (Multiply-Accumulate Operations):** *A standard measure of computational complexity in neural networks, representing the number of multiply-accumulate operations required by the model.* **(3) Max Memory:** *The peak memory usage during training.* **(4) Epoch Time:** *The time required to train the model for one epoch, averaged over three runs.* **(5) Infer Time:** *Infer Time indicates the average inference time per sample on CPU.*

The look-back window for each model are set as 720 for all models, and the max memory is recorded with a constant batch size of 12. The FFT operation, which takes only 0.08 seconds (for electricity) to determine period length and is not utilized during training and inferring, is negligible due to its minimal computational overhead. Moreover, most real-world time series data typically come with sufficient prior information on periodicity. The efficiency analysis presented in Table 3 highlights the remarkable advantages of TimeBase over other state-of-the-art models in terms of both static and runtime metrics. TimeBase achieves a substantial reduction in the number of parameters and computational complexity (MACs) compared to more parameter-heavy models like Informer and FEDformer. Specifically, TimeBase reduces the parameter count by up to 61% and the MACs by over 78%, while also using significantly less memory (29% reduction) and training faster (34% reduction in epoch time). These results demonstrate that TimeBase not only maintains strong predictive performance but also offers superior efficiency, making it well-suited for resource-constrained environments.

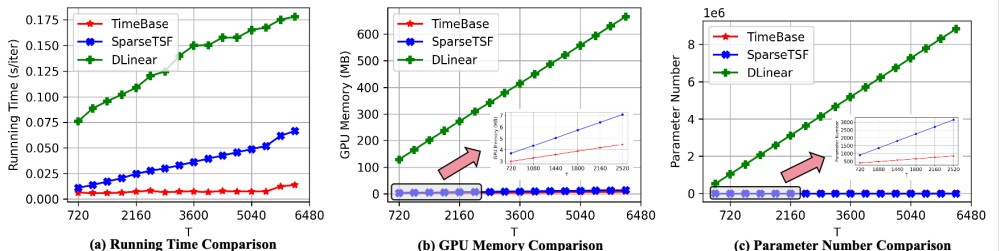

Figure 4: Comparison of efficiency metrics between TimeBase and other lightweight models with varying look-back windows. (a) Running time per iteration (s/iter), (b) GPU memory consumption, and (c) Parameter count as the look-back window increases from 720 to 6480.

**Efficiency in Ultra-long Look-back Window**   Additionally, we evaluate the efficiency of Time-Base under ultra-long look-back windows, comparing it with lightweight models (i.e., SparseTSF and DLinear), the current most lightweight model. The comparison focuses on three key metrics:

running time per iteration, GPU memory usage, and parameter count, as shown in Figure 4. As the look-back window increases from 720 to 6480, with a fixed batch size of 12 and prediction length of 720, TimeBase consistently demonstrates its lightweight nature. Even with a ninefold increase in input sequence length, TimeBase's running time only increases by 0.05 seconds, GPU memory usage expands by a factor of 3.8, and the number of parameters grows by only 3.1 times. These results highlight the model's extreme efficiency and scalability in handling ultra-long sequences.

## 5.4 Hyperparameter Analysis

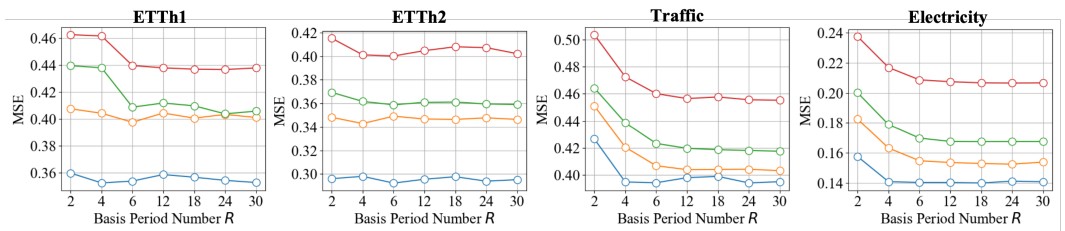

Figure 5: Effect of basis number $R$ on MSE across Traffic, Electricity, ETTh1, and ETTh2.

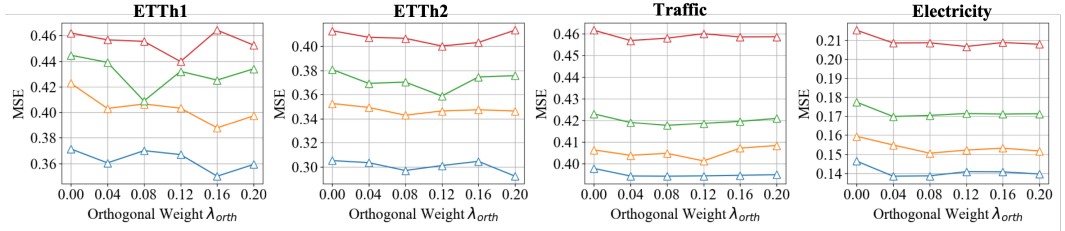

Figure 6: Effect of orthogonal loss weight $\lambda_{orth}$ across Traffic, Electricity, ETTh1, and ETTh2.

This section explores the impact of two key hyperparameters on the performance of TimeBase: the number of basis functions $R$ and the orthogonal loss weight $\lambda_{orth}$. In Figure 5, the number of basis number $R$ is varied from [2, 4, 6, 12, 18, 24, 30], and the corresponding MSE results for Traffic, Electricity, ETTh1, and ETTh2 datasets are reported. The results indicate that, in most cases, increasing $R$ has minimal impact on the model's performance. This suggests that $R = 6$ is sufficient to capture the essential basis information without degrading prediction accuracy. TimeBase is able to maintain strong performance with a small number of basis functions, demonstrating the model's efficiency in extracting representative components. Figure 6 shows the MSE results as the orthogonal loss weight $\lambda_{orth}$ is varied across [0, 0.04, 0.08, 0.12, 0.16, 0.20]. For datasets such as ETTh1, and ETTh2, prediction performance fluctuates with different values of $\lambda_{orth}$. However, for Traffic and Electricity, the performance remains relatively stable. Overall, increasing $\lambda_{orth}$ tends to improve performance to some extent, though its impact is dataset-dependent. These findings suggest that careful tuning of $\lambda_{orth}$ can lead to performance gains, particularly for certain datasets.

## 5.5 Ablation Study

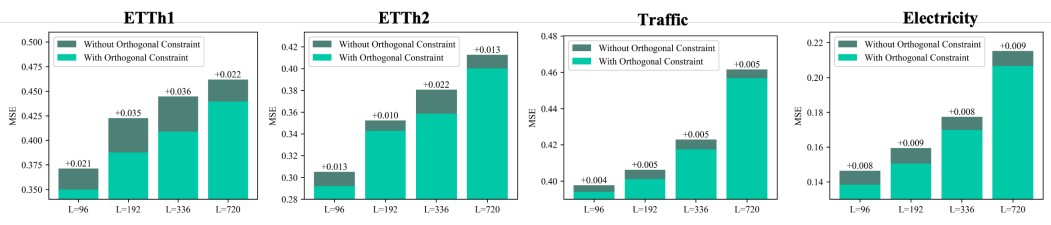

Figure 7: MSE comparison with and without orthogonal constraint across different prediction lengths for Traffic, Electricity, ETTh1, and ETTh2 datasets.

Figure 7 illustrates the MSE results for various prediction lengths, both with and without the orthogonal constraint. The inclusion of the orthogonal constraint consistently leads to improvements across all datasets, with gains up to 0.036 in MSE. This indicates that the orthogonal constraint helps the model learn more representative basis periods, enhancing both the model's representational capacity and predictive performance. The positive impact across multiple datasets and forecasting horizons demonstrates the value of incorporating the orthogonal constraint into the training process.

## 5.6 FORECASTING WITH LOW-QUALITY DATA

Table 4: Few-shot Forecasting

| Methods | TimeBase | | TimeLLM | | iTransformer | |
|---|---|---|---|---|---|---|
| Metric | MSE | MAE | MSE | MAE | MSE | MAE |
| ETTh1 | **0.571** | **0.524** | 0.572 | 0.531 | 0.591 | 0.523 |
| ETTh2 | **0.392** | **0.419** | 0.401 | 0.412 | 0.396 | 0.422 |
| ETTm1 | **0.403** | **0.423** | 0.409 | 0.433 | 0.461 | 0.439 |
| ETTm2 | **0.284** | **0.322** | 0.289 | 0.336 | 0.289 | 0.336 |

Table 5: Transfer Learning

| Methods | | TimeBase | | TimeLLM | | iTransformer | |
|---|---|---|---|---|---|---|---|
| Source | Target | MSE | MAE | MSE | MAE | MSE | MAE |
| ETTh1 | ETTh2 | **0.313** | **0.357** | 0.359 | 0.390 | 0.406 | 0.422 |
| ETTh2 | ETTh1 | **0.436** | **0.438** | 0.478 | 0.471 | 0.757 | 0.578 |
| ETTm1 | ETTm2 | **0.254** | **0.316** | 0.272 | 0.333 | 0.313 | 0.348 |
| ETTm2 | ETTm1 | 0.449 | **0.433** | **0.422** | 0.438 | 0.663 | 0.563 |

This section examines forecasting performance under low-quality data conditions, focusing on the performance of few-shot forecasting (training with 10% data) and transfer learning tasks (training on source data and testing on target data). Average forecasting performance among $L = [96, 196, 336, 720]$ in Table 4 and Table 5 shows that TimeBase outperforms SOTA models in both scenarios. Notably, it shows that basis period extraction technique of TimeBase really does well in data-scarce and transfer learning settings, demonstrating its robustness and adaptability when dealing with low-quality data.

## 5.7 BASIS VISUALIZATION

Figure 8 displays the basis periods extracted by TimeBase from the Electricity dataset, along with the Pearson correlation matrix between them. The results show that the Pearson correlation coefficients between most basis periods are close to zero, indicating low correlation among them. This suggests that TimeBase is capable of extracting distinct and representative basis periods from the approximate low-rank structure of long-term time series data. By identifying these representative basis periods, TimeBase can perform period-level forecasting using a compact set of period features.

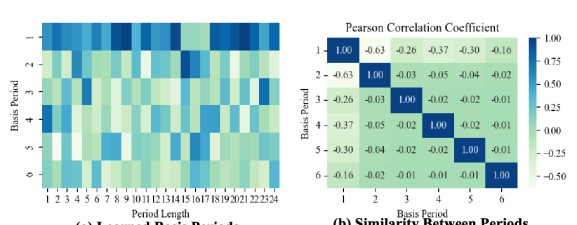

Figure 8: Visualization of the learned basis periods on the Electricity dataset and the corresponding Pearson correlation coefficients.

This approach significantly reduces the number of parameters required by the model while maintaining competitive forecasting performance. The ability to leverage such compact representations is key to TimeBase's parameter efficiency and contributes to its strong performance on long-term time series forecasting tasks.

## 6 CONCLUSION

To deploy long-term time series forecasting (LTSF) models in more realistic scenarios such as edge computing and mobile devices, we focus on exploring lightweight forecasting methods. Given the periodicity of long time series and the approximate low-rank nature resulting from similar patterns between adjacent cycles, we propose TimeBase. This model employs basis period extraction to identify representative periodic features, transforming step-level long-term time series forecasting into period-level forecasting. Theoretical analysis and extensive empirical results demonstrate that this approach not only ensures prediction accuracy but also significantly reduces the model size, achieving the lightest time series forecasting model to date, with only 0.39K parameters.

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

## A  DIFFERENCES BETWEEN TIMEBASE AND OTHER MLP-BASED MODELS

Table 6: Differences between TimeBase and other MLP-based Models

| Linear-based Model | TimeBase(Ours) | TimeMixer | Koopa | DLinear | MTS-Mixer | TSMixer | HDMixer | TiDE |
|---|---|---|---|---|---|---|---|---|
| Scale | Extremely Light (0.39 K) | Light (5.57M) | Normal (30.04 M) | Light (1.03M) | Light (2.02M) | Light (1.05M) | Light (4.81M) | Normal (31.07M) |
| Performance | Perfect (0.208) | Perfect (0.206) | Perfect (0.215) | Perfect (0.209) | Good (0.213) | Good (0.236) | Good (0.243) | Good (0.241) |
| Forecasting Type | Segment-level | Point-level | Point-level | Point-level | Point-level | Point-level | Point-level | Point-level |

We summarize the differences between TimeBase and other MLP/Mixer-based models in the Table 6, highlighting key differences across three crucial aspects: model scale, performance, and forecasting type. These differences underscore the efficiency and unique strengths of TimeBase in long-term time series forecasting. In terms of scale, TimeBase stands out with an extremely lightweight architecture that requires only 0.39K parameters for 720-horizon forecasting on the Electricity dataset. This is a striking contrast to other MLP-based models, whose parameter sizes range from 1.03M (for DLinear) to 31.07M (for TiDE). This significant reduction in model size—by a factor of over 1000—allows TimeBase to operate with minimal computational overhead, making it a highly efficient choice for real-time and resource-constrained applications. The compact size of TimeBase ensures that it is more suitable for deployment in environments with strict memory and computation limitations, without sacrificing performance. Regarding forecasting performance, TimeBase maintains competitive accuracy, achieving an MSE of 0.208 for 720-horizon forecasting, which is comparable to or better than most other models in the comparison. For instance,

TimeMixer, Koopa, and DLinear all have slightly higher MSE values (ranging from 0.206 to 0.215), indicating that TimeBase performs just as well, if not slightly better, in terms of predictive accuracy. While some models, such as MTS-Mixer and TSMixer, exhibit slightly higher performance, the differences are relatively small, and the trade-off for using such models—larger parameter sizes and greater computational costs—becomes more apparent when considering the overall efficiency of TimeBase. In terms of forecasting type, TimeBase adopts a segment-level approach, which distinguishes it from most other models that employ a point-level forecasting strategy. This segment-level forecasting method allows TimeBase to better capture the temporal dependencies in long-term time series data by focusing on the relationships between segments of data, rather than predicting individual time steps. This can lead to improved generalization and robustness in cases where the time series exhibits periodic or seasonal behavior, as is often the case in many real-world forecasting tasks.

## B DERIVATION OF PARAMETERS AND EFFECTIVENESS

### B.1 DERIVATION OF PARAMETER SCALE IN TIMEBASE

To determine the total number of parameters in TimeBase, we need to account for two components, i.e., $\text{BasisExtract}(\cdot)$ and $\text{PeriodForecast}(\cdot)$. $\text{BasisExtract}(\cdot)$ is implemented as a linear layer where the input size is $\lceil \frac{T}{P} \rceil$ and the output size is $R$. The number of parameters for this layer is $(\lceil \frac{T}{P} \rceil + 1) \times R$ wheres $\lceil \frac{T}{P} \rceil + 1$ accounts for the input features plus the bias term, multiplied by the output size $R$. $\text{PeriodForecast}(\cdot)$ is another linear layer where the input size is $R$ and the output size is $\lceil \frac{L}{P} \rceil$. The number of parameters for this layer is given by $(R + 1) \times \lceil \frac{L}{P} \rceil$, where $R + 1$ accounts for the input features plus the bias term, multiplied by the output size $\lceil \frac{L}{P} \rceil$.

Adding these two components together, the total parameter scale of TimeBase is given by:

$$\text{Number} = (\lceil \frac{T}{P} \rceil + 1) \times R + (R + 1) \times \lceil \frac{L}{P} \rceil = \lceil \frac{T + L + P}{P} \rceil \times R + \lceil \frac{L}{P} \rceil \tag{14}$$

### B.2 DERIVATION OF EFFECTIVENESS ON PERIOD ORIENTED FORECASTING

Each periodic time series $X_n \in \mathbb{R}^P$ can be decomposed into the basis periods $\{e_1, e_2, \ldots, e_R\}$ with an added residual term $\epsilon_n$. For the entire matrix representation $\mathbf{X}_{\text{his}}$, this holds for each row, where each row corresponds to a time period of length $P$. Thus, we can represent each row of $\mathbf{X}_{\text{his}}$ as a linear combination of the basis signals in $E$, weighted by matrix $W_E \in \mathbb{R}^{R \times P}$, along with the residual term $B$. Correspondingly, we have the relationship:

$$\mathbf{X}_{\text{his}}^T = E^T \times W_E + B \tag{15}$$

Rearranging this equation to solve for $E^T$, we obtain:

$$E^T \approx \mathbf{X}_{\text{basis}} = (\mathbf{X}_{\text{his}}^T - B) \times W_E^\dagger \tag{16}$$

where $W_E^\dagger$ represents the Moore-Penrose pseudoinverse of $W_E$. Thus, the learnable parameters are given by:

$$W_{\text{his}} = W_E^\dagger \quad \text{and} \quad B_{\text{his}} = -B \times W_E^\dagger \tag{17}$$

Thus, we can predict future time series by applying weights $W_{\text{pred}} \in \mathbb{R}^{P \times P}$ to combine the basis periods $E^T$ appropriately. Additionally, a bias term $B_{\text{pred}} \in \mathbb{R}^P$ is added to account for any remaining variability. Thus, the prediction equation is:

$$\mathbf{X}_{\text{pred}}^T = E^T \times W_{\text{pred}}^T + B_{\text{pred}} \tag{18}$$

Substituting the expression for $E^T$ of Eq( 16) and Eq(18) , we get:

$$\begin{aligned}
\mathbf{X}_{\text{pred}}^T &\approx \mathbf{X}_{\text{basis}} \times W_{\text{pred}}^T + B_{\text{pred}} \\
&= (\mathbf{X}_{\text{his}}^T \times W_{\text{his}} + B_{\text{his}}) \times W_{\text{pred}}^T + B_{\text{pred}}
\end{aligned} \tag{19}$$

### B.3 DERIVATION OF TIMEBASE'S GENERALIZATION

In this section, we analyze the generalization capability of TimeBase. Firstly, we demonstrate that effective basis vectors can still be extracted from non-periodic data. Subsequently, we prove that the generalization error of TimeBase is closely related to the quality of the extracted basis vectors.

**Effectiveness on Non-periodic Data**  Assume we have a historical time series matrix $\mathbf{X}_{\text{his}} \in \mathbb{R}^{N \times P}$, where $N$ represents the number of time segments (i.e., the time series is divided into $N$ segments) and $P$ denotes the length of each time segment, not equal to the period length. According to the definition of SVD, any matrix $\mathbf{X} \in \mathbb{R}^{N \times P}$ can be decomposed as follows:

$$\mathbf{X}_{\text{his}} = U\Sigma V^T, \tag{20}$$

where $U \in \mathbb{R}^{N \times N}$ is the left singular matrix containing the left singular vectors of $\mathbf{X}_{\text{his}}$, $\Sigma \in \mathbb{R}^{N \times P}$ is a diagonal matrix containing the singular values, which describe the "importance" of the matrix and $V^T \in \mathbb{R}^{P \times P}$ is the right singular matrix containing the right singular vectors of $\mathbf{X}_{\text{his}}$. The decomposition provides a structured representation of $\mathbf{X}_{\text{his}}$ in terms of orthogonal bases. Specifically, in the context of TimeBase, the right singular vectors in $V^T$ can be interpreted as candidate basis vectors $\mathbf{E} \in \mathbb{R}^{R \times P}$, where $R \leq \min(N, P)$ specifies the number of significant components to be retained. This dimensionality reduction highlights the inherent structure within the data and facilitates generalization, even in cases where $P$ does not deviate from the period length. Next, to reconstruct the time series data $\mathbf{X}_{\text{his}}$ using these basis vectors, it can be achieved by a linear layer in a deep learning framework, which applies the following transformation: $\mathbf{X}_{\text{his}}^T \approx \mathbf{E}^T W_{\text{E}} + B$. Based on this, we can get $E^T \approx (\mathbf{X}_{\text{his}}^T - B) \times W_E^\dagger$ as stated in Eq( 16).

**The Upper Bound of TimeBase's Error**  The prediction error is defined as:

$$\mathbf{r} = \mathbf{X}_{\text{pred}}^T - \mathbf{E}^T W_{\text{pred}} - B_{\text{pred}} \tag{21}$$

By ignoring the bias term $B_{\text{pred}}$, the norm of the error is:

$$\|\mathbf{r}\|_2 = \|\mathbf{X}_{\text{pred}} - W_{\text{pred}}\mathbf{E}\|_2 \tag{22}$$

To derive the optimal coefficient matrix $W_{\text{pred}}$, we need to solve the following optimization problem:

$$\begin{aligned}
&\min_{W_{\text{pred}}} \|\mathbf{X}_{\text{pred}} - W_{\text{pred}}\mathbf{E}\|_2^2 \\
&= \min_{W_{\text{pred}}} \text{Tr}\left((\mathbf{X}_{\text{pred}} - W_{\text{pred}}\mathbf{E})(\mathbf{X}_{\text{pred}} - W_{\text{pred}}\mathbf{E})^T\right) \\
&= \min_{W_{\text{pred}}} \text{Tr}(\mathbf{X}_{\text{pred}}\mathbf{X}_{\text{pred}}^T) - 2\,\text{Tr}(W_{\text{pred}}\mathbf{E}\mathbf{X}_{\text{pred}}^T) + \text{Tr}(W_{\text{pred}}\mathbf{E}\mathbf{E}^T W_{\text{pred}}^T)
\end{aligned} \tag{23}$$

Next, we take the derivative:

$$\begin{aligned}
&\nabla_{W_{\text{pred}}} \|\mathbf{X}_{\text{pred}} - W_{\text{pred}}\mathbf{E}\|_2^2 \\
&= \nabla_{W_{\text{pred}}} \text{Tr}(\mathbf{X}_{\text{pred}}\mathbf{X}_{\text{pred}}^T) - 2\,\text{Tr}(W_{\text{pred}}\mathbf{E}\mathbf{X}_{\text{pred}}^T) + \text{Tr}(W_{\text{pred}}\mathbf{E}\mathbf{E}^T W_{\text{pred}}^T) \\
&= -2\mathbf{X}_{\text{pred}}\mathbf{E}^T + 2W_{\text{pred}}\mathbf{E}\mathbf{E}^T
\end{aligned} \tag{24}$$

Setting the derivative equal to zero, we get:

$$W_{\text{pred}} = \mathbf{X}_{\text{pred}}\mathbf{E}^T(\mathbf{E}\mathbf{E}^T)^{-1} \tag{25}$$

Substituting the optimal coefficient $W_{\text{pred}}$ into the error expression, we obtain the error:

$$\begin{aligned}
\mathbf{r} &= \mathbf{X}_{\text{pred}} - \mathbf{X}_{\text{pred}}\mathbf{E}^T(\mathbf{E}\mathbf{E}^T)^{-1}\mathbf{E} \\
&= \mathbf{X}_{\text{pred}}(\mathbf{I} - \mathbf{E}^T(\mathbf{E}\mathbf{E}^T)^{-1}\mathbf{E})
\end{aligned} \tag{26}$$

Here, $\mathbf{P} = \mathbf{I} - \mathbf{E}^T(\mathbf{E}\mathbf{E}^T)^{-1}\mathbf{E}$ is a projection matrix. The norm property of the projection matrix can be bounded using the matrix spectral norm:

$$\mathbf{P} = \|\mathbf{I} - \mathbf{E}^T(\mathbf{E}\mathbf{E}^T)^{-1}\mathbf{E}\|_2 = 1 - \sigma_{\min}, \tag{27}$$

where $\sigma_{\min}$ represents the smallest singular value of $\mathbf{E}\mathbf{E}^T$. Since singular values are equivalent to eigenvalues in this context, we have:

$$\sigma_{\min} = \lambda_{\min}(\mathbf{E}\mathbf{E}^T) \tag{28}$$

Next, based on the fact that the spectral norm of the projection matrix is bounded above by $\frac{1}{\lambda_{\min}(\mathbf{E}\mathbf{E}^T)}$ Golub & Van Loan (2013), and utilizing the inequality property of matrix norms, $\|\mathbf{A}\mathbf{B}\|_2 \leq \|\mathbf{A}\|_2\|\mathbf{B}\|_2$ Horn & Johnson (2012), we obtain:

$$\|\mathbf{r}\|_2 \leq \frac{1}{\lambda_{\min}(\mathbf{E}\mathbf{E}^T)}\|\mathbf{X}_{\text{pred}}\|_2 \tag{29}$$

where $\lambda_{\min}(\mathbf{E}\mathbf{E}^T)$ denotes the smallest eigenvalue of the Gram matrix $\mathbf{E}\mathbf{E}^T$. This result highlights that the generalization capability of TimeBase to arbitrary time series relies on learning a well-represented basis matrix $\mathbf{E}$. If $\mathbf{E}$ exhibits a favorable eigenvalue distribution (i.e., $\lambda_{\min}(\mathbf{E}\mathbf{E}^T)$ is large), the upper bound on prediction error is lower, highlighting the importance of a high-quality basis vector space and the necessity of orthogonal constraint.

## C   PERIOD MEAN NORMALIZATION

We perform period normalization on the time series data, ensuring that each value within the same period is centered by subtracting the mean of all values at the same time index across different periods. Later, during prediction, the corresponding mean will be added back to the predicted values to restore the original scale. Given a time series matrix $\mathbf{X}_{\text{his}} \in \mathbb{R}^{N \times P}$, where $N$ is the number of periods and $P$ is the length of each period, we compute the mean $\mathbf{X}_{\text{mean}} \in \mathbb{R}^P$ across all periods at each time index:

$$\mathbf{X}_{\text{mean}}(j) = \frac{1}{N}\sum_{i=1}^{N} \mathbf{X}_{\text{his}}(i,j) \quad \text{for} \quad j = 1, 2, \ldots, P \tag{30}$$

Here, $\mathbf{X}_{\text{mean}}(j)$ represents the mean of the data at time index $j$ across all periods. We then subtract this mean from each element of the matrix $\mathbf{X}_{\text{his}}$ to obtain the normalized matrix $\mathbf{X}_{\text{norm}} \in \mathbb{R}^{N \times P}$:

$$\mathbf{X}_{\text{his}}(i,j) = \mathbf{X}_{\text{his}}(i,j) - \mathbf{X}_{\text{mean}}(j) \tag{31}$$

During the prediction phase, when generating the future period matrix $\mathbf{X}_{\text{pred}} \in \mathbb{R}^{N' \times P}$, where $N'$ is the number of future periods, we add the corresponding mean back to restore the original scale:

$$\mathbf{X}_{\text{pred}}(i,j) = \mathbf{X}_{\text{pred}}(i,j) + \mathbf{X}_{\text{mean}}(j) \quad \text{for} \quad i = 1, 2, \ldots, N' \quad \text{and} \quad j = 1, 2, \ldots, P \tag{32}$$

This ensures that the predicted values $\mathbf{X}_{\text{pred}}$ maintain the same overall trends as the original time series data after normalization.

## D   MORE DESCRIPTION OF DATASETS

We evaluate performance of long-term forecasting on **Weather**, **Traffic**, **Electricity** and four **ETT** datasets (i.e., ETTh1, ETTh2, ETTm1, and ETTm2), which have been extensively adopted for benchmarking long-term forecasting models. Adhering to the established protocol in Wu et al. (2021), we partition the datasets into training, validation, and test sets with a ratio of 6:2:2 for the last four ETT datasets and 7:1:2 for the remaining datasets. The input length, prediction length, and the variable number of each real-world dataset are presented in Table 7. The detailed dataset descriptions are as follows: **1) Weather** includes 21 indicators of weather, such as air temperature, and humidity. Its data is recorded every 10 min for 2020 in Germany. **2) Traffic** describes hourly road occupancy rates measured by 862 sensors on San Francisco Bay area freeways from 2015 to 2016. **3) Electricity** contains hourly electricity consumption (in Kwh) of 321 clients from 2012 to 2014. **4) ETT** consists of two hourly-level datasets (ETTh) and two 15minute-level datasets (ETTm). Each of them contains seven oil and load features of electricity transformers from July 2016 to July 2018.

Table 7: Dataset Statistics

| Dataset | Variate | Input Length | Predict Length | Forecastability* | Information | Frequency |
|---------|---------|--------------|----------------|------------------|-------------|-----------|
| ETTh1 | 7 | 720 | 96~720 | 0.46 | Electricity | Hourly |
| ETTh2 | 7 | 720 | 96~720 | 0.46 | Electricity | Hourly |
| ETTm1 | 7 | 720 | 96~720 | 0.46 | Electricity | 15mins |
| ETTm2 | 7 | 720 | 96~720 | 0.46 | Electricity | 15mins |
| Weather | 21 | 720 | 96~720 | 0.75 | Weather | 10mins |
| Electricity | 321 | 720 | 96~720 | 0.77 | Electricity | Hourly |
| Traffic | 862 | 720 | 96~720 | 0.68 | Transportation | Hourly |

*The forecastability is calculated by one minus the entropy of Fourier decomposition of time series. A larger value indicates better predictability.

# E MORE DETAILS OF BASELINES

We compare TimeBase with 10 baselines, which comprise the state-of-the-art long-term forecasting models: iTransformer Liu et al. (2024a), PatchTST Nie et al. (2023), DLinear Zeng et al. (2023), TimesNet Wu et al. (2023), FEDformer Zhou et al. (2022), Autoformer Wu et al. (2021), and Informer Zhou et al. (2021), relatively efficient models: FITS Xu et al. (2024), SparseTSF Lin et al. (2024), as well as LLM-based methods: TimeLLM Jin et al. (2024). We briefly describe the selected 10 state-of-the-art baselines as follows: **1) iTransformer** Liu et al. (2024a) simply inverts the duties of the attention mechanism and the feed-forward network to encode each individual series into *variate tokens* and for forecasting. **2) PatchTST** Nie et al. (2023) is a strong versatile transformer baseline using channel-independence and patching. **3) TimesNet** Wu et al. (2023) is a task-general foundational model for time series, reshaping 1-dim temporal data to 2-dim space and using 2-dim backbone to deal with the data. **4) FEDformer** Zhou et al. (2022) introduces a frequency-enhanced decomposer to model seasonal-trend time series in an efficient manner. **5) Autoformer** Wu et al. (2021) employs an auto-correlation mechanism and series decomposition block to improve long-sequence forecasting. **6) Informer** Zhou et al. (2021) utilizes a sparse self-attention mechanism and a distilling operation to handle long time series more efficiently. **7) DLinear** Zeng et al. (2023) is a simple linear-based model combined with a decomposition module. **8) FITS** Xu et al. (2024) introduces an innovative method for time series forecasting using a complex-valued neural network, effectively capturing both the magnitude and phase of the data. This dual representation enables a more thorough and efficient analysis of time series signals. **9) SparseTSF** Lin et al. (2024) simplifies the time series forecasting process by downsampling the original sequences at fixed intervals. Each downsampled segment is used to predict cross-period trends, reducing the complexity of the original forecasting task. **10) TimeLLM** Jin et al. (2024) adapts large language models for time series prediction through a reprogramming strategy, keeping the model architecture unchanged while optimizing for temporal forecasting.

# F MORE IMPLEMENTATION DETAILS

We build TimeBase using PyTorch 1.13.0 Paszke et al. (2019). All experiments are conducted on a single NVIDIA A100 GPU with 40GB of memory. The model is trained with the Adam optimizer Kingma (2014) with L2 loss over 30 epochs. After the first three epochs, a learning rate decay of 0.8 is applied, and early stopping is employed with a patience threshold of five epochs. In TimeBase, Channel Independence (CI) Nie et al. (2023) is involved to simplify the multivariate forecasting process to univariate time series forecasting. Due to its highly simplistic design, TimeBase requires minimal hyperparameter tuning. The period $P$ is set to the natural period of the dataset (e.g., $P = 24$ for ETTh1), or a smaller value is chosen when dealing with datasets that exhibit extremely long periods (e.g., $P = 4$ for Weather). We performed a grid search for TimeBase to find the optimal hyperparameters, specifically for the regularization parameter $\lambda_{orth} = [0.04, 0.08, 0.12, 0.16, 0.20]$ to accommodate varying loss scales between datasets, as well as the learning rate between 0.01 and 0.5. The loss function is MSE.

To enhance the reliability of our results, we re-run baselines in an uniform and fair setting. For SparseTSF, PatchTST, DLinear, FITS, Time-LLM, Fedformer, and TimesNet, we utilized their official code repositories. For Autoformer and Informer, we leveraged the code provided in the official DLinear repository to run these models. To ensure a fair comparison with other efficient LTSF baselines such as Xu et al. (2024); Lin et al. (2024), which utilize a uniform input length of 720, we also adopt an input length of 720 for all models. It is important to note that we have corrected the bug involving `test_loader` where `drop_last=True` during testing on the test set, ensuring that `drop_last=False` is used instead. All scripts and logs for running our model and baselines are available at `https://anonymous.4open.science/r/TimeBase-fixbug`.

## G PERFORMANCE ON FURTHER PREDICTION LENGTH

To better illustrate its strengths in long-term time series forecasting, we extended the maximum prediction horizon beyond 720 to include 1080, 1440, and 1800 time steps. We compared its performance against well-established LTSF models, such as iTransformer and DLinear, across multiple datasets (ETTh1, ETTh2, and Electricity). As shown in Table 8, 9, 10, the results underscore that TimeBase consistently outperforms these models in ultra-long-term forecasting tasks. It achieves this while maintaining linear growth in model complexity (measured by parameters, MACs), demonstrating scalability as the prediction length increases. Specifically, TimeBase not only yields lower Mean Squared Error (MSE) values but also achieves these results with significantly fewer parameters and MACs compared to its counterparts. For example, in the ETTh1 dataset at a prediction length of 1800, TimeBase achieves an MSE of 0.714 with only 0.7K parameters and 0.1M MACs. In contrast, iTransformer and DLinear exhibit higher MSEs of 0.812 and 0.796, respectively, while using 523.9K and 2.59M parameters, and 6.78M and 18.15M MACs. Similar trends are observed across the ETTh2 and Electricity datasets, where TimeBase demonstrates robust accuracy and efficiency advantages. These findings validate TimeBase's effectiveness in ultra-long-term forecasting tasks, particularly when resource efficiency is critical. Moreover, the linear growth in computational cost ensures its feasibility for deployment on edge devices. This positions TimeBase as a practical solution for real-world long-term forecasting scenarios.

Table 8: Further prediction length on Electricity. The input length is set as 720 for all models.

| | Electricity ‖ 1080 | | | Electricity ‖ 1440 | | | Electricity ‖ 1800 | | |
|---|---|---|---|---|---|---|---|---|---|
| | MSE | Param | MAC | MSE | Param | MAC | MSE | Param | MAC |
| **TimeBase** | **0.234** | **0.5 K** | **3.47 M** | **0.264** | **0.6 K** | **4.16 M** | **0.295** | **0.7 K** | **4.85 M** |
| iTransformer | 0.253 | 5.65 M | 1.85 G | 0.272 | 5.84 M | 1.91 G | 0.325 | 6.03 M | 1.97 G |
| DLinear | 0.255 | 1.6 M | 499.45 M | 0.290 | 2.1 M | 665.86 M | 0.321 | 2.59 M | 832.26 M |

Table 9: Further prediction length on ETTh2. The input length is set as 720 for all models.

| | ETTh2 ‖ 1080 | | | ETTh2 ‖ 1440 | | | ETTh2 ‖ 1800 | | |
|---|---|---|---|---|---|---|---|---|---|
| | MSE | Param | MAC | MSE | Param | MAC | MSE | Param | MAC |
| **TimeBase** | **0.478** | **0.5 K** | **0.07 M** | **0.543** | **0.6 K** | **0.09 M** | **0.552** | **0.7 K** | **0.1 M** |
| iTransformer | 0.501 | 431.1 K | 5.58 M | 0.575 | 477.4 K | 6.18 M | 0.597 | 523.9 K | 6.78 M |
| DLinear | 0.583 | 1.6 M | 10.89 M | 0.672 | 2.1 M | 14.52 M | 0.652 | 2.59 M | 18.15 M |

Table 10: Further prediction length on ETTh1 The input length is set as 720 for all models.

| | ETTh1 ‖ 1080 | | | ETTh1 ‖ 1440 | | | ETTh1 ‖ 1800 | | |
|---|---|---|---|---|---|---|---|---|---|
| | MSE | Param | MAC | MSE | Param | MAC | MSE | Param | MAC |
| **TimeBase** | **0.551** | **0.5 K** | **0.07 M** | **0.636** | **0.6 K** | **0.09 M** | **0.714** | **0.7 K** | **0.1 M** |
| iTransformer | 0.602 | 431.1 K | 5.58 M | 0.708 | 477.4 K | 6.18 M | 0.812 | 523.9 K | 6.78 M |
| DLinear | 0.582 | 160 M | 10.89 M | 0.693 | 2.1 M | 14.52 M | 0.796 | 2.59 M | 18.15 M |

## H    EXTENSION TO MULTI-SEASONALITY

In time series forecasting, many real-world datasets exhibit multiple seasonalities, which are crucial for accurate modeling and prediction. For example, traffic data often contains both daily and weekly seasonal patterns, which can significantly influence forecasting accuracy. TimeBase, initially designed for univariate time series forecasting, can be extended to handle such multi-seasonal data by learning distinct period bases for each individual seasonality and combining their outputs to generate more accurate predictions. This extension allows TimeBase to model complex periodic patterns while retaining its minimalistic architecture. Mathematically, the multi-seasonal extension of TimeBase can be formulated as follows:

$$\text{MSTimeBase} = \sum_i \text{TimeBase}(\mathbf{X}; \mathbf{P} = p_i) \tag{33}$$

where $\mathbf{X}$ represents the input data and $\mathbf{P} = p_i$ denotes the different period bases corresponding to each seasonality. By learning multiple period bases ($p \in [24, 168]$ hours, for example), the model can capture both short-term and long-term seasonal patterns and combine them to enhance the accuracy of the forecast.

To evaluate this extension, we applied the multi-seasonality approach to the Traffic dataset, considering both daily and weekly seasonalities. The results, summarized in Table 11, demonstrate that incorporating multiple seasonalities into TimeBase improves the prediction performance with only a modest increase in computational cost and model complexity. Specifically, the model's Mean Squared Error (MSE) and Mean Absolute Error (MAE) improve when compared to the original TimeBase model, despite only a slight increase in the number of parameters and computational cost. This extension underscores the versatility and power of TimeBase in dealing with complex, multi-seasonal patterns, making it suitable for a wide range of long-term time series forecasting (LTSF) tasks. The ability to extend TimeBase while maintaining its lightweight nature reflects its potential for scalable deployment in real-world applications, where seasonalities often vary and need to be captured for accurate forecasting.

Table 11: Performance of TimeBase extended to multi-seasonality. The prediction length is 720 for Traffic dataset.

| Model | MSE | MAE | MACs | Params | Basis_num |
|---|---|---|---|---|---|
| iTransformer | **0.450** | 0.313 | 1.01 G | 11.61 M | - |
| TimeBase | 0.456 | 0.301 | **9.93 M** | 0.51 K | 8 |
| MSTimeBase | 0.451 | **0.295** | 16.76 M | **0.49 K** | 6 |

## I    SUB-SEQUENCE LENGTH ANALYSIS

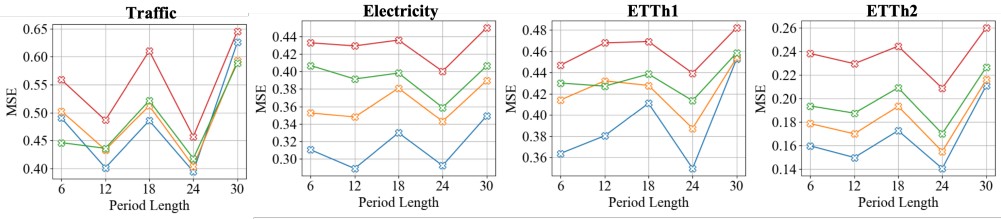

Figure 9: MSE results for different segmentation lengths ($P = [6, 12, 18, 24, 30]$) across various prediction lengths on Traffic, Electricity, ETTh1, and ETTh2 datasets.

This section explores the impact of segmentation length on the forecasting performance of TimeBase across the Traffic, Electricity, ETTh1, and ETTh2 datasets. The analysis evaluates how varying the length of sub-sequence segments, denoted as $P$, affects prediction accuracy for different forecasting horizons. Figure 9 shows that across all datasets and forecasting horizons, the best performance is consistently achieved when the segmentation length is set to $P = 24$. In contrast, shorter or longer

segmentation lengths ($P = 6, 18, 30$) result in noticeably higher MSE values, indicating suboptimal performance. This suggests that the choice of segmentation length significantly affects the model's ability to capture periodic patterns effectively. The superior performance at $P = 24$ highlights the importance of aligning the segmentation length with the inherent periodicity of the data. Deviation from this optimal segmentation length reduces the model's capacity to accurately represent the underlying time series dynamics, thus leading to a degradation in forecasting accuracy. This analysis underscores the necessity of selecting an appropriate segmentation length that corresponds to the periodic nature of the data. The findings suggest that segmenting the time series into periods of $P = 24$ yields the most representative and predictive sub-sequences, enhancing overall forecasting performance.

## J    EFFECTIVENESS ON SYNTHETIC DATA

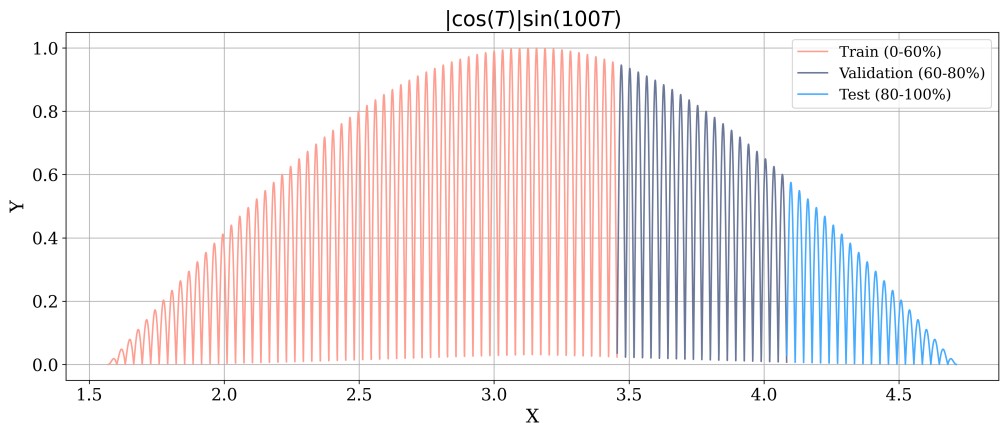

Figure 10: Visualization of the synthetic data generated using the equation $Y = |\cos(X)|\sin(100X)$. The dataset comprises 5000 samples, which are split into training (60%), validation (20%), and testing (20%) sets.

Table 12: Performance of TimeBase on synthetic data for a forecasting length of 100.

| Model | MSE | MAE | MACs | Params |
|---|---|---|---|---|
| TimeBase | 0.007 | 0.070 | 0.077M | 0.19K |
| DLinear | 0.015 | 0.081 | 0.1M | 100K |
| PatchTST | 0.011 | 0.085 | 1.28M | 219K |

Table 13: Performance of TimeBase on synthetic data for a forecasting length of 200.

| Model | MSE | MAE | MACs | Params |
|---|---|---|---|---|
| TimeBase | 0.010 | 0.082 | 0.093M | 0.23K |
| DLinear | 0.021 | 0.124 | 0.2M | 200K |
| PatchTST | 0.017 | 0.083 | 1.48M | 420K |

Table 14: Performance of TimeBase on synthetic data for a forecasting length of 300.

| Model | MSE | MAE | MACs | Params |
|---|---|---|---|---|
| TimeBase | 0.013 | 0.094 | 0.099M | 0.26K |
| DLinear | 0.039 | 0.180 | 0.3M | 300K |
| PatchTST | 0.023 | 0.136 | 1.69M | 622K |

The synthetic dataset shown in Figure 10, derived from the function $Y = |\cos(X)| \sin(100X)$, provides a controlled and challenging benchmark to evaluate the forecasting performance of TimeBase and compare it against state-of-the-art models like DLinear and PatchTST. The data was sampled with high frequency to capture intricate oscillations, divided into training, validation, and testing sets in a 6:2:2 ratio. The basis number is set as 6 for TimeBase to achieve efficient basis extraction. The results of different forecating length are shown in Table 12, 13, 14.The synthetic dataset, generated using $Y = |\cos(X)| \sin(100X)$, serves as a rigorous benchmark for assessing the forecasting capabilities of TimeBase compared to models like DLinear and PatchTST. TimeBase consistently demonstrated superior accuracy and efficiency across varying prediction lengths. For a forecasting length of 100, it achieved an MSE of 0.007 and an MAE of 0.070, utilizing only 0.19K parameters and 0.077M MACs—substantially less than its competitors. At a forecasting length of 200, TimeBase maintained robust accuracy (MSE = 0.010, MAE = 0.082) while still requiring minimal resources (0.23K parameters, 0.093M MACs). For a length of 300, it continued to excel in efficiency and delivered competitive accuracy (MSE = 0.013, MAE = 0.094), outperforming DLinear and approaching PatchTST's accuracy but at a fraction of the computational cost. These results emphasize TimeBase's ability to deliver precise, efficient forecasts, making it an ideal solution for resource-constrained environments while retaining adaptability across diverse time-series scenarios. These results highlight the efficacy of TimeBase in balancing precision and efficiency. Its minimalistic design achieves state-of-the-art accuracy for synthetic data forecasting while consuming significantly fewer computational resources. This makes TimeBase particularly suitable for resource-constrained environments, such as IoT devices and edge computing scenarios. Additionally, its performance stability across varying forecasting horizons underscores its adaptability to different time-series dynamics.

## K  COMPARISON OF PARAMETER NUMBERS FOR TIMEBASE AND BASELINE MODELS AT DIFFERENT MAINSTREAM INPUT LENGTHS

Table 15: Comparison of parameter numbers for TimeBase and baseline models at different input lengths in [96, 192, 336, 720]

| Input length | 96 | 192 | 336 | 720 |
|---|---|---|---|---|
| TimeBase | 0.24K | 0.26K | 0.30K | 0.39 K |
| iTransformer | 5.15M | 5.20M | 5.27M | 5.47M |
| DLinear | 0.13M | 0.27M | 0.49M | 1.03M |
| PatchTST | 1.5M | 2.61M | 4.27M | 8.69M |

In the Table 15, we present a detailed comparison of the parameter numbers for TimeBase and several prominent baseline models (iTransformer, DLinear, and PatchTST) across a range of input lengths. The input lengths vary from 96 to 720, covering both smaller and larger time series windows to evaluate how the parameter count scales with respect to input size. As shown in the table, TimeBase exhibits an exceptional parameter efficiency. For input lengths of 96, 192, 336, and 720, the parameter count increases from 0.24K to 0.39K. Notably, even at the longest input length (720), TimeBase maintains a remarkably low number of parameters compared to the baselines, making it an ideal choice for efficient time series forecasting, especially when dealing with large datasets or resource-constrained environments. In contrast, the baseline models show a significant increase in the number of parameters as the input length grows. iTransformer, for instance, starts with 5.15M parameters at input length 96 and escalates to 5.47M at input length 720. DLinear, although more efficient than iTransformer, still demonstrates a steep parameter increase, from 0.13M at 96 to 1.03M at 720. PatchTST, known for its high complexity, shows the most dramatic increase in parameter numbers, reaching up to 8.69M for an input length of 720. This parameter analysis highlights the core strength of TimeBase: its ability to scale effectively with input length while maintaining an ultra-compact parameter footprint. This is crucial in practical applications where resource limitations such as memory and computation time are a concern, allowing TimeBase to provide an optimal balance between model complexity and performance.

Table 16: Performance of TimeBase under different learning rate on traffic. The batch size is fixed as 128, input length is 720, forecasting length is 96, basis number is 6, orthogonal weight is 0.04.

| Learning Rate | 2e-1 | 1e-1 | 9e-2 | 8e-2 | 7e-2 | 6e-2 | 5e-2 | 4e-2 | **3e-2** | 2e-2 | 1e-2 |
|---|---|---|---|---|---|---|---|---|---|---|---|
| MSE | 0.395 | 0.394 | 0.396 | 0.394 | 0.393 | 0.394 | 0.394 | 0.398 | **0.394** | 0.395 | 0.395 |
| MAE | 0.268 | 0.268 | 0.271 | 0.268 | 0.268 | 0.267 | 0.267 | 0.269 | **0.267** | 0.268 | 0.269 |

Table 17: Performance of TimeBase under different learning rate on ETTh2. The batch size is fixed as 512, input length is 720, forecasting length is 96, basis number is 6, orthogonal weight is 0.2.

| Learning Rate | **2e-1** | 1e-1 | 9e-2 | 8e-2 | 7e-2 | 6e-2 | 5e-2 | 4e-2 | 3e-2 | 2e-2 | 1e-2 |
|---|---|---|---|---|---|---|---|---|---|---|---|
| MSE | **0.292** | 0.292 | 0.293 | 0.294 | 0.294 | 0.297 | 0.294 | 0.293 | 0.293 | 0.311 | 0.345 |
| MAE | **0.349** | 0.351 | 0.352 | 0.352 | 0.351 | 0.352 | 0.351 | 0.351 | 0.351 | 0.369 | 0.402 |

Table 18: Performance of TimeBase under different batch size on traffic. The learning rate is fixed as 3e-2, input length is 720, forecasting length is 96, basis number is 6, orthogonal weight is 0.04.

| Batch Size | 512 | 256 | **128** | 64 |
|---|---|---|---|---|
| MSE | 0.402 | 0.396 | **0.394** | 0.395 |
| MAE | 0.279 | 0.273 | **0.267** | 0.268 |

Table 19: Performance of TimeBase under different batch size on ETTh2. The learning rate is fixed as 2e-1, input length is 720, forecasting length is 96, basis number is 6, orthogonal weight is 0.2.

| Batch Size | **512** | 256 | 128 | 64 |
|---|---|---|---|---|
| MSE | **0.292** | 0.295 | 0.294 | 0.294 |
| MAE | **0.349** | 0.352 | 0.351 | 0.353 |

## L  PERFORMANCE UNDER DIFFERENT BATCHSIZE AND LEARNING RATE

The performance evaluation of TimeBase under varying learning rates and batch sizes provides valuable insights into its training dynamics and adaptability. As demonstrated in Tables 16 and 17, the model consistently achieves competitive results across a broad range of relatively large learning rates. For the Traffic dataset, a learning rate of $3 \times 10^{-2}$ strikes the optimal balance, resulting in an MSE of 0.394 and an MAE of 0.267. Similarly, on the ETTh2 dataset, a larger learning rate of $2 \times 10^{-1}$ yields the best performance, with an MSE of 0.292 and an MAE of 0.349. **These results highlight a notable characteristic of TimeBase: its ability to remain robust and effective even when subjected to relatively high learning rates.** This trait can be attributed to the inherently small parameter count of TimeBase, which makes the model less prone to overfitting and more tolerant of large gradient updates. Unlike more complex models that require finely tuned small learning rates to prevent instability, TimeBase can leverage larger learning rates to accelerate convergence without sacrificing accuracy. This adaptability not only enhances training efficiency but also reduces the need for extensive hyperparameter tuning, a practical advantage in real-world applications. In Tables 18 and 19, the impact of batch size on the model's performance reveals an interesting pattern. For the Traffic dataset, the model achieves its best results with a moderate batch size of 128 (MSE: 0.394, MAE: 0.267), whereas for the ETTh2 dataset, the optimal performance occurs with a larger batch size of 512 (MSE: 0.292, MAE: 0.349). However, these variations should not be overinterpreted, as in theory, changes in batch size should not significantly affect the model's performance if the learning rate is appropriately adjusted. The observed sensitivity to batch size in the Traffic dataset, where a larger batch size of 512 results in a noticeable drop in performance, is likely attributed to the fixed learning rate of $3 \times 10^{-2}$. This relatively small learning rate may have slowed the convergence speed, especially since the model was trained for only 30 epochs.

Table 20: Performance of TimeBase as a Plug-and-Play Component for Patch-Based Methods. The input length is set as 720.

| | | PatchTST+TimeBase | | | | PatchTST | | | | *Reduction* | |
|---|---|---|---|---|---|---|---|---|---|---|---|
| | | MSE | MAE | MACs | Params | MSE | MAE | MACs | Params | MACs | Params |
| ETTh1 | 96 | **0.364** | **0.398** | **0.77 M** | **0.03 M** | 0.377 | 0.408 | 11.05 M | 0.15 M | *93.00%* | *83.15%* |
| | 192 | **0.402** | **0.424** | **0.84 M** | **0.04 M** | 0.413 | 0.431 | 12.02 M | 0.29 M | *93.03%* | *87.93%* |
| | 336 | **0.423** | **0.437** | **1.24 M** | **0.06 M** | 0.436 | 0.446 | 13.47 M | 0.50 M | *90.81%* | *87.98%* |
| | 720 | 0.475 | 0.49 | **1.58 M** | **0.11 M** | **0.455** | **0.475** | 17.34 M | 1.05 M | *90.87%* | *89.59%* |
| ETTh2 | 96 | **0.275** | **0.339** | **1.28 M** | **0.03 M** | 0.276 | 0.339 | 11.05 M | 0.15 M | *88.42%* | *79.08%* |
| | 192 | **0.334** | **0.381** | **1.39 M** | **0.05 M** | 0.342 | 0.385 | 12.02 M | 0.29 M | *88.46%* | *83.68%* |
| | 336 | **0.36** | **0.407** | **1.25 M** | **0.06 M** | 0.364 | 0.405 | 13.47 M | 0.50 M | *90.75%* | *87.98%* |
| | 720 | 0.397 | 0.436 | **1.19 M** | **0.09 M** | **0.395** | **0.434** | 17.34 M | 1.05 M | *93.16%* | *91.79%* |
| ETTm1 | 96 | **0.29** | **0.345** | **28.87 M** | **0.52 M** | 0.298 | 0.352 | 258.69 M | 1.51 M | *88.84%* | *65.20%* |
| | 192 | **0.331** | **0.368** | **29.73 M** | **0.65 M** | 0.335 | 0.373 | 266.43 M | 2.61 M | *88.84%* | *75.23%* |
| | 336 | **0.364** | **0.386** | **31.02 M** | **0.83 M** | 0.366 | 0.394 | 278.05 M | 4.27 M | *88.84%* | *80.53%* |
| | 720 | **0.419** | **0.416** | **34.46 M** | **1.32 M** | 0.42 | 0.421 | 309.01 M | 8.69 M | *88.85%* | *84.78%* |
| ETTm2 | 96 | **0.165** | **0.256** | **57.59 M** | **0.65 M** | 0.165 | 0.26 | 258.69 M | 1.51 M | *77.74%* | *57.07%* |
| | 192 | **0.222** | **0.293** | **29.73 M** | **0.65 M** | 0.219 | 0.298 | 266.43 M | 2.61 M | *88.84%* | *75.23%* |
| | 336 | **0.273** | **0.332** | **61.89 M** | **1.26 M** | 0.268 | 0.333 | 278.05 M | 4.27 M | *77.74%* | *70.47%* |
| | 720 | **0.353** | **0.385** | **68.77 M** | **2.24 M** | 0.352 | 0.386 | 309.01 M | 8.69 M | *77.74%* | *74.19%* |
| Weather | 96 | **0.145** | **0.195** | **86.60 M** | **0.52 M** | 0.149 | 0.199 | 776.08 M | 1.51 M | *88.84%* | *65.20%* |
| | 192 | **0.189** | **0.238** | **89.18 M** | **0.65 M** | 0.193 | 0.243 | 799.30 M | 2.61 M | *88.84%* | *75.23%* |
| | 336 | 0.243 | 0.284 | **92.62 M** | **0.83 M** | **0.24** | **0.281** | 834.14 M | 4.27 M | *88.90%* | *80.57%* |
| | 720 | 0.314 | **0.334** | **103.38 M** | **1.32 M** | **0.312** | 0.334 | 927.04 M | 8.69 M | *88.85%* | *84.78%* |
| Electricity | 96 | **0.128** | **0.223** | **1.32 G** | **0.52 M** | 0.141 | 0.24 | 11.86 G | 1.51 M | *88.84%* | *65.20%* |
| | 192 | **0.145** | **0.238** | **1.36 G** | **0.65 M** | 0.156 | 0.256 | 12.22 G | 2.61 M | *88.84%* | *75.23%* |
| | 336 | **0.16** | **0.255** | **1.42 G** | **0.83 M** | 0.172 | 0.267 | 12.75 G | 4.27 M | *88.84%* | *80.53%* |
| | 720 | **0.197** | **0.288** | **1.58 G** | **1.32 M** | 0.207 | 0.299 | 14.17 G | 8.69 M | *88.85%* | *84.78%* |
| Traffic | 96 | **0.36** | 0.252 | **4.27 G** | **0.55 M** | 0.363 | **0.25** | 31.86 G | 1.51 M | *86.61%* | *63.57%* |
| | 192 | **0.371** | **0.256** | **4.39 G** | **0.70 M** | 0.382 | 0.258 | 32.81 G | 2.61 M | *86.61%* | *73.35%* |
| | 336 | **0.396** | **0.278** | **4.58 G** | **0.92 M** | 0.399 | 0.268 | 34.24 G | 4.27 M | *86.61%* | *78.52%* |
| | 720 | **0.422** | **0.284** | **5.09 G** | **1.51 M** | 0.432 | 0.289 | 38.05 G | 8.69 M | *86.62%* | *82.66%* |

## M  PLUG-AND-PLAY FRAMEWORK FOR PATCH-BASED TIME SERIES FORECASTING

Furthermore, TimeBase can serve as a plug-and-play tool to extremely reduce resource consumption in patch-based approaches. In implementation, after segmenting the time series into patches, TimeBase can expertly be employed to patch-based methods to extract the basis components. The experimental results shown in Table 20 underscore the exceptional efficiency and adaptability of the TimeBase framework when integrated with the PatchTST model. By preserving input sequence lengths of 720 across all datasets, we ensured a fair comparison of the models' predictive performance, computational complexity, and parameter counts. The findings reveal that TimeBase achieves comparable or even slightly improved accuracy metrics, such as MSE and MAE, while significantly reducing both the number of parameters and the computational load. On the ETTh1 dataset, TimeBase demonstrates remarkable reductions in MACs and parameters by approximately 90.87% and 89.59%, respectively, for the longest horizon of 720, without compromising the MSE and MAE results compared to the standalone PatchTST model. Similar trends are observed across

other datasets, including ETTh2, ETTm1, and Traffic, where TimeBase achieves substantial reductions in computational resources, exceeding 85%, while maintaining or enhancing forecasting accuracy. These reductions are particularly notable in resource-intensive datasets like Electricity and Traffic, where MACs are reduced by over 88%, showcasing TimeBase's ability to efficiently handle large-scale data. The reduced parameter counts achieved by TimeBase are critical in real-world scenarios requiring lightweight and scalable solutions. Despite its minimalist architecture, TimeBase effectively captures essential patterns and interactions within the time series data, allowing it to retain or even slightly improve accuracy. Its plug-and-play nature enables seamless integration with existing patch-based models, allowing users to achieve significant gains in efficiency without re-engineering their frameworks. These results demonstrate that TimeBase is not only an effective model in its own right but also a transformative enhancement for patch-based forecasting methods. By drastically reducing computational and memory overheads while maintaining high predictive accuracy, TimeBase sets a new benchmark for minimalist, efficient, and scalable time series forecasting solutions.

## N   FULL FORECASTING RESULTS OF 720 INPUT LENGTH

The full forecasting results of 720 input length are shown in Table. 21.

## O   FULL FORECASTING RESULTS OF 336 INPUT LENGTH

To further validate the effectiveness of TimeBase, we conducted experiments with a shorter input length of 336 to complement the primary results using an input length of 720. The full results for this setting are presented in Table 22, comparing TimeBase against state-of-the-art baselines across multiple datasets and prediction horizons. TimeBase consistently ranks among the top performers across different metrics, particularly excelling in short-term horizons like 96 and 192, where it achieves the lowest MSE and MAE in most cases. Notably, even with reduced input length, TimeBase maintains its ability to produce competitive results against more complex models such as PatchTST and TimesNet. This demonstrates its robustness and adaptability across varying input configurations. Moreover, the efficiency analysis in Table 23 reveals that TimeBase's minimalist design continues to shine under this setting. Its architecture, with a minimal parameter count and low computational overhead, ensures faster training and inference times compared to transformer-based models like FEDformer and Autoformer. This makes it particularly appealing for applications requiring both accuracy and scalability. Importantly, the results underline TimeBase's capacity to generalize well, even when provided with less temporal information, solidifying its position as a versatile model for long-term time series forecasting.

## P   FULL FORECASTING RESULTS OF 96 INPUT LENGTH

In experiments with a 96 input length, we observed that the performance of most methods, including Transformer-based models such as iTransformer and PatchTST, as well as decomposition-based approaches like TimesNet, declined to varying degrees. This indicates that shorter input lengths may limit the models' ability to capture long-term dependencies and global features effectively. Notably, TimesNet demonstrated some resilience under shorter input lengths due to its unique time-frequency decomposition mechanism. Similarly, TimeBase, with its minimalistic parameter, maintained relatively stable performance even with limited input lengths. As shown in Table 24, the performance of Transformer-based methods generally deteriorates at a 96 input length, underscoring their reliance on longer time series for effective modeling. However, TimeBase, leveraging its segmentation and basis signal extraction design, exhibits strong adaptability in such constrained scenarios, highlighting its robustness and efficiency across diverse forecasting settings.

Table 21: Full results of long-term time series forecasting, comparing TimeBase with other baselines. The top 4 results are highlighted in bold. **To ensure fair comparison, we have addressed the following concerns:(1) the `"drop_last=True"` bug in `test_loader` is corrected in all models; (2) input lengths of all models are set to 720.** **All scripts and logs for running our model and baselines are available at `https://anonymous.4open.science/r/TimeBase-fixbug`.**

| Methods | | TimeBase (ours) | | SparseTSF (2024) | | TIME-LLM (2024) | | FITS (2024) | | iTransformer (2024a) | | DLinear (2023) | | PatchTST (2023) | | TimesNet (2023) | | FEDformer (2022) | | Autoformer (2021) | | Informer (2021) | |
|---|---|---|---|---|---|---|---|---|---|---|---|---|---|---|---|---|---|---|---|---|---|---|---|
| Metric | | MSE | MAE | MSE | MAE | MSE | MAE | MSE | MAE | MSE | MAE | MSE | MAE | MSE | MAE | MSE | MAE | MSE | MAE | MSE | MAE | MSE | MAE |
| ETTh1 | 96 | **0.349** | **0.384** | **0.362** | **0.389** | 0.390 | 0.420 | 0.380 | **0.402** | 0.389 | 0.421 | **0.378** | **0.402** | **0.377** | 0.408 | 0.437 | 0.454 | 0.485 | 0.500 | 0.555 | 0.558 | 1.269 | 0.855 |
| | 192 | **0.387** | **0.410** | **0.404** | **0.412** | 0.427 | 0.443 | **0.415** | **0.424** | 0.424 | 0.446 | 0.415 | **0.425** | **0.413** | 0.431 | 0.456 | 0.469 | 0.481 | 0.498 | 0.599 | 0.575 | 1.487 | 0.943 |
| | 336 | **0.408** | **0.418** | **0.435** | **0.428** | 0.459 | 0.467 | **0.439** | **0.439** | 0.456 | 0.469 | 0.449 | 0.449 | **0.436** | **0.446** | 0.494 | 0.494 | 0.522 | 0.521 | 0.853 | 0.702 | 1.544 | 0.945 |
| | 720 | **0.439** | **0.446** | **0.426** | **0.448** | 0.452 | 0.476 | **0.433** | **0.457** | 0.545 | 0.532 | 0.507 | 0.517 | **0.455** | **0.475** | 0.632 | 0.578 | 0.604 | 0.575 | 0.899 | 0.730 | 1.481 | 0.975 |
| ETTh2 | 96 | **0.292** | 0.350 | 0.294 | **0.346** | 0.300 | 0.362 | **0.271** | **0.336** | 0.305 | 0.361 | 0.294 | 0.360 | **0.276** | **0.339** | 0.349 | 0.403 | 0.401 | 0.451 | 0.541 | 0.559 | 5.189 | 1.812 |
| | 192 | **0.341** | 0.387 | **0.340** | **0.377** | 0.365 | 0.395 | **0.332** | **0.374** | 0.405 | 0.421 | 0.412 | 0.437 | **0.342** | **0.385** | 0.500 | 0.488 | 0.425 | 0.464 | 1.207 | 0.866 | 6.514 | 2.011 |
| | 336 | **0.358** | 0.410 | **0.360** | **0.398** | 0.367 | 0.417 | **0.355** | **0.396** | 0.411 | 0.436 | 0.471 | 0.478 | **0.364** | **0.405** | 0.445 | 0.465 | 0.427 | 0.471 | 0.825 | 0.719 | 5.284 | 1.859 |
| | 720 | **0.400** | 0.448 | **0.383** | **0.425** | 0.411 | 0.449 | **0.378** | **0.423** | 0.448 | 0.470 | 0.740 | 0.609 | **0.395** | **0.434** | 0.438 | 0.465 | 0.462 | 0.493 | 1.772 | 1.062 | 4.955 | 1.884 |
| ETTm1 | 96 | **0.310** | **0.354** | 0.314 | 0.359 | 0.316 | 0.366 | **0.313** | **0.357** | 0.315 | 0.369 | **0.307** | **0.350** | 0.298 | 0.352 | 0.359 | 0.391 | 0.406 | 0.441 | 0.455 | 0.464 | 0.632 | 0.574 |
| | 192 | **0.338** | **0.371** | 0.348 | 0.376 | **0.338** | 0.379 | **0.339** | **0.369** | 0.349 | 0.388 | 0.347 | 0.381 | **0.335** | **0.373** | 0.368 | 0.398 | 0.450 | 0.477 | 0.562 | 0.514 | 1.131 | 0.802 |
| | 336 | **0.364** | **0.386** | 0.368 | 0.386 | 0.368 | 0.396 | **0.367** | **0.385** | 0.381 | 0.409 | **0.367** | 0.387 | **0.366** | 0.394 | 0.429 | 0.438 | 0.436 | 0.466 | 0.737 | 0.608 | 1.391 | 0.923 |
| | 720 | **0.413** | **0.414** | 0.419 | 0.413 | 0.430 | 0.435 | **0.417** | **0.417** | 0.437 | 0.439 | **0.415** | **0.415** | 0.420 | 0.421 | 0.477 | 0.474 | 0.462 | 0.479 | 0.503 | 0.502 | 1.397 | 0.973 |
| ETTm2 | 96 | **0.166** | **0.256** | 0.167 | 0.259 | 0.183 | 0.271 | **0.166** | **0.256** | 0.179 | 0.274 | **0.163** | **0.257** | **0.165** | 0.260 | 0.200 | 0.288 | 0.339 | 0.406 | 0.325 | 0.391 | 1.870 | 1.002 |
| | 192 | **0.218** | **0.293** | **0.219** | **0.297** | 0.241 | 0.313 | **0.218** | **0.293** | 0.239 | 0.314 | 0.223 | 0.304 | **0.219** | **0.298** | 0.274 | 0.337 | 0.397 | 0.452 | 0.369 | 0.414 | 2.807 | 1.314 |
| | 336 | **0.270** | **0.328** | 0.271 | 0.330 | 0.292 | 0.345 | **0.271** | **0.328** | 0.309 | 0.356 | 0.291 | 0.355 | **0.268** | **0.333** | 0.340 | 0.382 | 0.449 | 0.491 | 0.418 | 0.452 | 4.442 | 1.661 |
| | 720 | **0.352** | **0.380** | 0.353 | 0.380 | 0.362 | 0.392 | **0.352** | **0.380** | 0.387 | 0.407 | 0.407 | 0.433 | **0.352** | **0.386** | 0.384 | 0.407 | 0.451 | 0.499 | 0.612 | 0.594 | 5.258 | 1.914 |
| Weather | 96 | **0.174** | **0.230** | 0.174 | 0.231 | **0.155** | **0.212** | 0.176 | 0.232 | **0.159** | **0.212** | 0.174 | 0.242 | **0.149** | **0.199** | 0.176 | 0.234 | 0.289 | 0.342 | 0.323 | 0.389 | 0.283 | 0.361 |
| | 192 | **0.215** | **0.264** | 0.216 | 0.267 | **0.191** | **0.242** | 0.216 | 0.268 | **0.203** | **0.252** | 0.215 | 0.277 | **0.193** | **0.243** | 0.219 | 0.270 | 0.340 | 0.394 | 0.389 | 0.423 | 0.445 | 0.461 |
| | 336 | **0.260** | **0.299** | 0.260 | 0.299 | **0.246** | **0.286** | 0.261 | 0.299 | **0.253** | **0.291** | 0.262 | 0.319 | **0.240** | **0.281** | 0.277 | 0.311 | 0.370 | 0.408 | 0.497 | 0.495 | 0.587 | 0.526 |
| | 720 | **0.323** | **0.343** | 0.325 | 0.345 | **0.313** | **0.331** | 0.325 | 0.346 | **0.317** | **0.337** | 0.319 | 0.359 | **0.312** | **0.334** | 0.344 | 0.356 | 0.420 | 0.421 | 0.573 | 0.520 | 0.953 | 0.703 |
| Electricity | 96 | **0.139** | **0.231** | **0.139** | 0.239 | **0.135** | 0.235 | 0.147 | 0.235 | **0.135** | **0.233** | 0.141 | 0.244 | 0.141 | 0.240 | 0.202 | 0.308 | 0.226 | 0.341 | 0.225 | 0.334 | 0.395 | 0.460 |
| | 192 | **0.153** | **0.245** | **0.155** | **0.250** | 0.156 | 0.262 | 0.159 | 0.256 | **0.155** | **0.253** | 0.155 | 0.258 | 0.156 | **0.256** | 0.218 | 0.322 | 0.220 | 0.336 | 0.223 | 0.332 | 0.405 | 0.460 |
| | 336 | **0.169** | **0.262** | 0.167 | 0.265 | **0.160** | **0.248** | 0.169 | 0.270 | 0.169 | 0.267 | 0.170 | 0.275 | 0.172 | **0.267** | 0.232 | 0.332 | 0.224 | 0.337 | 0.233 | 0.341 | 0.404 | 0.460 |
| | 720 | **0.208** | **0.294** | 0.208 | 0.300 | **0.197** | 0.298 | 0.214 | 0.302 | **0.204** | 0.301 | 0.209 | 0.309 | **0.207** | **0.299** | 0.299 | 0.375 | 0.271 | 0.378 | 0.261 | 0.364 | 0.429 | 0.477 |
| Traffic | 96 | 0.394 | **0.267** | **0.389** | 0.268 | **0.377** | 0.280 | 0.402 | 0.275 | **0.374** | 0.273 | 0.396 | **0.272** | **0.363** | **0.250** | 0.605 | 0.325 | 0.664 | 0.431 | 0.668 | 0.401 | 0.829 | 0.487 |
| | 192 | 0.403 | **0.274** | **0.399** | 0.270 | **0.385** | 0.281 | 0.419 | 0.286 | **0.393** | 0.283 | 0.404 | **0.275** | **0.382** | **0.258** | 0.627 | 0.340 | 0.613 | 0.382 | 0.703 | 0.439 | 0.902 | 0.525 |
| | 336 | **0.417** | **0.281** | **0.417** | 0.279 | **0.399** | 0.288 | 0.423 | 0.292 | **0.409** | 0.292 | 0.417 | **0.283** | **0.399** | **0.268** | 0.631 | 0.349 | 0.612 | 0.379 | 0.666 | 0.421 | 0.949 | 0.545 |
| | 720 | 0.456 | **0.301** | **0.449** | 0.297 | **0.436** | **0.294** | 0.459 | 0.311 | **0.450** | 0.314 | 0.457 | 0.310 | **0.432** | **0.289** | 0.700 | 0.371 | 0.664 | 0.410 | 0.697 | 0.424 | 1.430 | 0.793 |

Table 22: Full results of long-term time series forecasting under 336 input length, comparing TimeBase with other baselines. The top 4 results are highlighted in bold. **All scripts and logs for running our model and baselines in this settings has also been updated to `https://anonymous.4open.science/r/TimeBase-fixbug`.**

| Methods | | TimeBase (ours) | | SparseTSF (2024) | | FITS (2024) | | iTransformer (2024a) | | DLinear (2023) | | PatchTST (2023) | | TimesNet (2023) | | FEDformer (2022) | | Autoformer (2021) | | Informer (2021) | |
|---|---|---|---|---|---|---|---|---|---|---|---|---|---|---|---|---|---|---|---|---|---|
| Metric | | MSE | MAE | MSE | MAE | MSE | MAE | MSE | MAE | MSE | MAE | MSE | MAE | MSE | MAE | MSE | MAE | MSE | MAE | MSE | MAE |
| ETTh1 | 96 | **0.362** | **0.382** | 0.403 | **0.414** | **0.376** | **0.397** | 0.399 | 0.418 | **0.398** | 0.418 | **0.385** | **0.406** | 0.423 | 0.437 | 0.405 | 0.442 | 0.483 | 0.479 | 0.991 | 0.766 |
| | 192 | **0.399** | **0.404** | **0.424** | **0.426** | **0.407** | **0.414** | 0.448 | 0.449 | 0.432 | 0.438 | **0.414** | **0.421** | 0.483 | 0.483 | 0.434 | 0.459 | 0.488 | 0.489 | 1.210 | 0.876 |
| | 336 | **0.424** | **0.419** | **0.429** | **0.431** | **0.430** | **0.428** | 0.466 | 0.463 | 0.458 | 0.458 | **0.441** | **0.440** | 0.490 | 0.479 | **0.438** | 0.462 | 0.524 | 0.514 | 1.149 | 0.822 |
| | 720 | **0.434** | **0.439** | **0.436** | **0.446** | **0.436** | **0.453** | 0.505 | 0.507 | 0.505 | 0.516 | **0.456** | **0.471** | 0.618 | 0.559 | 0.508 | 0.510 | 0.605 | 0.590 | 1.267 | 0.905 |
| ETTh2 | 96 | **0.295** | **0.347** | **0.289** | **0.344** | **0.274** | **0.337** | 0.299 | 0.358 | 0.328 | 0.386 | **0.275** | **0.337** | 0.352 | 0.404 | 0.384 | 0.429 | 0.494 | 0.506 | 3.888 | 1.561 |
| | 192 | **0.345** | **0.384** | **0.345** | **0.382** | **0.335** | **0.376** | 0.368 | 0.399 | 0.420 | 0.443 | **0.339** | **0.379** | 0.403 | 0.432 | 0.419 | 0.454 | 0.492 | 0.523 | 3.974 | 1.583 |
| | 336 | **0.361** | **0.398** | **0.365** | **0.400** | **0.359** | **0.397** | 0.423 | 0.437 | 0.505 | 0.495 | **0.368** | **0.400** | 0.390 | 0.429 | 0.409 | 0.454 | 0.939 | 0.711 | 3.133 | 1.403 |
| | 720 | **0.397** | **0.436** | **0.398** | **0.434** | **0.396** | **0.433** | 0.427 | 0.447 | 0.778 | 0.628 | **0.391** | **0.429** | 0.472 | 0.483 | 0.419 | 0.472 | 0.790 | 0.664 | 3.017 | 1.479 |
| ETTm1 | 96 | **0.298** | **0.346** | 0.306 | 0.347 | **0.303** | **0.345** | 0.318 | 0.358 | **0.303** | **0.347** | **0.293** | **0.343** | 0.330 | 0.375 | 0.328 | 0.395 | 0.601 | 0.530 | 0.762 | 0.655 |
| | 192 | **0.340** | **0.367** | 0.341 | **0.368** | **0.338** | **0.366** | 0.343 | 0.382 | **0.338** | **0.368** | **0.331** | **0.368** | 0.361 | 0.394 | 0.363 | 0.414 | 0.568 | 0.510 | 1.150 | 0.840 |
| | 336 | **0.372** | **0.389** | 0.373 | **0.385** | **0.372** | **0.385** | 0.381 | 0.402 | **0.373** | **0.389** | **0.366** | 0.392 | 0.428 | 0.434 | 0.410 | 0.442 | 0.551 | 0.508 | 1.524 | 0.988 |
| | 720 | **0.428** | **0.421** | 0.429 | **0.417** | 0.429 | **0.416** | 0.439 | 0.436 | **0.428** | **0.424** | **0.419** | 0.425 | 0.461 | 0.455 | 0.448 | 0.469 | 0.520 | 0.507 | 1.176 | 0.837 |
| ETTm2 | 96 | **0.167** | **0.256** | **0.168** | **0.256** | **0.168** | **0.256** | 0.176 | 0.267 | 0.173 | 0.269 | **0.164** | **0.254** | 0.190 | 0.282 | 0.272 | 0.345 | 0.321 | 0.384 | 1.311 | 0.881 |
| | 192 | **0.220** | **0.291** | **0.221** | **0.292** | **0.220** | **0.291** | 0.239 | 0.311 | 0.238 | 0.320 | **0.220** | **0.292** | 0.244 | 0.317 | 0.309 | 0.371 | 0.335 | 0.398 | 1.855 | 1.051 |
| | 336 | **0.275** | **0.326** | **0.277** | **0.327** | **0.274** | **0.326** | 0.287 | 0.341 | 0.316 | 0.375 | **0.277** | **0.329** | 0.294 | 0.346 | 0.346 | 0.392 | 0.468 | 0.480 | 2.581 | 1.298 |
| | 720 | **0.368** | **0.383** | **0.368** | **0.382** | **0.369** | **0.383** | 0.382 | 0.397 | 0.460 | 0.465 | **0.369** | **0.386** | 0.389 | 0.401 | 0.427 | 0.440 | 0.437 | 0.451 | 5.427 | 1.867 |
| Weather | 96 | **0.147** | **0.200** | 0.177 | **0.227** | 0.176 | 0.228 | **0.159** | **0.209** | 0.176 | 0.236 | **0.151** | **0.200** | **0.170** | 0.228 | 0.257 | 0.326 | 0.310 | 0.385 | 0.284 | 0.339 |
| | 192 | **0.189** | **0.241** | 0.221 | 0.264 | 0.219 | 0.262 | **0.203** | **0.249** | 0.218 | 0.278 | **0.195** | **0.242** | **0.214** | **0.262** | 0.293 | 0.349 | 0.371 | 0.424 | 0.488 | 0.473 |
| | 336 | **0.243** | **0.283** | 0.267 | **0.297** | 0.267 | 0.297 | **0.254** | **0.288** | 0.263 | 0.314 | **0.249** | **0.283** | 0.270 | 0.300 | 0.409 | 0.444 | 0.390 | 0.425 | 0.623 | 0.556 |
| | 720 | **0.319** | **0.336** | 0.334 | 0.343 | 0.334 | 0.343 | **0.323** | **0.336** | 0.325 | 0.363 | **0.321** | **0.335** | 0.341 | 0.352 | 0.409 | 0.434 | 0.532 | 0.499 | 1.136 | 0.768 |
| Electricity | 96 | **0.141** | **0.232** | 0.147 | 0.241 | **0.143** | **0.240** | **0.138** | **0.236** | 0.156 | 0.260 | **0.130** | **0.222** | 0.194 | 0.300 | 0.192 | 0.310 | 0.199 | 0.313 | 0.355 | 0.438 |
| | 192 | **0.156** | **0.246** | 0.158 | **0.251** | **0.157** | **0.252** | **0.157** | 0.254 | 0.170 | 0.273 | **0.150** | **0.242** | 0.210 | 0.314 | 0.228 | 0.346 | 0.207 | 0.318 | 0.371 | 0.447 |
| | 336 | **0.171** | **0.261** | 0.174 | **0.268** | **0.173** | **0.268** | 0.174 | 0.272 | 0.185 | 0.289 | **0.166** | **0.260** | 0.220 | 0.323 | 0.251 | 0.366 | 0.223 | 0.334 | 0.384 | 0.456 |
| | 720 | **0.209** | **0.294** | **0.212** | **0.299** | **0.211** | **0.299** | 0.212 | 0.305 | 0.220 | 0.321 | **0.210** | **0.298** | 0.239 | 0.335 | 0.251 | 0.360 | 0.244 | 0.351 | 0.414 | 0.472 |
| Traffic | 96 | 0.423 | **0.281** | **0.415** | **0.279** | **0.412** | **0.282** | **0.413** | 0.303 | 0.469 | 0.350 | **0.367** | **0.250** | 0.616 | 0.343 | 0.646 | 0.420 | 0.662 | 0.406 | 0.766 | 0.434 |
| | 192 | **0.436** | **0.288** | **0.426** | **0.283** | **0.424** | **0.286** | 0.437 | 0.318 | 0.483 | 0.356 | **0.389** | **0.264** | 0.614 | 0.333 | 0.654 | 0.420 | 0.632 | 0.389 | 0.809 | 0.461 |
| | 336 | **0.449** | **0.292** | **0.438** | **0.290** | **0.437** | **0.291** | 0.457 | 0.330 | 0.497 | 0.363 | **0.398** | **0.266** | 0.649 | 0.346 | 0.629 | 0.392 | 0.654 | 0.402 | 0.871 | 0.496 |
| | 720 | **0.479** | **0.310** | **0.465** | **0.305** | **0.465** | **0.308** | 0.497 | 0.354 | 0.526 | 0.378 | **0.457** | **0.311** | 0.659 | 0.363 | 0.647 | 0.398 | 0.658 | 0.406 | 1.059 | 0.605 |

Table 23: Efficiency comparison of TimeBase and other state-of-the-art models on the Electricity dataset with a forecasting length of 720. **To ensure fair comparison, the look-back window is set as 336 for all models.**

| Model | Parameters | MACs | Max Mem.(MB) | Epoch Time(s) | Infer Time (CPU) |
|---|---|---|---|---|---|
| Informer | 12.45M | 5.44G | 1096.51 | 63.35 | 48.59ms |
| Autoformer | 12.14M | 6.16G | 3229.78 | 93.53 | 85.76ms |
| FEDformer | 12.14M | 6.74G | 1700.37 | 451.69 | 128.21ms |
| PatchTST | 6.31M | 11.21G | 10882.3 | 290.3 | 130.13ms |
| DLinear | 485.3K | 156M | 123.8 | 25.4 | 3.33ms |
| iTransformer | 5.27M | 1.72G | 809.45 | 76.31 | 29.51ms |
| SparseTSF | 0.5K | 5.93M | 118.43 | 31.43 | 2.52ms |
| TimeBase(ours) | **0.3K** | **2.03M** | **85.36** | **19.35** | **0.97ms** |

Table 24: Full results of long-term time series forecasting under 96 input length, comparing TimeBase with other baselines. The top 4 results are highlighted in bold. **All scripts and logs for running our model and baselines in this settings has also been updated to `https://anonymous.4open.science/r/TimeBase-fixbug`.**

| Methods | | TimeBase (ours) | | SparseTSF (2024) | | FITS (2024) | | iTransformer (2024a) | | DLinear (2023) | | PatchTST (2023) | | TimesNet (2023) | | FEDformer (2022) | | Autoformer (2021) | | Informer (2021) | |
|---|---|---|---|---|---|---|---|---|---|---|---|---|---|---|---|---|---|---|---|---|---|
| Metric | | MSE | MAE | MSE | MAE | MSE | MAE | MSE | MAE | MSE | MAE | MSE | MAE | MSE | MAE | MSE | MAE | MSE | MAE | MSE | MAE |
| ETTh1 | 96 | **0.381** | **0.39** | **0.385** | **0.391** | **0.385** | 0.393 | 0.393 | **0.408** | 0.428 | 0.435 | 0.399 | 0.41 | 0.389 | 0.412 | **0.373** | 0.412 | 0.443 | 0.453 | 0.963 | 0.78 |
| | 192 | **0.428** | **0.416** | **0.435** | **0.42** | **0.436** | **0.422** | 0.446 | 0.44 | 0.476 | 0.463 | 0.448 | **0.436** | 0.439 | 0.442 | **0.423** | 0.445 | 0.448 | 0.453 | 1.008 | 0.782 |
| | 336 | **0.469** | **0.438** | **0.476** | **0.44** | **0.475** | 0.443 | 0.494 | 0.467 | 0.517 | 0.487 | 0.489 | **0.456** | 0.494 | 0.471 | **0.453** | 0.464 | 0.514 | 0.491 | 1.047 | 0.79 |
| | 720 | **0.467** | **0.461** | **0.472** | **0.464** | **0.466** | **0.459** | 0.516 | 0.501 | 0.543 | 0.53 | 0.488 | **0.476** | 0.516 | 0.494 | **0.476** | 0.491 | 0.505 | 0.506 | 1.151 | 0.845 |
| ETTh2 | 96 | **0.308** | **0.349** | **0.312** | **0.356** | 0.315 | 0.359 | **0.302** | **0.35** | 0.379 | 0.422 | **0.293** | **0.343** | 0.329 | 0.37 | 0.345 | 0.387 | 0.394 | 0.418 | 2.854 | 1.334 |
| | 192 | **0.393** | **0.4** | 0.398 | **0.403** | 0.395 | 0.405 | **0.384** | **0.4** | 0.509 | 0.495 | **0.381** | **0.396** | **0.394** | 0.41 | 0.436 | 0.442 | 0.444 | 0.444 | 6.137 | 2.068 |
| | 336 | **0.426** | **0.433** | **0.429** | **0.438** | 0.437 | 0.443 | **0.424** | **0.433** | 0.626 | 0.558 | **0.418** | **0.428** | 0.471 | 0.468 | 0.508 | 0.496 | 0.472 | 0.478 | 4.823 | 1.842 |
| | 720 | **0.427** | **0.445** | 0.43 | 0.457 | **0.429** | **0.451** | **0.429** | **0.446** | 0.862 | 0.671 | **0.423** | **0.442** | 0.439 | 0.451 | 0.478 | 0.488 | 0.488 | 0.492 | 4.118 | 1.696 |
| ETTm1 | 96 | 0.352 | 0.379 | 0.357 | **0.375** | 0.354 | **0.375** | **0.341** | 0.377 | **0.349** | **0.376** | **0.323** | **0.36** | **0.336** | 0.377 | 0.373 | 0.42 | 0.516 | 0.484 | 0.631 | 0.566 |
| | 192 | **0.388** | **0.394** | 0.394 | **0.393** | 0.392 | **0.393** | **0.382** | 0.396 | **0.387** | 0.395 | **0.36** | **0.382** | 0.389 | 0.402 | 0.407 | 0.434 | 0.545 | 0.498 | 0.736 | 0.626 |
| | 336 | **0.419** | **0.415** | 0.425 | **0.414** | 0.425 | **0.415** | **0.417** | 0.417 | **0.417** | 0.417 | **0.393** | **0.403** | 0.42 | 0.421 | 0.447 | 0.46 | 0.655 | 0.544 | 1.288 | 0.907 |
| | 720 | **0.482** | **0.448** | 0.488 | **0.449** | 0.486 | **0.449** | 0.487 | 0.457 | **0.475** | 0.452 | **0.456** | **0.44** | **0.482** | 0.46 | 0.508 | 0.488 | 0.706 | 0.573 | 0.977 | 0.747 |
| ETTm2 | 96 | **0.182** | **0.263** | 0.185 | **0.267** | **0.182** | **0.266** | **0.184** | 0.269 | 0.198 | 0.297 | **0.176** | **0.259** | 0.188 | 0.268 | 0.189 | 0.282 | 0.268 | 0.329 | 0.383 | 0.464 |
| | 192 | **0.245** | **0.303** | **0.248** | **0.306** | **0.247** | **0.305** | 0.253 | 0.314 | 0.288 | 0.364 | **0.241** | **0.301** | 0.251 | 0.306 | 0.258 | 0.325 | 0.274 | 0.332 | 0.827 | 0.717 |
| | 336 | **0.305** | **0.34** | **0.308** | **0.342** | **0.307** | **0.342** | 0.315 | 0.351 | 0.39 | 0.432 | **0.302** | **0.341** | 0.316 | 0.345 | 0.327 | 0.365 | 0.333 | 0.367 | 1.387 | 0.908 |
| | 720 | **0.406** | **0.397** | **0.418** | **0.408** | **0.417** | 0.408 | 0.418 | 0.409 | 0.56 | 0.526 | **0.41** | **0.404** | 0.423 | **0.407** | 0.435 | 0.425 | 0.458 | 0.44 | 3.906 | 1.474 |
| Weather | 96 | **0.169** | **0.216** | 0.197 | 0.237 | 0.196 | 0.236 | **0.187** | **0.227** | 0.199 | 0.258 | **0.174** | **0.216** | **0.169** | **0.219** | 0.243 | 0.326 | 0.294 | 0.36 | 0.362 | 0.424 |
| | 192 | **0.215** | **0.255** | 0.244 | 0.273 | 0.241 | 0.272 | **0.234** | **0.266** | 0.239 | 0.297 | **0.22** | **0.256** | **0.225** | **0.264** | 0.28 | 0.341 | 0.289 | 0.348 | 0.482 | 0.474 |
| | 336 | **0.271** | **0.296** | 0.293 | 0.308 | 0.293 | 0.308 | 0.288 | **0.305** | **0.285** | 0.334 | **0.275** | **0.295** | **0.282** | **0.304** | 0.37 | 0.409 | 0.363 | 0.403 | 0.434 | 0.445 |
| | 720 | **0.351** | **0.347** | 0.368 | 0.357 | 0.366 | **0.355** | 0.363 | **0.354** | **0.349** | 0.385 | **0.352** | **0.346** | **0.359** | 0.355 | 0.403 | 0.419 | 0.426 | 0.43 | 1.067 | 0.771 |
| Electricity | 96 | 0.203 | **0.275** | 0.21 | **0.28** | 0.207 | 0.28 | **0.171** | **0.261** | 0.225 | 0.316 | **0.169** | **0.255** | 0.211 | 0.31 | **0.187** | 0.302 | **0.202** | 0.317 | 0.321 | 0.408 |
| | 192 | **0.199** | **0.275** | 0.206 | **0.282** | 0.201 | 0.282 | **0.182** | **0.271** | 0.225 | 0.319 | **0.177** | **0.263** | 0.251 | 0.334 | **0.196** | 0.309 | 0.224 | 0.333 | 0.347 | 0.43 |
| | 336 | **0.216** | **0.29** | 0.222 | **0.295** | 0.225 | 0.297 | **0.2** | **0.289** | 0.237 | 0.333 | **0.193** | **0.28** | 0.264 | 0.346 | **0.213** | 0.326 | 0.231 | 0.336 | 0.347 | 0.431 |
| | 720 | **0.247** | **0.321** | 0.26 | **0.328** | 0.256 | 0.329 | **0.243** | **0.324** | 0.272 | 0.363 | **0.233** | **0.313** | 0.298 | 0.37 | **0.24** | 0.35 | 0.292 | 0.39 | 0.391 | 0.454 |

