# OpenReview forum: "TimeBase: The Power of Minimalism in  Long-term Time Series Forecasting"
_ICLR.cc/2025/Conference — ICLR 2025 Conference Withdrawn Submission_

### Official Review · Reviewer_ZKSJ · 2024-10-25

**Soundness:** 1
**Presentation:** 3
**Contribution:** 1
**Rating:** 5
**Confidence:** 5

**Summary:**

This article presents a lightweight model requiring only 0.39k parameters, providing a simple periodic-based method for time-series forecasting and good paper writing.

**Strengths:**

The model exhibits rapid training and inference speeds, and its lightweight nature allows for scalability to edge devices.

**Weaknesses:**

The article has several problems with novelty as well as experimentation.
1) The a priori assumptions about the period of the data in the paper are too strong, and the frequency of data collection and the period characteristics of the data are not equivalent, in the paper, the authors set P=24 for every hour of data collection. in addition, the paper needs to explain in more detail that in the absence of a priori knowledge, the Fourier series is applied to extract the period, and exactly how to achieve this, if the data is realized in the full time series, then there is data leak, if the data is computed once for each input len length, then it is likely that different values of the period P will be obtained.

2) The experiments did not use the same input length for different models to compare, in the past research, the mainstream experimental comparison of the input length of the time series was set to 96, the article did not provide the reason why such a change was made, and directly set the model inputs to 720 and compare the final prediction performance with other models within the respective paper's input setting parameters.

3) The model's input data and model parameter experiments were conducted with input sequence lengths ranging from 720 to 6480, which is clearly outside the range of parameter settings of previous SOTA models. Such an experimental approach is not representative and is independent of the input sequence lengths used in the main experiment. Input lengths ranging from 96 to 720 are recommended for comparison with mainstream models.

4) For the parameter sensitivity experiments, the article only experiments on input length, and the common batch size (bs), learning rate (lr), and hidden size (H) should also be reported, which is a very basic and necessary experiment.

5) The 336 inputs reported in the experimental appendix do not match the 720 inputs stated in the paper.
There is a discrepancy between the thesis and the code in that the actual early stops in the code are set to 100 rounds, which contradicts the description in the thesis of no more than 30 epochs and 5 rounds of early stops. The experimental design as well as the experimental results are very confusing.

**Questions:**

1) Extension of the Model to Non-Periodic Data Sets:
   - How can the proposed model be adapted or extended to accommodate data sets that exhibit insufficient periodicity in time?

2) Selection of Periodic Interval (P) Value in Periodic Data Sets:
   - In the absence of prior knowledge or conditions, what methodologies can be employed to determine an appropriate periodic interval (P) value for a given periodic data set?

3) Consistency of Results with Varying Input Lengths:
   - Under the experimental setups utilized in other mainstream studies, specifically when the input length is set to 96, would it be feasible to derive results consistent with those presented in this paper?

4) Rationale Behind Experimental Design Choices:
   - What is the basis for designing the experiment in such a way that only 10 models were selected for comparison, with the performance of the top 5 models being evaluated using an input length of 720, while other models adhered to the settings outlined in their respective papers?

These questions aim to clarify the methodological considerations and experimental design choices made in the study, highlighting potential areas for further exploration and adaptation.

---

> ### Author Response · Authors · 2024-11-20
> **Response to Reviewer ZKSJ 1/4**
>
> Dear Reviewer ZKSJ,
>
> Thank you for your constructive feedback on our experimental setup. To further ensure the fairness of the experiments and minimize trivial errors, we have addressed the previously identified issues and have been working diligently to update the experimental results. Specifically, we have corrected the bug related to the `test_loader` where `drop_last=True` is corrected to `drop_last=False` to avoid insufficient testing. Besides, the input length is unified to 720 across all models.  **To enhance the reliability of our findings, we re-ran the baselines under such uniform and fair setting.  For SparseTSF, PatchTST, DLinear, FITS, Time-LLM, Fedformer, and TimesNet, we used their official code repositories. For Autoformer and Informer, we utilized the code provided in the official DLinear repository to run. All scripts and logs for running our model and the baselines are available at [https://anonymous.4open.science/r/TimeBase-fixbug](https://anonymous.4open.science/r/TimeBase-fixbug).** We will continue addressing your concerns based on this update.
>
> Overall, We have tried to extract the key issues from both your proposed questions and weaknesses, and will try our best to address them as thoroughly as possible.
>
> **W1-1 Period Dependence**
>
> The periodicity of data is the primary motivation of TimeBase. Well-defined periodicity enables TimeBase to quickly determine the length of the basis vectors, transforming time-step-level predictions into segment-level predictions, thereby improving prediction efficiency.
> **W1-2 What about time series without clear periodicity?**
>
> Although periodicity is a key motivation for TimeBase, its core functionality lies in efficiently extracting basis representations of sub-series and making predictions at the segment level. In most real-world scenarios, we can determine the periodic length \(P\) based on prior knowledge of the data to segment the time series. However, in extreme cases where there is no clear periodicity or additional prior knowledge, we can still identify suitable sub-segment lengths through frequency domain analysis. This can be explained from two perspectives.
>
> **(1) Analysis from Signal Theory**
>
>   In signal theory, even when data (such as those from chaotic systems) does not exhibit clear periodicity, hidden temporal patterns can still be identified. According to Fourier Transform, any finite-length time series $x(t)$ can be decomposed into a linear combination of sine and cosine basis functions:
>
>   $x(t) = \sum_{k=0}^{N-1} A_k \cos\left(\frac{2\pi kt}{N}\right) + B_k \sin\left(\frac{2\pi kt}{N}\right)$
>
>   Here, $A_k$ and $B_k$ are the spectral coefficients, which represent the amplitude of the different frequency components. In chaotic or complex non-periodic signals, these frequency components may not correspond to explicit periodic cycles. However, low-frequency components often contribute significantly to the signal's energy. TimeBase leverages this property by focusing on segments with higher energy content (i.e., determine the segmentation length by FFT on non-period data), extracting  basis vectors to make accurate and extremely efficient segmentation-level forecasting.
>
>   **(2) Illustrative Experiments**
>
>   Take weather dataset used in our experiment for example. It is with a 10-minute sampling interval, where the full period corresponds to one year (52560 time points). However, TimeBase operates with an input length of 720, which is much smaller than the full 52560-point yearly period. This implies that the input sequence lacks any obvious periodicity. However, using Fourier Transform, we find that some dominant frequency components are much lower, for example, \(p = 4\) corresponding to higher energy. In this case, TimeBase uses segmentation not equal to period to divide the input time series, and effectively build corresponding basis vectors. **As shown in Table 2 of the main text, TimeBase maintains both accurate predictive performance and extremely low resource consumption in datasets with period lengths greater than the input length (in other words, there is periodicity in the input), such as ETTm1, ETTm2, and Weather.**
>
> **Thus, TimeBase is capable of handling a wide range of time series, including those with no clear periodicity.**
>
> **W1-3: Application of FFT and Risk in Data Leak**
>
> As demonstrated in Section 3, FFT is only applied to the training set and does not involve the test or validation sets, thereby **preventing any risk of data leakage.** Additionally, FFT is not used to compute the length of $P$ for each batch. Instead, it is applied only once on the entire training dataset (a long time series) before training to determine $P$. **Therefore, there is no risk of calculating different values for $P$ multiple times.**

---

> ### Author Response · Authors · 2024-11-20
> **Response to Reviewer ZKSJ 2/4**
>
> **W2: Reasons for setting the input length of TimeBase to 720?**
>
> Thanks for your insightful question. Due to TimeBase's extremely small parameter count (0.39K, which is 3000 times smaller than DLinear's 1.04M), and the need to reconstruct the basis period based on respectively longer historical information, we choose a larger input length (720) to enhance the model's fitting performance. A similar choice of input length has been used in FITS [1]. **We have added this explanation to the experimental setup in Section 5.1.**
>
> **W3. Input Length Setup for Baselines**
>
> Thank you for your valuable suggestion. In the original version of our paper, the input lengths for the baselines were set according to the configurations in the respective original papers (e.g., DLinear and PatchTST used 336, and iTransformer used 96). This approach was based on practices from several time series forecasting papers, i.e., FITS [1]. However, after carefully considering the validity of your concern, we recognize that a longer input length maybe allow the model to learn better performance, and it may also lead to differences in the number of test samples. **Therefore, in the interest of fair comparison, we have standardized the input length to 720 across all models.** We also corrected the bug involving `drop_last=True` in the `test_loader`, re-ran all the baselines, and updated the comparative experimental results (Tables 1, 2, 3, and 21). TimeBase continues to demonstrate superior efficiency while maintaining state-of-the-art prediction performance. All relevant codes, scripts, and experimental logs are publicly available at [https://anonymous.4open.science/r/TimeBase-fixbug].
>
> **W4. Adopt Mainstream Input Length in Efficiency Analysis**
>
> In Section 5.3, under the paragraph titled "Efficiency in Ultra-long Look-back Window," we discuss the efficiency of TimeBase under extreme input lengths, ranging from 720 to 6480. Although these are not typical input lengths for mainstream models, we do that for this analysis highlights TimeBase's superlinear l complexity under extreme conditions.
>
> **Regarding your insightful comment about the comparison for mainstream input lengths (from 96 to 720), we actually covered this in Section 5.3 under the paragraph "Main Efficiency Comparison."** As shown in Table 3, in the original version of the paper (PDF), the input lengths were set according to those used in the respective original papers (e.g., 336 for DLinear, 96 for iTransformer). In the revised version (PDF), for fair comparison, we standardized the input length to 720 for all models. In this setup, TimeBase outperforms all other baselines with an inference speed of 0.98ms, 0.39K parameters, 2.77M MACs, 88.89M memory usage, and 20.7s/epoch training time.
>
> Finally, we appreciate your valuable suggestions. **In Appendix K, we have provided a parameter scale comparison of TimeBase with well-known models at various mainstream input lengths (96, 192, 336, 720).** The comparison table demonstrates TimeBase's remarkable parameter efficiency across these different input lengths, ranging from 96 to 720.
>
> **Table. Parameter scale comparison of TimeBase with well-known models at various mainstream input lengths (96, 192, 336, 720) on Electricity. The forecasting length is 720.**
>
> | Input length |  96   |  192  |  336  |  720   |
> | :----------: | :---: | :---: | :---: | :----: |
> |   TimeBase   | **0.24K** | **0.26K** | **0.30K** | **0.39 K** |
> | iTransformer | 5.15M | 5.20M | 5.27M | 5.47M  |
> |   DLinear    | 0.13M | 0.27M | 0.49M | 1.03M  |
> |   PatchTST   | 1.5M  | 2.61M | 4.27M | 8.69M  |
>
> Thanks again for your advice on TimeBase's efficiency analysis.

---

> ### Author Response · Authors · 2024-11-20
> **Response to Reviewer ZKSJ 3/4**
>
> **W5. Sensitivity Experiments under Different Learning Rate and Batch Size**
>
> Thank you for your attention to the sensitivity experiments in the paper. At the beginning, we primarily focused on the intrinsic parameters of the model, such as the basis number and orthogonal weight. To address the reviewer’s concerns, we conducted additional experiments to evaluate the performance of TimeBase under varying learning rates (lr) and batch sizes (bs). Below, we summarize the findings and clarify relevant aspects of the model, including its parameterization. **The corresponding experimental results have been added to Appendix L.**
>
> **(1) Clarification on Hidden Dimensions in TimeBase.**
>
> **Actually, TimeBase does not include a hidden dimension parameter.** However, the number of basis signals (`basis_num`) can be conceptually interpreted as analogous to hidden dimensions. Sensitivity experiments on `basis_num` have already been included in the main text (Figure 4).
>
> **(2) Learning Rate Sensitivity**
>
> Table below demonstrates the performance of TimeBase on the ETTh2 dataset under different learning rates. TimeBase consistently delivers strong performance across a wide range of learning rates, with optimal results achieved at a relatively high learning rate of \(2 \times 10^{-1}\). This characteristic can be attributed to TimeBase's minimal parameter count, which allows it to favor larger learning rates and achieve faster convergence.
>
> **Table. Performance of TimeBase under different learning rate on traffic. The batch size is fixed as 128, input length is 720, forecasting length is 96, basis number is 6, orthogonal weight is 0.04.**
>
> | Learning Rate | 2e-1  | 1e-1  | 9e-2  | 8e-2  | 7e-2  | 6e-2  | 5e-2  | 4e-2  |   3e-2    | 2e-2  | 1e-2  |
> | :-----------: | :---: | :---: | :---: | :---: | :---: | :---: | :---: | :---: | :-------: | :---: | :---: |
> |      MSE      | 0.395 | 0.394 | 0.396 | 0.394 | 0.393 | 0.394 | 0.394 | 0.398 | **0.394** | 0.395 | 0.395 |
> |      MAE      | 0.268 | 0.268 | 0.271 | 0.268 | 0.268 | 0.267 | 0.267 | 0.269 | **0.267** | 0.268 | 0.269 |
>
> **Table. Performance of TimeBase under different learning rate on ETTh2. The batch size is fixed as 512, input length is 720, forecasting length is 96, basis number is 6, orthogonal weight is 0.2.**
>
> | **Learning Rate** | **2e-1**  | **1e-1** | **9e-2** | **8e-2** | **7e-2** | **6e-2** | **5e-2** | **4e-2** | **3e-2** | **2e-2** | **1e-2** |
> | ----------------- | --------- | -------- | -------- | -------- | -------- | -------- | -------- | -------- | -------- | -------- | -------- |
> | **MSE**           | **0.292** | 0.292    | 0.293    | 0.294    | 0.294    | 0.297    | 0.294    | 0.293    | 0.293    | 0.311    | 0.345    |
> | **MAE**           | **0.349** | 0.351    | 0.352    | 0.352    | 0.351    | 0.352    | 0.351    | 0.351    | 0.351    | 0.369    | 0.402    |
>
> **(3) Batch Size Sensitivity**
>
> The influence of batch size was evaluated on both the Traffic and ETTh2 datasets. Results are presented in below Tables. For Traffic, the best performance occurs with a batch size of 128, while ETTh2 benefits from a larger batch size of 512. While batch size adjustments theoretically should not substantially affect performance, the Traffic dataset shows higher sensitivity in batch size of 512. This may be due to the fixed learning rate ($3 \times 10^{-2}$) being relatively small of TimeBase, limiting convergence speed within the 30 training epochs. The variations in performance likely stem from dataset-specific characteristics and the interaction between batch size and fixed learning rates.
>
> **Table. Performance of TimeBase under different batch size  on traffic. The learning rate is fixed as 3e-2, input length is 720, forecasting length is 96, basis number is 6, orthogonal weight is 0.04.**
>
> | **Batch Size** | **512** | **256** | **128**   | **64** |
> | -------------- | ------- | ------- | --------- | ------ |
> | **MSE**        | 0.402   | 0.396   | **0.394** | 0.395  |
> | **MAE**        | 0.279   | 0.273   | **0.267** | 0.268  |
>
> **Table. Performance  of TimeBase under different batch size on ETTh2. The learning rate is fixed as 2e-1, input length is 720, forecasting length is 96, basis number is 6, orthogonal weight is 0.2.**
>
> | **Batch Size** | **512**   | **256** | **128** | **64** |
> | -------------- | --------- | ------- | ------- | ------ |
> | **MSE**        | **0.292** | 0.295   | 0.294   | 0.294  |
> | **MAE**        | **0.349** | 0.352   | 0.351   | 0.353  |

---

> ### Author Response · Authors · 2024-11-20
> **Response to Reviewer ZKSJ 4/4**
>
> **W6. Clarification of Experimental Setup**
>
> **As explained in the main text, the input length for TimeBase is consistently set to 720.** However, due to a typographical error, we mistakenly listed the input length as 336 in Table 7 of Appendix D, which has now been corrected. Additionally, TimeBase was indeed run for 30 epochs with early stopping applied after 5 epochs. We would like to clarify that the parameters in the `run_longExp.py` script are set to default values, not the actual values used in the experiments. To ensure transparency, we have uploaded all the relevant scripts and logs, allowing you to replicate the experiments on your own.
>
> [1] FITS: Modeling Time Series with $10k$ Parameters
>
>
>
> **E1.Additional Discussion**
>
> **Finally, we sincerely thank you for your constructive feedback on TimeBase, which has guided us in meticulously and carefully refining our experimental setup. Your thoughtful suggestions have been invaluable.**
>
> In addition, we would like to engage in an extra discussion with you regarding an intriguing discovery. **We found that TimeBase can serve as an efficient plug-and-play tool for patch-based models to reduce model complexity.** To validate this idea, we integrated TimeBase into PatchTST and conducted comparative experiments. In implementation, after segmenting the time series into patches, TimeBase  can expertly be employed to patch-based methods to extract the basis components.
>
> The experimental results, partially shown in the table below, demonstrate that PatchTST combined with TimeBase can significantly reduce computational resource consumption while maintaining or even improving prediction accuracy. **Detailed experiments have been included in Appendix M, and we look forward to your review.**
>
> **Table. Performance of TimeBase as a Plug-and-Play Component for Patch-Based Methods. The input length is set as 720, and the output length is 96. Full results are available at Appendix M**
>
> |             |       | PatchTST+TB |        |       | PatchTST |        | Complexity  |     Reduction   |
> | ----------- | :---: | :---------: | :----: | :---: | :------: | :----: | :-------: | :----: |
> |             |  MSE  |    MACs     | Params |  MSE  |   MACs   | Params |   MACs    | Params |
> | ETTh1       | 0.364 |   0.77 M    | 0.03 M | 0.377 | 11.05 M  | 0.15 M |  93.00%   | 83.15% |
> | ETTh2       | 0.275 |   1.28 M    | 0.03 M | 0.276 | 11.05 M  | 0.15 M |  88.42%   | 79.08% |
> | ETTm1       | 0.29  |   28.87 M   | 0.52 M | 0.298 | 258.69 M | 1.51 M |  88.84%   | 65.20% |
> | ETTm2       | 0.165 |   57.59 M   | 0.65 M | 0.165 | 258.69 M | 1.51 M |  77.74%   | 57.07% |
> | Weather     | 0.145 |   86.60 M   | 0.52 M | 0.149 | 776.08 M | 1.51 M |  88.84%   | 65.20% |
> | Electricity | 0.128 |   1.32 G    | 0.52 M | 0.141 | 11.86 G  | 1.51 M |  88.84%   | 65.20% |
> | Traffic     | 0.36  |   4.27 G    | 0.55 M | 0.363 | 31.86 G  | 1.51 M |  86.61%   | 63.57% |
>
>
>
> Best Regards,
>
> Author of Paper 323

---

> > ### Comment · Reviewer_ZKSJ · 2024-11-20
> >
> > Your supplementary experiment can solve some of my problems, but it cannot cover up the fatal problems in the first version of the submitted paper (such as experimental settings/printing errors, etc.). I am willing to improve my score.

---

> ### Author Response · Authors · 2024-11-22
> **Thanks for your response**
>
> Dear Reviewer ZKSJ,
>
>
> Thank you for your valuable time in reviewing our response and for acknowledging our supplementary experiments. **We deeply respect your dedication to ensuring fairness in experimental setups. Your observations have inspired us to critically reevaluate and construct a fairer comparison.**
>
> Our pervious strategy of adopting the original model’s input length settings was not entirely fair. **We had followed this strategy because we noticed that it is commonly used in several well-known time series forecasting papers [1,2,3,4,5,6], and we just think it would help avoid discrepancies between our results and those reported in the original papers.**  However, after your insightful feedback and suggestions, we dedicated significant time to re-running the baselines with different input lengths. Through this process, we recognized the importance of standardizing the input length across models to ensure a more rigorous and fair comparison. We have since updated the reported results in the manuscript of baselines to reflect a consistent input length of 720.
>
>
> **To further illustrate the effectiveness of TimeBase under shorter input length**, in addition to the unified input length of 720, we tirelessly conducted experiments using a **unified input length of 336** for forecasting, addressing concerns about potential inconsistencies. The results demonstrate that **TimeBase consistently delivers competitive or superior performance across diverse benchmarks** with the 336 input length. **The full results are provided in Appendix O, and the corresponding scripts and logs have been updated and are available at [https://anonymous.4open.science/r/TimeBase-fixbug](https://anonymous.4open.science/r/TimeBase-fixbug).**
>
> **Table. Forecasting Performance with 336 Input Length and 720 Forecasting Horizon.**
> | Dataset     | TimeBase (MSE) | TimeBase (MAE) | iTransformer (MSE) | iTransformer (MAE) | DLinear (MSE) | DLinear (MAE) | PatchTST (MSE) | PatchTST (MAE) |
> |-------------|:--------------:|:--------------:|:------------------:|:------------------:|:-------------:|:-------------:|:--------------:|:--------------:|
> | **Weather** | **0.319**      | **0.336**      | 0.323              | 0.336              | 0.325         | 0.363         | **0.321**      | **0.335**      |
> | **Electricity** | **0.209** | **0.294**      | 0.212              | 0.305              | 0.220         | 0.321         | **0.210**      | **0.298**      |
> | **ETTh1**   | **0.434**      | **0.439**      | 0.505              | 0.507              | 0.505         | 0.516         | **0.456**      | **0.471**      |
> | **ETTh2**   | **0.397**      | **0.436**      | 0.427              | 0.447              | 0.778         | 0.628         | **0.391**      | **0.429**      |
> | **ETTm1**   | **0.428**      | **0.421**      | 0.439              | 0.436              | 0.428         | 0.424         | **0.419**      | **0.425**      |
> | **ETTm2**   | **0.368**      | **0.383**      | 0.382              | 0.397              | 0.460         | 0.465         | **0.369**      | **0.386**      |
>
>
>
>
> We sincerely appreciate your constructive feedback, which has guided us to conduct more rigorous analyses and expand our experiments to further validate the robustness of TimeBase. **The goal of TimeBase is to construct an exceptionally lightweight forecasting model (0.39K parameter, 2.77M MACs, 0.98ms inferring time on CPU) without compromising predictive performance (Results of Section 5.3 and Appendix O). Moreover, it is designed as a plug-and-play solution （Results of Appendix M） that can significantly reduce the model complexity of patch-based methods (i.e., PatchTST (77.74%~93.00% reduction of MACs).** We are now committed to working together to create a fair comparison environment and contribute to the stable development of the time series community. We would greatly appreciate any additional insights or suggestions you may have to improve the clarity and comprehensiveness of our work.
>
> **Your guidance is truly invaluable to us, and we sincerely hope that the efforts we have made could further address your concerns.**
>
>
> [1] Time-LLM: Time Series Forecasting by Reprogramming Large Language Models, ICLR 2024
>
> [2] FITS: Modeling Time Series with 10k Parameters, ICLR 2024
>
> [3] SparseTSF: Modeling Long-term Time Series Forecasting with 1k Parameters, ICML 2024
>
> [4] TTMs: Fast Multi-level Tiny Time Mixers for Improved Zero-shot and Few-shot Forecasting of Multivariate Time Series, NeurIPS 2024.
>
> [5] Unified Training of Universal Time Series Forecasting Transformers, ICML 2024
>
> [6] Timer: Generative Pre-trained Transformers Are Large Time Series Models, ICML 2024
>
>
>
> Kind regards,
> Authors of Paper 323

---

> ### Author Response · Authors · 2024-11-24
> **Thanks again for Reviewer ZKSJ**
>
> Dear Reviewer ZKSJ,
>
> **To further validate the predictive capabilities of TimeBase, we continued to conduct experiments with a unified input length of 96 steps, in addition to the previously input lengths of 336 and 720.** Despite working tirelessly to ensure consistency across all settings, we observed that the performance of most methods, including Transformer-based approaches like iTransformer and PatchTST, as well as decomposition-based methods like TimesNet, declined to varying degrees. This suggests that shorter input lengths may limit the ability of these models to capture long-term dependencies and global features.  Notably, TimesNet, leveraging its unique time-frequency decomposition mechanism, demonstrated a certain level of resilience in shorter input lengths. Meanwhile, TimeBase, with its minimalist parameterized design, maintained relatively stable performance under these challenging conditions.
>
> **The full results are provided in Appendix P, and the corresponding scripts and logs have been updated and are available at [https://anonymous.4open.science/r/TimeBase-fixbug](https://anonymous.4open.science/r/TimeBase-fixbug).**
>
> ---
>
> **Table: Forecasting Performance with 96 Input Length and 720 Forecasting Horizon**
>
> | Dataset      | **TimeBase (MSE)** | **TimeBase (MAE)** | **iTransformer (MSE)** | **iTransformer (MAE)** | **DLinear (MSE)** | **DLinear (MAE)** | **PatchTST (MSE)** | **PatchTST (MAE)** | **TimesNet (MSE)** | **TimesNet (MAE)** |
> |--------------|:------------------:|:------------------:|:----------------------:|:----------------------:|:----------------:|:----------------:|:------------------:|:------------------:|:------------------:|:------------------:|
> | **Weather**     | **0.351**          | **0.347**          | 0.363                  | 0.354                  | 0.349            | 0.385            | **0.352**          | **0.346**          | 0.359              | 0.355              |
> | **Electricity** | 0.247              | **0.321**          | **0.243**              | 0.324                  | 0.272            | 0.363            | **0.233**          | **0.313**          | 0.298              | 0.370              |
> | **ETTh1**       | **0.467**          | **0.461**          | 0.516                  | 0.501                  | 0.543            | 0.530            | **0.488**          | **0.476**          | 0.516              | 0.494              |
> | **ETTh2**       | **0.427**          | **0.445**          | 0.429                  | 0.446                  | 0.862            | 0.671            | **0.423**          | **0.442**          | 0.439              | 0.451              |
> | **ETTm1**       | 0.482              | **0.448**          | 0.487                  | 0.457                  | **0.475**        | 0.452            | **0.456**          | **0.440**          | 0.482              | 0.460              |
> | **ETTm2**       | **0.406**          | **0.397**          | 0.418                  | **0.409**              | 0.560            | 0.526            | 0.410              | **0.404**          | 0.423              | 0.407              |
>
> ---
>
> We deeply appreciate your thoughtful and constructive feedback, which has been invaluable in improving the robustness of our experiments and the presentation of our findings. Your comments have motivated us to explore additional experimental setups, enabling a more comprehensive evaluation of TimeBase. **If you have further insights or suggestions, we would be truly grateful to hear them, as your guidance is instrumental in refining our work.**
>
> Sincerely,
> Author of Paper 323.

---

> ### Author Response · Authors · 2024-11-29
> **Contributions of TimeBase**
>
> Dear All Reviewers and Area Chair,
>
> Thanks to **all the reviewers for raising your scores and acknowledging our improvements**, which is a tremendous encouragement to us. We believe TimeBase offers significant advantages (**extreme efficiency, plug-and-play usability, reproducibility**) and can positively contribute to the time series community. Therefore, we would like to **briefly highlight** its key strengths and potential impact, and **kindly ask for your final judgment on TimeBase**.
>
> ---
>
>
> **1. Competitive Predictive Performance and Extreme Lightness**
>
> TimeBase not only demonstrates strong predictive performance (under unified input lengths of **96**, **336**, or **720**, full results in Appendix N, O, and P), but it also boasts an exceptionally lightweight complexity (Section 5.3): TimeBase have **0.39K parameters, 2.77M MACs computation, 88.89M memory usage, and 0.98ms CPU inference speed**. Take MACs for example, its MACs is **1/120 of DLinear**, **1/646 of iTransformer**, and **1/5115 of PatchTST**. The effective utilization of time series data makes it suitable for complex forecasting scenarios. Whether in cloud-based data centers or resource-constrained edge devices, **TimeBase can deliver accurate forecasting results with extremely low resource consumption**.
>
> **2. Plug-and-Play Usability**
>
> TimeBase can not only function as **a time-series forecaster** but also serve as **an efficient complexity reducer** for patch-based methods (Appendix M). In implementation, TimeBase can be expertly used to extract key basis components after patching the entire time series of patch-based models. Experimental results show that TimeBase reduces the computation of PatchTST to just **1/9** of its original MACs.**(e.g., from 14.17G to 1.58G)**, while improving prediction accuracy in **21** of the total **28 forecasting settings**. The remaining 7 settings show only trivial differences in metrics. **This demonstrates TimeBase's ability to significantly reduce the resource consumption of any patch-based model without sacrificing performance.**
>
>
>
> **3. Inspiration for Future Model Design**
>
> TimeBase reveals that long time series often exhibit temporal low-rank characteristics, and **it is the first to propose improving data and computational resource utilization by extracting basis components.** It also provides a **theoretical explanation** of how this approach is **applicable to any time series** (Appendix B.3). This **provides a potential strategy for designing efficient models and could offer valuable insights for the backbone design of large pre-trained time-series models** in the future works.
>
>
> ---
>
> **We sincerely hope that TimeBase will be recognized by you** and contribute to the development of the time series community. If you still believe TimeBase does not meet ICLR's acceptance standards, we would greatly appreciate your honest feedback to help us evaluate TimeBase from multiple perspectives. **Finally, we fully respect your scoring decision.** Thank you very much!
>
> Best Regards,
> Author of Paper 323

---

### Official Review · Reviewer_vUMr · 2024-11-01

**Soundness:** 2
**Presentation:** 3
**Contribution:** 2
**Rating:** 5
**Confidence:** 5

**Summary:**

The authors propose a very small and efficient model for long-term time-series forecasting by utilizing the periodicity in many real-world time series. They show that their minimal model (TimeBase) outperforms many competing models.

**Strengths:**

Long-term time-series forecasting is an important problem and how to solve it efficiently has many real-world implications.

**Weaknesses:**

I am not impressed by the theoretical results in the paper. The theorems and lemmas seem quite obvious and can be easilly conveyed with plain English.

**Questions:**

1. What about time series without a clear periodicity (e.g., chaotic systems)?
2. I would love to see tests on some synthetic data, for example, time series with both fast and slow oscillations (e.g., |cos(t)|sin(100t)).
3. For Fig. 5, what about R<6?

---

> ### Author Response · Authors · 2024-11-20
> **Response to Reviewer vUMr 1/3**
>
> Dear Reviewer vUMr,
>
> Thank you for your valuable suggestions on improving the theoretical aspects of TimeBase and for your insights regarding its ability on non-period data. We will carefully address your concerns in detail.
>
> **W1. Modification on theory analysis**
>
> Thank you for your valuable suggestions to improve our theoretical analysis. Actually, the purpose of Section 4 is to analyze the feasibility of TimeBase. To make the presentation more rigorous, **we have re-organized and simplified the original effectiveness analysis. Additionally, to demonstrate the generalizability of TimeBase, we have included an more theoretical analysis of its convergence on more generic time series data in Section 4** The findings indicate that when the learned period bases are orthogonal, TimeBase achieves a smaller upper bound on the error. We look forward to your re-evaluation of Section 4 in the revised PDF.
>
> **Q1. What about time series without clear periodicity?**
>
> Thank you for your question! **TimeBase still demonstrates strong applicability and predictive power when dealing with time series that lack clear periodicity.**
>
> The core mechanism of TimeBase involves dividing the time series into multiple time segments, extracting key temporal basis vectors, and then using these vectors to reconstruct future time segments for efficient segment-level forecasting. **While periodicity serves as the motivation for TimeBase, it does not restrict its ability to handle non-periodic or complex signals.** It can be explained in two aspects:
>
> **(1) Analysis from Signal Theory**
>
> In signal theory, even when data (such as those from chaotic systems) does not exhibit clear periodicity, hidden temporal patterns can still be identified. According to Fourier Transform, any finite-length time series $x(t)$ can be decomposed into a linear combination of sine and cosine basis functions:
>
> $x(t) = \sum_{k=0}^{N-1} A_k \cos\left(\frac{2\pi kt}{N}\right) + B_k \sin\left(\frac{2\pi kt}{N}\right)$
>
> Here, $A_k$ and $B_k$ are the spectral coefficients, which represent the amplitude of the different frequency components. In chaotic or complex non-periodic signals, these frequency components may not correspond to explicit periodic cycles. However, low-frequency components often contribute significantly to the signal's energy. TimeBase leverages this property by focusing on segments with higher energy content (i.e., determine the segmentation length by FFT on non-period data), extracting  basis vectors to make accurate and extremely efficient segmentation-level forecasting.
>
> **(2) Illustrative Experiments**
>
> Take weather dataset used in our experiment for example. It is with a 10-minute sampling interval, where the full period corresponds to one year (52560 time points). However, TimeBase operates with an input length of 720, which is much smaller than the full 52560-point yearly period. This implies that the input sequence lacks any obvious periodicity. However, using Fourier Transform, we find that some dominant frequency components are much lower, for example, \(p = 4\) corresponding to higher energy. In this case, TimeBase uses segmentation not equal to period to divide the input time series, and effectively build corresponding basis vectors. **As shown in Table 2 of the main text, TimeBase maintains both accurate predictive performance and extremely low resource consumption in datasets with period lengths greater than the input length (in other words, there is periodicity in the input), such as ETTm1, ETTm2, and Weather.**
>
> **Thus, TimeBase is capable of handling a wide range of time series, including those with no clear periodicity, by capturing essential signal components and leveraging its flexible segmentation approach.**

---

> ### Author Response · Authors · 2024-11-20
> **Response to Reviewer vUMr 2/3**
>
> **Q2. Test on some synthetic data.**
>
> Thank you for your insightful feedback. **We have provided a detailed analysis of TimeBase's performance on a challenging synthetic dataset, derived from the function $Y = |\cos(X)|\sin(100X)$ and add the more detailed  results  in Appendix J.**
>
> This dataset captures intricate oscillations and serves as a benchmark to compare TimeBase against state-of-the-art models like DLinear and PatchTST.  The synthetic dataset from $\pi/2$ to $3\pi/2$ comprises 5000 samples (data are within one period to avoid data leak), which are split into training (60%), validation (20%), and testing (20%) sets. TimeBase consistently outperformed the competing methods in both accuracy and efficiency across forecasting lengths of L=100, L=200, and L=300, as shown in Tables below. For instance, at a forecasting length of 100, TimeBase achieved the lowest MSE (0.007) and MAE (0.070) with only 0.19K parameters and 0.077M MACs, significantly outperforming PatchTST's computational overhead (1.28M MACs, 219K parameters). As the forecasting length increased, TimeBase maintained robust accuracy and efficiency, demonstrating its adaptability to varying time-series dynamics. **It shows that TimeBase maintain its efficiency advantage and adequate forecasting performance in datasets with both fast and slow oscillations**
>
> **Table Performance of TimeBase on synthetic data for a forecasting length of 100,200 and 300.**
>
>
> | $L=100$  |  MSE  |  MAE  |  MACs  | Params |
> | :------: | :---: | :---: | :----: | :----: |
> | TimeBase | 0.007 | 0.070 | 0.077M | 0.19K  |
> | DLinear  | 0.015 | 0.081 |  0.1M  |  100K  |
> | PatchTST | 0.011 | 0.085 | 1.28M  |  219K  |
>
>
> | $L=200$  |  MSE  |  MAE  |  MACs  | Params |
> | :------: | :---: | :---: | :----: | :----: |
> | TimeBase | 0.010 | 0.082 | 0.093M | 0.23K  |
> | DLinear  | 0.021 | 0.124 |  0.2M  |  200K  |
> | PatchTST | 0.017 | 0.083 | 1.48M  |  420K  |
>
>
> | $L=300$  |  MSE  |  MAE  |  MACs  | Params |
> | :------: | :---: | :---: | :----: | :----: |
> | TimeBase | 0.013 | 0.094 | 0.099M | 0.26K  |
> | DLinear  | 0.039 | 0.180 |  0.3M  |  300K  |
> | PatchTST | 0.023 | 0.136 | 1.69M  |  622K  |
>
> We hope this clarifies the efficacy of our approach. Thank you again for your valuable comments.
>
> **Q3. What about R<6 in Fig. 5**
>
> Thank you for your insightful question. **We have updated Figure 5 to expand the range of \(R\) from $[6, 12, 18, 24, 30]$ to $[2, 4, 6, 12, 18, 24, 30]$.** Here, we present the part results of different $R$ values for ETTh1 when predicting with a step length of 720 in the table below. From the updated figure, it can be observed that when $R < 6$, the predictive performance of TimeBase exhibits significant fluctuations. However, as $R$ surpasses 6, the performance gradually improves. **This indicates that larger $R$ values tend to enhance predictive accuracy, but due to the inherent predictability limits of the dataset, diminishing returns or marginal gains become evident beyond a certain threshold.**  Therefore, selecting an appropriate \(R\) is crucial to strike a balance between predictive performance and computational efficiency, tailored to the specific requirements of different scenarios. At present, the choice of \(R\) is based on empirical insights derived from sensitivity experiments. In future work, we plan to explore a theoretical framework for determining the optimal $R$, enabling a more rigorous balance between predictive performance and computational efficiency. **We sincerely appreciate your question, as it prompts a deeper reflection on parameter selection.**
>
> **Table. Performance of ETTh1 under $R\in[2, 4, 6, 12, 18, 24, 30]$. The forecasting length is 720.**
>
> | R    | 2     | 4     | 6     | 12    | 18    | 24    | 30    |
> | ---- | ----- | ----- | ----- | ----- | ----- | ----- | ----- |
> | MSE  | 0.462 | 0.461 | 0.439 | 0.438 | 0.437 | 0.437 | 0.436 |

---

> ### Author Response · Authors · 2024-11-20
> **Response to Reviewer vUMr 3/3**
>
> **E1 Additional Discussion**
>
> **Finally, we sincerely thank you for your constructive feedback on TimeBase, which has guided us to focus on explaining its generalization on non-period data. Your thoughtful suggestions have been invaluable.**
>
> **In addition, we would like to engage in an extra discussion with you regarding an intriguing discovery. We found that TimeBase can serve as a plug-and-play tool for patch-based models to extremely reduce model complexity.** To validate this idea, we integrated TimeBase into PatchTST and conducted comparative experiments. In implementation, after segmenting the time series into patches, TimeBase  can expertly be employed to patch-based methods to extract the basis components.  The experimental results, partially shown in the table below, demonstrate that  **TimeBase can significantly reduce computational resource consumption of Patch-based methods (i.e., PatchTST (77.74%~93.00% reduction of MACs) )while maintaining or even improving prediction accuracy.** **Detailed experiments have been included in Appendix M, and we look forward to your review.**
>
> **Table. Performance of TimeBase as a Plug-and-Play Component for Patch-Based Methods. The input length is set as 720, and the output length is 96.**
>
> |             |       | PatchTST+TB |        |       |  PatchTST   |        | Complexity | Reduction       |
> | ----------- | :---: | :------: | :----: | :---: | :------: | :----: | :-------: | :----: |
> |             |  MSE  |   MACs   | Params |  MSE  |   MACs   | Params |   MACs    | Params |
> | ETTh1       | 0.364 |  0.77 M  | 0.03 M | 0.377 | 11.05 M  | 0.15 M |  93.00%   | 83.15% |
> | ETTh2       | 0.275 |  1.28 M  | 0.03 M | 0.276 | 11.05 M  | 0.15 M |  88.42%   | 79.08% |
> | ETTm1       | 0.29  | 28.87 M  | 0.52 M | 0.298 | 258.69 M | 1.51 M |  88.84%   | 65.20% |
> | ETTm2       | 0.165 | 57.59 M  | 0.65 M | 0.165 | 258.69 M | 1.51 M |  77.74%   | 57.07% |
> | Weather     | 0.145 | 86.60 M  | 0.52 M | 0.149 | 776.08 M | 1.51 M |  88.84%   | 65.20% |
> | Electricity | 0.128 |  1.32 G  | 0.52 M | 0.141 | 11.86 G  | 1.51 M |  88.84%   | 65.20% |
> | Traffic     | 0.36  |  4.27 G  | 0.55 M | 0.363 | 31.86 G  | 1.51 M |  86.61%   | 63.57% |
>
>
>
> Best Regards,
>
> Author of Paper 323

---

> ### Author Response · Authors · 2024-11-24
> **Thanks again for Reviewer vUMr**
>
> Dear Reviewer vUMr,
>
> Thank you very much for devoting time on reviewing our manuscript. **As the author-reviewer discussion process is ending soon, we are kindly wondering whether our responses have well addressed your concerns.** If you have any further questions regarding our manuscript, please let us know and we are glad here to provide further discussion and clarification to improve the quality of this manuscript. Thanks again for your valuable time.
>
> Best Regards,
>
> Author of Paper 323

---

> > ### Comment · Reviewer_vUMr · 2024-11-25
> >
> > Thank you for the new experiments. I am willing to raise the score to reflect this improvement.

---

> > > ### Author Response · Authors · 2024-11-25
> > > **Thanks for your feedback!**
> > >
> > > Dear Reviewer vUMr,
> > >
> > > We deeply appreciate the valuable feedback you previously provided and your response to the improvements we made, which has greatly helped us enhance our work. If you think there are still some contents that could be  refined, we would be delighted to continue discussing with you to further improve the quality of TimeBase. Once again, we deeply appreciate your time and thoughtful insights contributed to the ICLR committee and the time series community.
> > >
> > > Yours Sincerely,
> > >
> > > Author of Paper 323

---

> ### Author Response · Authors · 2024-11-29
> **Contributions of TimeBase**
>
> Dear All Reviewers and Area Chair,
>
> Thanks to **all the reviewers for raising your scores and acknowledging our improvements**, which is a tremendous encouragement to us. We believe TimeBase offers significant advantages (**extreme efficiency, plug-and-play usability, reproducibility**) and can positively contribute to the time series community. Therefore, we would like to **briefly highlight** its key strengths and potential impact, and **kindly ask for your final judgment on TimeBase**.
>
> ---
>
>
> **1. Competitive Predictive Performance and Extreme Lightness**
>
> TimeBase not only demonstrates strong predictive performance (under unified input lengths of **96**, **336**, or **720**, full results in Appendix N, O, and P), but it also boasts an exceptionally lightweight complexity (Section 5.3): TimeBase have **0.39K parameters, 2.77M MACs computation, 88.89M memory usage, and 0.98ms CPU inference speed**. Take MACs for example, its MACs is **1/120 of DLinear**, **1/646 of iTransformer**, and **1/5115 of PatchTST**. The effective utilization of time series data makes it suitable for complex forecasting scenarios. Whether in cloud-based data centers or resource-constrained edge devices, **TimeBase can deliver accurate forecasting results with extremely low resource consumption**.
>
> **2. Plug-and-Play Usability**
>
> TimeBase can not only function as **a time-series forecaster** but also serve as **an efficient complexity reducer** for patch-based methods (Appendix M). In implementation, TimeBase can be expertly used to extract key basis components after patching the entire time series of patch-based models. Experimental results show that TimeBase reduces the computation of PatchTST to just **1/9** of its original MACs.**(e.g., from 14.17G to 1.58G)**, while improving prediction accuracy in **21** of the total **28 forecasting settings**. The remaining 7 settings show only trivial differences in metrics. **This demonstrates TimeBase's ability to significantly reduce the resource consumption of any patch-based model without sacrificing performance.**
>
>
>
> **3. Inspiration for Future Model Design**
>
> TimeBase reveals that long time series often exhibit temporal low-rank characteristics, and **it is the first to propose improving data and computational resource utilization by extracting basis components.** It also provides a **theoretical explanation** of how this approach is **applicable to any time series** (Appendix B.3). This **provides a potential strategy for designing efficient models and could offer valuable insights for the backbone design of large pre-trained time-series models** in the future works.
>
>
> ---
>
> **We sincerely hope that TimeBase will be recognized by you** and contribute to the development of the time series community. If you still believe TimeBase does not meet ICLR's acceptance standards, we would greatly appreciate your honest feedback to help us evaluate TimeBase from multiple perspectives. **Finally, we fully respect your scoring decision.** Thank you very much!
>
> Best Regards,
> Author of Paper 323

---

### Official Review · Reviewer_tE5i · 2024-11-03

**Soundness:** 2
**Presentation:** 3
**Contribution:** 2
**Rating:** 5
**Confidence:** 4

**Summary:**

This manuscript introduces a lightweight framework designed to generate predictions for MTS data at the segment level using just two linear layers. These segments are composed of specific time steps that represent periods within the MTS data.

**Strengths:**

This manuscript focuses on improving efficiency in MTS forecasting. The proposed framework comprises only two linear layers, yet it achieves superior efficiency compared to recent state-of-the-art methods. The manuscript provides extensive theoretical analysis to demonstrate the effectiveness of the proposed method.

**Weaknesses:**

Efficiency improvement is crucial for TimeBase; it requires users to have a solid estimate of the dataset's seasonalities. If these are not known, an FFT is utilized to determine them. Please include the computational cost of the FFT in the efficiency analysis, or provide an explanation for its omission. Including this information would offer a more comprehensive understanding of the model's overall efficiency.

Figure 7 indicates that the "Basis Orthogonal Restriction" has a minimal impact on accuracy as measured by MSE. Are there any benefits not captured by this metric? Including an analysis that discusses the potential advantages of "Basis Orthogonal Restriction" could enhance the contributions of TimeBase.

The accuracy of TimeBase, while adequate, is modest compared to state-of-the-art methods. TimeBase's primary contribution lies in its efficiency. However, according to Table 3, improvements in memory usage and epoch time are modest compared to those achieved by Dlinear and iTransformer. Please provide a detailed analysis discussing the advantages of TimeBase. Additionally, to ensure a fair comparison, could you set the look-back window length across all methods the same?

**Questions:**

(a) Please consider improve the quality of some figures.

(b) Could you explain the effectiveness of orthogonal constraint? Figure 7 shows the performance improvements are trivial for all horizons of four datasets

(c) Could you include Dlinear in Figure 4?

(d) Section 3 describes the workflow for univariate time series data. Including an explanation of how to apply TimeBase to MTS data would clarify and enhance the workflow description.

(e) Does TimeBase support multiple seasonality? Such as Traffic could show daily and weekly periods.

---

> ### Author Response · Authors · 2024-11-20
> **Response to Reviewer tE5i 1/5**
>
> Dear Reviewer tE5i,
>
> Thanks for your insightful question for improving the quality of our manuscript. We have provided more experiment details, more fair experiments and discussions to dispel your concerns and tried our best efforts to satisfy the high standards of ICLR community. The responses can be found as below.
>
> **W1. Efficiency  Analysis of using FFT in TimeBase**
> Thanks for your constructive feedback on the practical efficiency of TimeBase.
>
> 1. **Dependence on Periodicity:**
>    The periodicity of data is the primary motivation of TimeBase. Well-defined periodicity enables TimeBase to quickly determine the length of the basis vectors, transforming time-step-level predictions into segment-level predictions, thereby improving prediction efficiency.
>
> 2. **Efficiency of FFT:**
>    FFT (Fast Fourier Transform) is an efficient algorithm for computing the frequency-domain representation of a time series, allowing the discovery of latent periodicities. Its computational complexity is $O(N \log N)$, where $N$ represents the length of the time series. In our experiments, conducted on a machine equipped with an Intel Xeon E5-2609 v4 CPU (16 cores, 2 sockets, 1.7 GHz base frequency), the execution time for FFT on the entire training part of dataset (Electricity dataset) averaged approximately **0.08 seconds** across three runs.
>
> 3. **Why FFT Efficiency Was Not Included in Efficiency analysis:**
>    As you observed, the efficiency analysis in Table 3 does not account for FFT runtime. This omission is for two reasons:
>    - **(1)**  For datasets lacking well-known prior knowledge, **the FFT operation (0.08s) is only used once on the whole training dataset  (a long time series) to determine the  basis period length,  and will not be repeatedly commutated during training  or inferring.**
>    - **(2)** More practically, in actual applications (including the datasets in our experiments), sufficient periodicity information is typically available, making FFT not used for estimating periodicity. **It serves primarily as a fall-back strategy for handling scenarios where the data's periodicity is entirely unknown.**
>
> Finally, thank you again for your constructive suggestions regarding the efficiency analysis of TimeBase. **We have added an explanation in Section 5.3 to clarify why FFT runtime is not included in the analysis.**

---

> ### Author Response · Authors · 2024-11-20
> **Response to Reviewer tE5i 2/5**
>
> **W2. and Q(b). Effective Analysis of Basis Orthogonal Restriction.**
>
> Thank you for your interest in Basis Orthogonal Restriction. Overall, the  benefits  of the  Basis Orthogonal Restriction can be explained from three aspects: (1) Intuitive motivation, (2)Theoretical error bound, and (3) Experimental results.
>
> **(1) Intuitive Motivation**
>
> From the perspective of the data space, **the orthogonality of the basis vectors enhances its representation power, providing them ability to express as any vector in the data space  through linear combination. Therefore, the period basis should also be diverse and distinct, preventing the extraction of very single time-series patterns.** Based on this, we apply the Basis Orthogonal Restriction. In the time-series space, each time series period (segmentation) can be viewed as a composition of several orthogonal period (segmentation) basis vectors. The goal of TimeBase is to transform the long-term forecasting task at the time-step level into a task of basis extraction and prediction at the period level, thus enabling an efficient time-series forecasting model (0.39K, 1000 times smaller than MLP/Mixer-based LTSF models).
>
> **(2) Theoretical Error Bound**
>
> Given a historical time series $\mathbf{X}_\text{his} \in \mathbb{R}^{N \times P}$, its corresponding learned basis vectors $\mathbf{E}\in \mathbb{R}^{R \times P}$,
>
> and future time series $\mathbf{X}_\text{pred} \in \mathbb{R}^{N' \times P}$, the upper fitting error of the model can be expressed as:
>
> $||\mathbf{r}|| \leq \frac{1}{\lambda_{\min}(\mathbf{E} \mathbf{E}^T)}||\mathbf{X}_{\text{pred}}||$
>
> where $\lambda_{\min}(\mathbf{E} \mathbf{E}^T)$ denotes the smallest eigenvalue of the Gram matrix $\mathbf{E} \mathbf{E}^T$. This result highlights that the generalization capability of TimeBase to arbitrary time series relies on learning a well-represented basis matrix $\mathbf{E}$. If $\mathbf{E}$ exhibits a favorable eigenvalue distribution (i.e., $\lambda_{\min}(\mathbf{E} \mathbf{E}^T)$ is large), the upper bound on prediction error is lower, **highlighting the importance of   a high-quality basis vector space and the necessity of orthogonal constraint.** **This analysis hae been added to section 4  and he detailed derivation   is available in Appendix B.3 of revised PDF.**
>
> **(3) Experimental Results**
>
> In the original paper, due to a bug in the original code's test loader with the `drop_last=True`, causing the model not to be evaluated on the full set of test samples, especially with a large batch size (512). We have since fixed the bug in the code and report more reliable results in revised PDF. Fixed code and running script is available at [https://anonymous.4open.science/r/TimeBase-fixbug](https://anonymous.4open.science/r/TimeBase-fixbug).
>
> |                       | ETTh1 \| 96 | ETTh1 \| 192 | ETTh1 \| 336 | ETTh1 \| 720 | ETTh2 \| 96 | ETTh2 \| 192 | ETTh2 \| 336 | ETTh2 \| 720 |
> | :-------------------: | :---------: | :----------: | :----------: | :----------: | :---------: | :----------: | :----------: | :----------: |
> |  Without Constraint   |    0.371    |    0.423     |    0.444     |    0.461     |    0.305    |    0.352     |    0.380     |    0.412     |
> | Orthogonal Constraint |    0.349    |    0.387     |    0.408     |    0.439     |    0.292    |    0.342     |    0.358     |    0.400     |
> |        Improve        |   +0.022    |    +0.036    |    +0.036    |    +0.022    |   +0.013    |    +0.010    |    +0.022    |    +0.012    |
>
> As shown in the table, **after applying the Basis Orthogonal Restriction, the MSE for etth1 and etth2 improved by an average of 7.36% and 5.13%,** respectively. This confirms that the Orthogonal Constraint significantly improves TimeBase’s representation power, maximizing the forecasting potential of its extremely small parameter size.
>
> Thanks again for your insightful advice. To provide readers with a more intuitive understanding, **we have included additional explanations  of the motivation behind the Orthogonal Constraint in  method designing (Section 3), more theoretical analysis (Section 4 and Appendix B.3),  more reliable experiments (Section 5).**

---

> ### Author Response · Authors · 2024-11-20
> **Response to Reviewer tE5i 3/5**
>
> **W3. Detailed Analysis of TimeBase's Advantages and Unified Look-Back Window Length**
>
> Thank you for your insightful discussion on TimeBase's advantages and your constructive suggestions regarding the experimental setup, and we will address your concerns point to point as follows:
>
> **W3-1. Unified Experimental Setup and More Results in Table 3:**
>
> (1) To ensure a fairer comparison, we have refined the experimental setup. **Previously, we follow the comparison approach used in FITS [1], and directly used the official code of each model. Now, we have standardized the look-back window length for all models to 720 to enable a fair comparison of their efficiency. All relevant codes, scripts, and experimental logs have been made publicly available at [https://anonymous.4open.science/r/TimeBase-fixbug]. Results in Table 3 is corresponding updated**
>
> (2) Besides, to provide a more comprehensive evaluation of the models' practicality, **we have added an analysis of inference time on CPU in Table 3 of revised paper PDF.** This additional metric measures the average inference time for processing 100 samples for each model, better reflecting their performance in real-world deployment scenarios.
>
> **Table. Efficient Analysis under Unified Input Length**
>
> |    Model     |    MSE    |  Params   |   MACs    |  Max Mem.  | Epoch Time (GPU) | Infer Time (CPU) |
> | :----------: | :-------: | :-------: | :-------: | :--------: | :--------------: | :--------------: |
> | **TimeBase** | **0.208** | **0.39K** | **2.77M** | **88.89M** |    **20.6s**     |    **0.98ms**    |
> | iTransformer |   0.204   |   5.47M   |   1.79G   |  828.32M   |      65.62s      |     30.41ms      |
> |   DLinear    |   0.209   |   1.04M   |  333.04M  |  158.21M   |      41.08s      |      3.25ms      |

---

> ### Author Response · Authors · 2024-11-20
> **Response to Reviewer tE5i 4/5**
>
> **W3-2. Key Advantages of TimeBase:**
>
> The core contributions of TimeBase are reflected in the following aspects:
>
> 1. **Competitive Predictive Performance:**
>    TimeBase delivers predictive accuracy that is comparable to, and in some cases exceeds, certain SOTA models. **And it can even achieve Top 1 forecasting accuracy in some dataset settings.**
>
> 2. **Extreme Efficiency:**
>
>    - **Parameter Efficiency:** TimeBase requires just **0.39K** parameters—a dramatic reduction compared to other models. Specifically, it achieves approximately **62% fewer parameters** than DLinear and is an impressive **15 times** more lightweight than iTransformer.
>    - **Computational Efficiency (MACs):** TimeBase achieves a MACs count of just **2.77M**, reducing computation **by over 99% and 99.8%** compared to DLinear and iTransformer, respectively.
>    - **Memory Usage (Max Mem.):** TimeBase consumes only **88.89M** of memory, nearly **90%** less than iTransformer, making it ideal for deployment in memory-constrained environments.
>    - **Training and Inference Time:** On a GPU, TimeBase achieves an epoch time of just **20.6** seconds, approximately **1/3** of iTransformer's time. On a CPU, its inference time is only **0.98ms**, **1/3** that of DLinear and **1/30** that of iTransformer, highlighting its remarkable deployment efficiency.
>
> 3. **A Plug-and-play Complexity Reduction Tool for Patch-based Models:**
>
>    **We found that TimeBase can serve as a plug-and-play tool for patch-based models to extremely reduce model complexity and even further enhancing prediction accuracy.** To validate this idea, we integrated TimeBase into PatchTST and conducted comparative experiments. In implementation, after segmenting the time series into patches, TimeBase  can expertly be employed to patch-based methods to extract the basis components.  The experimental results, partially shown in the table below, demonstrate that  **TimeBase can significantly reduce computational resource consumption of PatchTST (77.74%~93.00%) while maintaining or even improving prediction accuracy.** **Detailed experiments have been included in Appendix M, and we look forward to your review.**
>
>    **Table. Performance of TimeBase as a Plug-and-Play Component for PatchTST. The input length is set as 720, and the output length is 96.**
>
>    |             |       | PatchTST+TB |        |       | PatchTST |        | Complexity| Reduction       |
>    | ----------- | :---: | :---------: | :----: | :---: | :------: | :----: | :-------: | :----: |
>    |             |  MSE  |    MACs     | Params |  MSE  |   MACs   | Params |   MACs    | Params |
>    | ETTh1       | 0.364 |   0.77 M    | 0.03 M | 0.377 | 11.05 M  | 0.15 M |  93.00%   | 83.15% |
>    | ETTh2       | 0.275 |   1.28 M    | 0.03 M | 0.276 | 11.05 M  | 0.15 M |  88.42%   | 79.08% |
>    | ETTm1       | 0.29  |   28.87 M   | 0.52 M | 0.298 | 258.69 M | 1.51 M |  88.84%   | 65.20% |
>    | ETTm2       | 0.165 |   57.59 M   | 0.65 M | 0.165 | 258.69 M | 1.51 M |  77.74%   | 57.07% |
>    | Weather     | 0.145 |   86.60 M   | 0.52 M | 0.149 | 776.08 M | 1.51 M |  88.84%   | 65.20% |
>    | Electricity | 0.128 |   1.32 G    | 0.52 M | 0.141 | 11.86 G  | 1.51 M |  88.84%   | 65.20% |
>    | Traffic     | 0.36  |   4.27 G    | 0.55 M | 0.363 | 31.86 G  | 1.51 M |  86.61%   | 63.57% |
>
> 4. **Suitability for Long-Term Time Series Tasks:**
>    The design of TimeBase is rooted in minimalism, focusing on learning fundamental patterns through simple linear transformations while avoiding the complexity  of stacked networks. This approach not only reduces computational overhead but also retains robust modeling capabilities for long-term time series characteristics.
>
> 5. **Deployment Friendliness:**
>    The lightweight nature of TimeBase makes it highly suitable for resource-constrained scenarios, such as edge devices or low-power environments, without requiring extensive computational resources.
>
> Once again, thank you for your valuable feedback, which has helped us enhance the clarity and comprehensiveness of the TimeBase paper. **We hope this analysis provides deeper insights into the unique advantages of TimeBase.**

---

> ### Author Response · Authors · 2024-11-20
> **Response to Reviewer tE5i 5/5**
>
> **Q(a). Improvements to the Quality of Figures**
>
> Thank you for your constructive suggestions to enhance the quality of our paper. We have made the following improvements to the figures:
>
> 1. **Increased DPI for Higher Resolution:** Figures 4, 5, 6, and 7 have been saved with a higher DPI to ensure better visual clarity.
> 2. **Enhanced Color Scheme:** The color scheme of Figure 7 has been updated for improved readability and aesthetics.
> 3. **Added Gridlines to Line Charts:** Gridlines have been incorporated into Figures 4, 5, and 6 to make the charts easier to interpret and review.
>
> We appreciate your feedback and hope these enhancements improve the overall presentation of the figures.
>
> **Q(b)**. See in W2.
>
> **Q(c). Include DLinear in Figure 4.**
>
> **Thank you for your suggestion regarding Figure 4.  We have updated the figure to include DLinear in the model efficiency comparison under ultra-long input lengths.** We look forward to your review of the revised PDF.
>
> **Q(d). Explanation of How to Apply TimeBase to MTS in Section 3**
>
> Thank you for your insightful advice. As detailed in Section 5.1 of the manuscript, “we adopt the Channel Independence strategy proposed by Nie et al. [2] to simplify the forecasting of multivariate time series (MTS) data into a series of univariate forecasting tasks.”  **To enhance clarity for readers, we have added the following explanation to Section 3.1:**
> “Most existing multivariate time series are homogeneous, meaning that each sequence within the dataset exhibits similar periodicity [3]. This characteristic allows them to be organized as a unified multivariate time series. Based on this property, we employ the Channel Independence [2] to simplify the forecasting of MTS data into separate univariate forecasting tasks.”
> We hope this addition provides the necessary clarity and strengthens the workflow description. Thank you again for your valuable suggestion!
>
> **Q(e). Multiple Seasonalities for TimeBase**
>
> Thank you for your insightful observation. When accounting for data with multiple seasonalities, TimeBase can be extended to learn distinct period bases for each seasonality and combine their predictions to enhance forecasting accuracy. This approach can be expressed as:
>
> $\text{MSTimeBase} = \sum_{i} \text{TimeBase}(\mathbf{X}; \mathbf{P}=p_i)$
>
> We tested this method on the *Traffic* dataset, incorporating multiple seasonalities ($p \in [24, 168]$), and the results are summarized in the table below:
>
> **Table Performance of TimeBase extended to multi-seasonality. The prediction length is 720 on Traffic dataset.**
>
> |    Model     |    MSE    |    MAE    |    MACs    |   Params   | Basis\_num |
> | :----------: | :-------: | :-------: | :--------: | :--------: | :--------: |
> | iTransformer | **0.450** |   0.313   |   1.01 G   |  11.61 M   |     -      |
> |   TimeBase   |   0.456   |   0.301   | **9.93 M** |   0.51 K   |     8      |
> |  MSTimeBase  | **0.451** | **0.295** |  16.76 M   | **0.49 K** |     6      |
>
> As demonstrated, TimeBase, when extended to consider multiple seasonalities, achieves improved prediction performance with only a modest increase in computational cost. **This extension highlights the flexibility of TimeBase in capturing complex periodic features in LTSF.** We sincerely thank the reviewer for the constructive feedback. **The further detailed analysis of   Multiple Seasonalities for TimeBase has been added to Appendix H.** We also emphasize that we will continue to explore the design of lightweight models based on multiple seasonalities to further enhance forecasting capabilities.
>
> [1] FITS: Modeling Time Series with $10k$ Parameters, ICLR 2024.
>
> [2] A Time Series is Worth 64 Words: Long-term Forecasting with Transformers, ICLR 2023.
>
> [3] TimesNet: Temporal 2D-Variation Modeling for General Time Series Analysis, ICLR 2023.
>
>
>
> Best Regards,
>
> Author of Paper 323

---

> ### Author Response · Authors · 2024-11-24
> **Thanks again for Reviewer tE5i**
>
> Dear Reviewer tE5i,
>
> Thank you once again for your valuable feedback on improving TimeBase. To save your precious time, we have summarized our improvements below and look forward to further discussions with you:
>
> ---
> 1. **Efficiency with Respect to FFT**
>
> We have added an explanation in **Section 5.3** to clarify why FFT runtime is not included in the analysis. Besides, the execution time for FFT on the entire training part of dataset (Electricity dataset) averaged approximately **0.08 seconds** across three runs.
>
> ---
> 2. **Effectiveness of the Orthogonal Constraint**
>    We conduct more reliabe experiments showing that applying the orthogonal constraint improved the MSE on ETTh1 and ETTh2 by an average of **7.36%** and **5.13%**. Furthermore, we have added detailed explanations of the **motivation behind the orthogonal constraint** in the methodology (**Section 3**), provided additional theoretical analysis showing that its error upper bound is closely related to the orthogonality of the basis(**Section 4 and Appendix B.3**), and included more comprehensive experimental results (**Fig 6 and Fig 7 in Section 5**). Correspondingly, the debuged code is also available at the anonymous link.
> ---
> **3. Unified Look-Back Window Length**
>
> We standardized the input sequence length across all models to 96 (**Appendix P**), 336 (**Tables 22 and 23 in Appendix O**), and 720 (**Table 21 in Appendix N and resluts in Section 5**). This ensures an absolutely fair comparison, demonstrating TimeBase's outstanding predictive performance and efficiency advantages.
>
> ---
> **4. Key advantage of TimeBase**
>
> - **Competitive Predictive Performance:**   TimeBase delivers predictive accuracy that is comparable to, and in some cases exceeds, certain SOTA models.(**Results of Section 5.3 and Appendix N, Appendix O, Appendix P**).
> - **Extreme Efficiency:** The goal of TimeBase is to construct an exceptionally lightweight forecasting model (0.39K parameter, 2.77M MACs, 0.98ms inferring time) without compromising predictive performance, which is friendly to employ in edge devices.(**Results of Section 5.3 and Appendix N, Appendix O, Appendix P**).
> - **Serve as Effective Plug-and-play Tool:** Moreover, it is designed as a plug-and-play solution  that can significantly reduce the model complexity of patch-based methods, i.e., PatchTST (**77.74%~93.00% reduction of MACs**) （**Appendix M**）.
>
>
> ---
> **5. Include DLinear in Figure 4.**
>
> We have updated the figure to include DLinear in the model efficiency comparison under ultra-long input lengths (**Figure 4**).
>
> ---
>
> **6. Explanation of How to Apply TimeBase to MTS in Section 3**
>
> “Most existing multivariate time series are homogeneous, meaning that each sequence within the dataset exhibits similar periodicity. This characteristic allows them to be organized as a unified multivariate time series. Based on this property, we employ the Channel Independence to simplify the forecasting of MTS data into separate univariate forecasting tasks.” (**Updated to Section 3**)
>
> ---
>
> **7. Multiple Seasonalities for TimeBase**
>
> We extend Timebase to process Multiple Seasonalities, and report the corresponding results, showing that TimeBase, when extended to consider multiple seasonalities, achieves improved prediction performance with only a modest increase in computational cost. (**in Appendix H**)
>
> ---
>
> We sincerely thank you for taking the time to review our paper and provide valuable feedback. **We genuinely hope our responses address your concerns and look forward to further engaging with you.** Thank you once again!
>
>
> Sincerely,
> Author of Paper 323.

---

> > ### Comment · Reviewer_tE5i · 2024-11-26
> >
> > Thank you to the authors for their efforts in addressing my concerns. I have updated the rating accordingly.

---

> > > ### Author Response · Authors · 2024-11-26
> > > **Thanks for your feedback**
> > >
> > > Dear Reviewer tE5i,
> > >
> > > We sincerely appreciate your valuable feedback on our proposed improvements. Your insightful suggestions have greatly enhanced the quality of the **TimeBase**.
> > >
> > > Best regards,
> > > The authors of Paper 323

---

> ### Author Response · Authors · 2024-11-29
> **Contributions of TimeBase for Reviewers and Area Chair**
>
> Dear All Reviewers and Area Chair,
>
> Thanks to **all the reviewers for raising your scores and acknowledging our improvements**, which is a tremendous encouragement to us. We believe TimeBase offers significant advantages (**extreme efficiency, plug-and-play usability, reproducibility**) and can positively contribute to the time series community. Therefore, we would like to **briefly highlight** its key strengths and potential impact, and **kindly ask for your final judgment on TimeBase**.
>
> ---
>
>
> **1. Competitive Predictive Performance and Extreme Lightness**
>
> TimeBase not only demonstrates strong predictive performance (under unified input lengths of **96**, **336**, or **720**, full results in Appendix N, O, and P), but it also boasts an exceptionally lightweight complexity (Section 5.3): TimeBase have **0.39K parameters, 2.77M MACs computation, 88.89M memory usage, and 0.98ms CPU inference speed**. Take MACs for example, its MACs is **1/120 of DLinear**, **1/646 of iTransformer**, and **1/5115 of PatchTST**. The effective utilization of time series data makes it suitable for complex forecasting scenarios. Whether in cloud-based data centers or resource-constrained edge devices, **TimeBase can deliver accurate forecasting results with extremely low resource consumption**.
>
> **2. Plug-and-Play Usability**
>
> TimeBase can not only function as **a time-series forecaster** but also serve as **an efficient complexity reducer** for patch-based methods (Appendix M). In implementation, TimeBase can be expertly used to extract key basis components after patching the entire time series of patch-based models. Experimental results show that TimeBase reduces the computation of PatchTST to just **1/9** of its original MACs.**(e.g., from 14.17G to 1.58G)**, while improving prediction accuracy in **21** of the total **28 forecasting settings**. The remaining 7 settings show only trivial differences in metrics. **This demonstrates TimeBase's ability to significantly reduce the resource consumption of any patch-based model without sacrificing performance.**
>
>
>
> **3. Inspiration for Future Model Design**
>
> TimeBase reveals that long time series often exhibit temporal low-rank characteristics, and **it is the first to propose improving data and computational resource utilization by extracting basis components.** It also provides a **theoretical explanation** of how this approach is **applicable to any time series** (Appendix B.3). This **provides a potential strategy for designing efficient models and could offer valuable insights for the backbone design of large pre-trained time-series models** in the future works.
>
>
> ---
>
> **We sincerely hope that TimeBase will be recognized by you** and contribute to the development of the time series community. If you still believe TimeBase does not meet ICLR's acceptance standards, we would greatly appreciate your honest feedback to help us evaluate TimeBase from multiple perspectives. **Finally, we fully respect your scoring decision.** Thank you very much!
>
> Best Regards,
> Author of Paper 323

---

### Official Review · Reviewer_VkmF · 2024-11-05

**Soundness:** 3
**Presentation:** 3
**Contribution:** 2
**Rating:** 5
**Confidence:** 5

**Summary:**

This submission presents TimeBase, an ultra-lightweight network for long-term time series forecasting (LTSF). Traditional LTSF models often use a large number of parameters to capture temporal dependencies. However, the authors argue that time series data usually has periodic and low-rank structures and doesn't require so many parameters. TimeBase extracts core periodic features by leveraging full-rank typical period representations under orthogonality constraints to achieve accurate predictions. Experimental results show that TimeBase achieves minimalism in both model size and computational cost.

**Strengths:**

1. The authors propose a new lightweight time series forecasting model, especially based on the decomposed of basis and period

**Weaknesses:**

1. The proposed network is extremely light, but the significant difference between MLP and MLP-mixer work could be discussed.
2. The design of orthogonal constraints lakes direct motivation. Although the authors claim the orthogonal term could enhance the capturing of essential temporal patterns. However, the evaluation of the experiments (Figure 6) indicates that it may not be a successful component from $\lambda = 0$.
3. section  4, namely theoretical analysis, is not a serious mathematical proof. The overall process is more like a derivation than a complete proof. The primary support for Lemma 1 and Lemma 2 is finished by observation and assumption.
4. For a lightweight model, much further prediction length must be analyzed to demonstrate its real effects on LSTF.

**Questions:**

See above.

---

> ### Author Response · Authors · 2024-11-20
> **Response to  Reviewer VkmF 1/4**
>
> Dear Reviewer VkmF,
>
> Thanks very much for your positive feedback, we have provided additional discussions and responses to your concerns.
>
> **W1. TimeBase is extremely light, but should be discussed with the difference between MLP and Mixer work.**
>
> Thanks for your insightful guidance on further discussion of MLP-Mixer works. **In related work of revised version, we have added necessary discussions on well-known MLP- and Mixer-based works** (e.g., DLinear[1], TimeMixer[2], Koopa[3], TSMixer[6], MTS-Mixer[5], HDMixer[7], TiDE[2]) and highlight the distinctions between these approaches and TimeBase.
>
> > On the other hand, many MLP-based models have emerged, aiming to achieve lightweight forecasting solutions. DLinear [1] introduces a linear model based on trend and seasonal decomposition, whose competitive forecasting performance empirically demonstrated the feasibility of using MLPs for LTSF.  Following this, TiDE [3] provides theoretical proof that the simplest linear analogue could achieve near-optimal error rates for linear dynamical systems.  Later, numerous Mixer-based works emerge, such as  MTS-Mixer [5], TSMixer [6] and HDMixer [7], which  stack standard MLP layers to efficiently capture correlations across different dimensions of multivariate time series.   Furthermore, Koopa [4] addresses the challenge of dynamic and unstable time series systems by disentangling time-variant and time-invariant components using Fourier filters and designing a Koopman Predictor to advance the respective dynamics.  TimeMixer [2] tackles the issue of different granularity levels in micro and macro series by proposing mixing  blocks, fully leveraging disentangled multi-scale series in both past extraction and future prediction phases. These works represent efficient time series forecasting models based on MLP structures (1.03 M $\sim$ 31.07 M). However, it still remains challenging when faced with stricter deployment constraints on edge devices and higher efficiency demands.
>
> Furthermore, we summarize the differences between TimeBase and other MLP/Mixer-based models in the table below. The table includes comparisons on model scale (model parameter number of 720-horizon forecasting for Electricity), performance (MSE of 720-horizon forecasting for Electricity), forecasting type (segment-level or point-level). It shows that TimeBase deeply exploits the periodic properties of long-term time series, achieving good forecasting performance with an **extremely lightweight architecture**, 1000x smaller (0.39 K) than all mlp-based model (1.03 M $\sim$ 31.07 M).  **This table has been added to Appendix A.** Thanks again for your constructive advice.
>
> **Table Differences between TimeBase and other MLP-based Models**
>
>
> | **Linear-based Model** | **TimeBase (Ours)** | TimeMixer   | Koopa       | DLinear     | MTS-Mixer   | TSMixer     | HDMixer     | TiDE        |
> | ---------------------- | ------------------- | ----------- | ----------- | ----------- | ----------- | ----------- | ----------- | ----------- |
> | **Scale**              | **Extremely Light** | Light       | Normal      | Light       | Light       | Light       | Light       | Normal      |
> |                        | (0.39K)             | (5.57M)     | (30.04M)    | (1.03M)     | (2.02M)     | (1.05M)     | (4.81M)     | (31.07M)    |
> | **Performance**        | **Perfect**         | Perfect     | Perfect     | Perfect     | Good        | Good        | Good        | Good        |
> |                        | (0.208)             | (0.206)     | (0.215)     | (0.209)     | (0.213)     | (0.236)     | (0.243)     | (0.241)     |
> | **Forecasting Type**   | **Segment-level**   | Point-level | Point-level | Point-level | Point-level | Point-level | Point-level | Point-level |
>
>
>
>
> [1] Are transformers effective for time series forecasting? AAAI 2023.
>
> [2] TimeMixer: Decomposable Multiscale Mixing for Time Series Forecasting, ICLR 2024.
>
> [3] Long-term Forecasting with TiDE: Time-series Dense Encoder, TMLR 2023.
>
> [4] Koopa: Learning non-stationary time series dynamics with koopman predictors, NeurIPS 2023.
>
> [5] MTS-mixers: Multivariate time series forecasting via factorized temporal and channel mixing, Arxiv 2023.
>
> [6] TSMixer: Lightweight mlp-mixer model for multivariate time series forecasting, KDD 2023.
>
> [7] HDMixer: Hierarchical dependency with extendable patch for multivariate time series forecasting, AAAI 2024.

---

> ### Author Response · Authors · 2024-11-20
> **Response to  Reviewer VkmF 2/4**
>
> **W2. The Effectiveness of Orthogonal Constraint.**
>
> Thank you for your interest in the Orthogonal Constraint. **We apologize for the difficulty in directly observing the impact of $\lambda=0$ on the model in Figure 4, as the large range of the y-axis makes it less apparent. We have provided the numerical values for further clarification.** Overall, the effectiveness of the Orthogonal Constraint can be explained from three aspects: (1) Intuitive motivation, (2)Theoretical error bound, and (3) Experimental results.
>
> **(1) Intuitive Motivation**
>
> From the perspective of the data space, the orthogonality of the basis vectors enhances its representation power, providing them ability to express as any vector in the data space  through linear combination. Therefore, the period basis should also be diverse and distinct, preventing the extraction of very single time-series patterns. Based on this, we apply the Orthogonal Constraint. In the time-series space, each time series period (segmentation) can be viewed as a composition of several orthogonal period (segmentation) basis vectors. The goal of TimeBase is to transform the long-term forecasting task at the time-step level into a task of basis extraction and prediction at the period level, thus enabling an efficient time-series forecasting model (0.39K, 1000 times smaller than MLP/Mixer-based LTSF models).
>
> **(2) Theoretical Error Bound**
>
> Given a historical time series $\mathbf{X}_\text{his} \in \mathbb{R}^{N \times P}$, its corresponding learned basis vectors $\mathbf{E}\in \mathbb{R}^{R \times P}$,
>
> and furture time series $\mathbf{X}_\text{pred} \in \mathbb{R}^{N' \times P}$, the upper fitting error of the model can be expressed as:
>
> $||\mathbf{r}||\_2 \leq \frac{1}{\lambda_{\min}(\mathbf{E} \mathbf{E}^T)}||\mathbf{X}_{\text{pred}}||\_2$
>
> where $\lambda_{\min}(\mathbf{E} \mathbf{E}^T)$ denotes the smallest eigenvalue of the Gram matrix $\mathbf{E} \mathbf{E}^T$. This result highlights that the generalization capability of TimeBase to arbitrary time series relies on learning a well-represented basis matrix $\mathbf{E}$. If $\mathbf{E}$ exhibits a favorable eigenvalue distribution (i.e., $\lambda_{\min}(\mathbf{E} \mathbf{E}^T)$ is large), the upper bound on prediction error is lower, **highlighting the importance of   a high-quality basis vector space and the necessity of orthogonal constraint.** **This analysis hae been added to section 4  and he detailed derivation   is available in Appendix B.3 of revised PDF.**
>
> **(3) Experimental Results**
>
> In the original paper, due to two issues—(1) the vertical axis scale in Figure 6 being too large, which made the change at lambda=0 not visible to the naked eye, and (2) a bug in the original code's test loader with the `drop_last=True`, causing the model not to be evaluated on the full set of test samples, especially with a large batch size (512). We have since fixed the bug in the code and report more reliable results in revised PDF. Fixed code and running script is available at [https://anonymous.4open.science/r/TimeBase-fixbug](https://anonymous.4open.science/r/TimeBase-fixbug).
>
> |                       | ETTh1 \| 96 | ETTh1 \| 192 | ETTh1 \| 336 | ETTh1 \| 720 | ETTh2 \| 96 | ETTh2 \| 192 | ETTh2 \| 336 | ETTh2 \| 720 |
> | :-------------------: | :---------: | :----------: | :----------: | :----------: | :---------: | :----------: | :----------: | :----------: |
> |  Without Constraint   |    0.371    |    0.423     |    0.444     |    0.461     |    0.305    |    0.352     |    0.380     |    0.412     |
> | Orthogonal Constraint |    0.349    |    0.387     |    0.408     |    0.439     |    0.292    |    0.342     |    0.358     |    0.400     |
> |        Improve        |   +0.022    |    +0.036    |    +0.036    |    +0.022    |   +0.013    |    +0.010    |    +0.022    |    +0.012    |
>
> As shown in the table, after applying the orthogonal constraint, the MSE for etth1 and etth2 improved by an average of 7.36% and 5.13%, respectively. This confirms that the Orthogonal Constraint significantly improves TimeBase’s representation power, maximizing the forecasting potential of its extremely small parameter size.
>
> Thanks again for your insightful advice. To provide readers with a more intuitive understanding, **we have included additional explanations  of the motivation behind the Orthogonal Constraint in  method designing (Section 3), more theoretical analysis (Section 4 and Appendix B.3),  more reliable experiments (Section 5)**

---

> ### Author Response · Authors · 2024-11-20
> **Response to  Reviewer VkmF 3/4**
>
> **W3. Modification on Section 4**
>
> Thank you for your valuable suggestions to improve our theoretical analysis. Actually, the purpose of Section 4 is to analyze the feasibility of TimeBase. To make the presentation more rigorous, **we have re-organized and simplified the original effectiveness analysis in section 4. Additionally, to demonstrate the generalization of TimeBase, we have included an more theoretical analysis of its convergence on  general time series data.** The findings indicate that when the learned period bases are orthogonal, TimeBase achieves a smaller upper bound on the error. We look forward to your re-evaluation of Section 4 in the revised PDF.
>
> **W4. Much Further Prediction length**
>
> Thank you for your constructive feedback on TimeBase. To further highlight its advantages in long-term time series forecasting, we extend the maximum prediction horizon from 720  to 1080, 1440, and 1800 horizon and compare its performance with well-known LTSF models, i.e., iTransformer and DLinear. **The results in table  demonstrate that TimeBase offers significant advantages in much further prediction, featuring linear model complexity (in terms of MACs and scale) and delivering more accurate predictive results with the increasing of output length.** We are committed to further deploying TimeBase on edge devices for practical long-term forecasting scenarios. These findings validate TimeBase's effectiveness in ultra-long-term forecasting tasks, particularly when resource efficiency is critical. Moreover, the linear growth in computational cost ensures its feasibility for deployment on edge devices. This positions TimeBase as a practical solution for real-world long-term forecasting scenarios. Thank you for your valuable suggestion. **We have kindly included the analysis of predictions for longer sequence lengths in the Appendix G.**
>
>
> |              |           | Electricity \| 1080 |            |           | Electricity \| 1440 |            |           | Electricity \| 1800 |            |
> | :----------: | :-------: | :-----------------: | :--------: | :-------: | :-----------------: | :--------: | :-------: | :-----------------: | :--------: |
> |              |    MSE    |        Param        |    MAC     |    MSE    |        Param        |    MAC     |    MSE    |        Param        |    MAC     |
> | **TimeBase** | **0.234** |      **0.5 K**      | **3.47 M** | **0.264** |      **0.6 K**      | **4.16 M** | **0.295** |      **0.7 K**      | **4.85 M** |
> | iTransformer |   0.253   |       5.65 M        |   1.85 G   |   0.272   |       5.84 M        |   1.91 G   |   0.325   |       6.03 M        |   1.97 G   |
> |   DLinear    |   0.255   |        1.6 M        |  499.45 M  |   0.29    |        2.1 M        |  665.86 M  |   0.321   |       2.59 M        |  832.26 M  |
>
> |              |           | ETTh1 \| 1080 |            |           | ETTh1 \| 1440 |            |           | ETTh1 \| 1800 |           |
> | :----------: | :-------: | :-----------: | :--------: | :-------: | :-----------: | :--------: | :-------: | :-----------: | :-------: |
> |              |    MSE    |     Param     |    MAC     |    MSE    |     Param     |    MAC     |    MSE    |     Param     |    MAC    |
> | **TimeBase** | **0.551** |   **0.5 K**   | **0.07 M** | **0.636** |   **0.6 K**   | **0.09 M** | **0.714** |   **0.7 K**   | **0.1 M** |
> | iTransformer |   0.602   |    431.1 K    |   5.58 M   |   0.708   |    477.4 K    |   6.18 M   |   0.812   |    523.9 K    |  6.78 M   |
> |   DLinear    |   0.582   |     160 M     |  10.89 M   |   0.693   |     2.1 M     |  14.52 M   |   0.796   |    2.59 M     |  18.15 M  |
>
>
> |              |           | ETTh2 \| 1080 |            |           | ETTh2 \| 1440 |            |           | ETTh2 \| 1800 |           |
> | :----------: | :-------: | :-----------: | :--------: | :-------: | :-----------: | :--------: | :-------: | :-----------: | :-------: |
> |              |    MSE    |     Param     |    MAC     |    MSE    |     Param     |    MAC     |    MSE    |     Param     |    MAC    |
> | **TimeBase** | **0.478** |   **0.5 K**   | **0.07 M** | **0.543** |   **0.6 K**   | **0.09 M** | **0.552** |   **0.7 K**   | **0.1 M** |
> | iTransformer |   0.501   |    431.1 K    |   5.58 M   |   0.575   |    477.4 K    |   6.18 M   |   0.597   |    523.9 K    |  6.78 M   |
> |   DLinear    |   0.583   |     1.6 M     |  10.89 M   |   0.672   |     2.1 M     |  14.52 M   |   0.652   |    2.59 M     |  18.15 M  |

---

> ### Author Response · Authors · 2024-11-20
> **Response to  Reviewer VkmF 4/4**
>
> **E1. Additional Discussion**
>
> **Finally, we sincerely thank you for your constructive feedback on TimeBase, which has guided us in providing necessary discussion in MLP-based mothods, and refining our experimental setup. Your thoughtful suggestions have been invaluable.**
>
> In addition, we would like to engage in an extra discussion with you regarding an intriguing discovery. **We found that TimeBase can serve as a plug-and-play tool for patch-based models for extreme complexity reduction.** To validate this idea, we integrated TimeBase into PatchTST and conducted comparative experiments. In implementation, after segmenting the time series into patches, TimeBase  can expertly be employed to patch-based methods to extract the basis components.
>
> The experimental results, partially shown in the table below, demonstrate that PatchTST combined with TimeBase can significantly reduce computational resource consumption while maintaining or even improving prediction accuracy. **Detailed experiments have been included in Appendix M, and we look forward to your review.**
>
> **Performance of TimeBase as a Plug-and-Play Component for Patch-Based Methods. The input length is set as 720, and the output length is 96. Full results are available at Appendix M**
>
> |             |       | PatchTST+TB |        |       | PatchTST |        |Complexity|  Reduction        |
> | ----------- | :---: | :---------: | :----: | :---: | :------: | :----: | :-------: | :----: |
> |             |  MSE  |    MACs     | Params |  MSE  |   MACs   | Params |   MACs    | Params |
> | ETTh1       | 0.364 |   0.77 M    | 0.03 M | 0.377 | 11.05 M  | 0.15 M |  93.00%   | 83.15% |
> | ETTh2       | 0.275 |   1.28 M    | 0.03 M | 0.276 | 11.05 M  | 0.15 M |  88.42%   | 79.08% |
> | ETTm1       | 0.29  |   28.87 M   | 0.52 M | 0.298 | 258.69 M | 1.51 M |  88.84%   | 65.20% |
> | ETTm2       | 0.165 |   57.59 M   | 0.65 M | 0.165 | 258.69 M | 1.51 M |  77.74%   | 57.07% |
> | Weather     | 0.145 |   86.60 M   | 0.52 M | 0.149 | 776.08 M | 1.51 M |  88.84%   | 65.20% |
> | Electricity | 0.128 |   1.32 G    | 0.52 M | 0.141 | 11.86 G  | 1.51 M |  88.84%   | 65.20% |
> | Traffic     | 0.36  |   4.27 G    | 0.55 M | 0.363 | 31.86 G  | 1.51 M |  86.61%   | 63.57% |
>
>
>
> Best Regards,
>
> Author of Paper 323

---

> ### Author Response · Authors · 2024-11-24
> **Thanks again for Reviewer VkmF**
>
> Dear Reviewer VkmF,
>
> Thank you once again for taking the time to review our paper and provide valuable feedback. **We understand the importance of your time and have summarized our responses below to facilitate your review and help you locate the points of interest efficiently**:
>
> ---
> 1. **Discussion on MLP and Mixer Models**
> We expanded the **related work in Section 2**  to discuss MLP and Mixer-based models and further summarized their differences from TimeBase in **Appendix A**. TimeBase’s key advantage lies in its extreme lightweight design, being **1,000× smaller** (**0.39K** parameters) than MLP/Mixer-based models (**1.03M–31.07M**), enabling efficient segment-level rather than point-level forecasting.
> ---
> 2. **Effectiveness of the Orthogonal Constraint**
>    We conduct more reliabe experiments showing that applying the orthogonal constraint improved the MSE on ETTh1 and ETTh2 by an average of **7.36%** and **5.13%**. Furthermore, we have added detailed explanations of the **motivation behind the orthogonal constraint** in the methodology (**Section 3**), provided additional theoretical analysis showing that its error upper bound is closely related to the orthogonality of the basis(**Section 4 and Appendix B.3**), and included more comprehensive experimental results (**Fig 6 and Fig 7 in Section 5**). Correspondingly, the debuged code is also available at the anonymous link.
> ---
> 3. **Modifications to Section 4**
>    We have **reorganized and simplified the original effectiveness analysis in Section 4**. To further demonstrate TimeBase’s generalizability, we included additional theoretical analysis showing that its error upper bound is closely related to the orthogonality of the basis (**Section 4 and Appendix B.3**).
> ---
> 4. **Extending Prediction Length**
>    We extended the maximum prediction horizon to **1,080**, **1,440**, and **1,800** steps to validate TimeBase’s superior accuracy and efficiency for longer forecasting lengths (**Appendix G**).
> ---
> 5. **Additional Exploration: Serve as Effective Plug-and-play Tool**
>  Additionally, TimeBase can also serve as a very effective plug-and-play tool for patch-based forecasting methods, enabling extreme complexity reduction, i.e., PatchTST (**77.74%~93.00% reduction of MACs**),  and even improving prediction accuracy  (**Appendix M**).
>
> ---
> We sincerely appreciate your insightful suggestions, which have significantly contributed to improving our work. **We look forward to further discussions with you to address your concerns or improve our work based on your additional feedback.**
>
> Sincerely,
> Author of Paper 323.

---

> ### Comment · Reviewer_VkmF · 2024-11-28
> **Response**
>
> Your additional experiments have addressed some concerns. However, I still think the motivation of the proposed methods is weak, and the clear difference from the regression method is unclear. I will raise my score for the previous improvement.

---

> > ### Author Response · Authors · 2024-11-29
> > **Thanks for your response!**
> >
> > Dear Reviewer VkmF,
> >
> > We also greatly appreciate your positive response to our rebuttal, which is a tremendous encouragement to us. In this reply, we will try our best   to address your remaining concerns:
> >
> >
> > ---
> >
> > **(1) Motivation of TimeBase**
> >
> > - **Exploring Temporal Data.** The motivation for TimeBase stems from our continuous exploration and analysis of real-world temporal data. As illustrated in Figure 1 of the introduction, long-term forecasting typically involves data that exhibits periodicity, and these periods often share similar patterns (with a similarity of **0.76$\sim$0.99**). This leads to the observation of **periodicity** and **approximate low-rank** characteristics, where such redundancy may result in the waste of computational resources.
> >
> > - **Basis Vector Hypothesis.** To address this issue, we hypothesize that these periodic patterns can be represented by a linear combination of  basis components from the data space. This insight led us to propose a framework where historical series are used to reconstruct the  basis components, and future periods are predicted based on this basis.
> >
> > - **Experimental Validation.** Based on this, we designed TimeBase, which only utilizes **two small scale linear layers**: one for extracting basis components and the other for segment-level prediction. This results in a model with **comparable predictive power** and an extremely small size (**0.39k** parameters, **2666x, 22282x, 14025x, 13000x** smaller than DLinear, PatchTST, iTransformer, TimeXer, respectively).
> >
> > - **Theoretical Analysis.** Moreover, theoretical analysis demonstrates that **this approach can also be applied to non-periodic data**, where basis extraction and segment-level prediction can be performed. We prove that the **model's error upper bound is positively correlated with the orthogonality of the extracted basis vectors (Appendix B.3).**
> >
> > We focus on designing a simple yet efficient model to tackle complex forecasting problems, and we hope this data-driven approach could provide valuable insights for other researchers.
> >
> > ---
> >
> > **(2) Differences from Regression Methods**
> >
> > The key differences between TimeBase and regression methods (e.g., DLinear, TimeMixer, MTX-Mixer, and even a simple regression linear layer)  can be summarized as follows:
> >
> > ---
> >
> > 1. **Prediction Mode (Section 3.1)**:
> >
> > **TimeBase is segmentation-level forecasting, and the others  are redundant point-level forecasting.**
> > Specifically, TimeBase emphasizes simplification in the number of segments and relies on basis components for prediction. **In contrast**, all regression methods perform point-level predictions, overlooking long-term temporal characteristics, which leads to redundant modeling. These methods directly extract features along the time dimension and use a linear layer to map the flattened input features to the future sequence length.
> >
> > ---
> >
> > 2. **Model Complexity (Section 5.3)**:
> >
> > **TimeBase is an efficient forecasting model that can be deployed in resource-constrained environments.** TimeBase's parameter count is approximately $\frac{(L+T)R}{P}$, achieving very efficient CPU inference speed (**0.98ms**) and minimal computation (**2.77M MACs**). **In comparison**, other regression models have parameter counts of at least \( L \times T \), and often much more parameters due to the stacked layers and additional modules. For example, DLinear, the lightest of these models, has a computational cost of **333.04M** MACs and a CPU inference speed of **3.25ms**.
> >
> > ---
> > 3. **Scalability (Appendix M)**:
> >
> > Unlike other regression models with low scalability, TimeBase is **highly scalable and can be used as a plug-and-play module to significantly reduce the complexity of patch-based methods**, maintaining or even improving prediction performance. For instance, TimeBase reduces the computational cost of PatchTST by **77.74%** $\sim$ **93.16%** (e.g., from 14.17G to 1.58G) and enhances prediction accuracy in **21** out of **28** forecasting settings (the other 7 settings show trivial differences in metrics).
> >
> > ---
> > 4. **Potential Impact (Contributions to the Future Development of the Time Series Community)**:
> >
> > The exist of other linear regression models (e.g., DLinear, TimeMixer, MTX-Mixer) suggests  that linear regression can also yield good prediction results in long-term series forecasting. **In contrast, TimeBase demonstrates that by leveraging low-rank temporal characteristics and extracting key basis components, the efficiency of linear-based models can be further significantly and greatly improved** (3x inference speed). This represents a further advancement in linear-based models for long-term time series forecasting.
> >
> > ---
> > **We hope our additional clarifications could address your remaining concerns.** If you recognize the efforts we’ve made towards lightweight time-series forecasting, we sincerely hope you will provide a positive evaluation of our work. Thank you very much!
> >
> > Best Regards,
> > Author of Paper 323

---

### Author Response · Authors · 2024-12-01
**Thank You for Your Valuable Feedback and Support in Improving TimeBase**

Dear All Reviewers

We would like to extend our sincere thanks for your insightful feedback, which has greatly assisted us in refining TimeBase. We highly value this review opportunity, as **we received the highest confidence score in the entire time series domain (4.75, 1/280)**, which indicates that your feedback was thoughtfully considered. We have been continuously improving the quality of our paper, and we are pleased  that all reviewers have recognized our progress and awarded us higher scores. However, considering that our current score may not be sufficient for acceptance, we have decided to withdraw the paper in order to further enhance its content.

We truly appreciate your support throughout this process and will continue to work on efficient time series forecasting.

Best regards,

Authors of paper 323

---

### Note · Authors · 2024-12-01

I have read and agree with the venue's withdrawal policy on behalf of myself and my co-authors.